# Simple Guidance Mechanisms
# for Discrete Diffusion Models

**Yair Schiff**[\*][†]**, Subham Sekhar Sahoo**[†]**, Hao Phung**[†]**, Guanghan Wang**[†]**,
Alexander Rush, & Volodymyr Kuleshov**
Department of Computer Science, Cornell University
`{yzs2,sss284,gw354,htp26,amr459,vk379}@cornell.edu`

**Hugo Dalla-Torre, Sam Boshar, Bernardo P. de Almeida, & Thomas Pierrot**
InstaDeep
`{h.dalla-torre,s.boshar,b.dealmeida,t.pierrot}@instadeep.com`

## Abstract

Diffusion models for continuous data gained widespread adoption owing to their high quality generation and control mechanisms. However, controllable diffusion on discrete data faces challenges given that continuous guidance methods do not directly apply to discrete diffusion. Here, we provide a straightforward derivation of classifier-free and classifier-based guidance for discrete diffusion, as well as a new class of diffusion models that leverage uniform noise and that are more guidable because they can continuously edit their outputs. We improve the quality of these models with a novel continuous-time variational lower bound that yields state-of-the-art performance, especially in settings involving guidance or fast generation. Empirically, we demonstrate that our guidance mechanisms combined with uniform noise diffusion improve controllable generation relative to autoregressive and diffusion baselines on several discrete data domains, including genomic sequences, small molecule design, and discretized image generation. Code to reproduce our experiments is available here.

## 1 Introduction

Diffusion models (Sohl-Dickstein et al., 2015; Ho et al., 2020) gained widespread adoption in image generation and signal processing in part due to their high controllability via mechanisms such as classifier-based (Dhariwal & Nichol, 2021a) and classifier-free guidance (Nichol et al., 2021; Ho & Salimans, 2022). Tasks where guidance plays a key role include MRI denoising (Song & Ermon, 2019), 3D reconstruction (Poole et al., 2022; Gao et al., 2024), and conditional generation (Saharia et al., 2022; Gokaslan et al., 2024).

However, applying controllable diffusion-based generation to tasks where the data is discrete (e.g., molecule design or text generation) presents challenges. First, standard diffusion models and their guidance mechanisms are not directly applicable, since they require taking gradients with respect to the data, and these are not defined in discrete settings. Second, popular discrete extensions of diffusion (Sahoo et al., 2024a; Shi et al., 2024) cannot perform multiple editing passes on generated tokens, hence are not ideal for controllable generation. Third, the performance of discrete diffusion models (measured by perplexity) lags behind autoregressive (AR) models, especially for classes of diffusion that are amenable to control, such as uniform noise (Austin et al., 2021; Lou et al., 2023).

Here, we propose discrete diffusion models and guidance mechanisms that are effective at controllable generation and that address the above challenges. First, we provide straightforward and easy-to-implement adaptations of classifier-based and classifier-free guidance for discrete diffusion models. Second, we revisit uniform noise diffusion language models (UDLM), which undo random token perturbations and are particularly amenable to guidance, since they can repeatedly edit their samples (Austin et al., 2021) and thus correct errors. We address performance issues that plagued

---

[\*]Corresponding author. Work done while at InstaDeep. [†]Equal contribution.

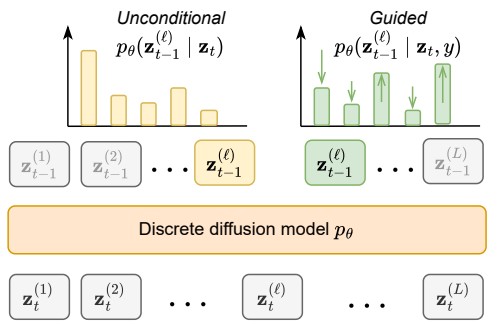 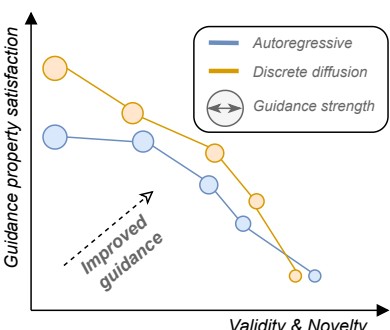

Figure 1: *(Left)* Adapting guidance to discrete diffusion. Models output a factorized discrete distribution for each denoised token. With our guidance mechanisms, we adjust these probabilities according to a guidance model – either a conditional diffusion model in classifier-free guidance (Section 3.1) or a separately trained classifier for classifier-based guidance (Section 3.2). *(Right)* Relative to autoregressive models, which make local predictions one token at a time, discrete diffusion models denoise the entire sequence at every iteration, allowing for more guidable outputs.

previous iterations of uniform noise discrete diffusion by deriving a continuous-time version of the evidence lower bound, which tightens the variational gap (Kingma et al., 2021; Sahoo et al., 2024a).

We demonstrate the effectiveness of guidance with discrete diffusion models on several domains: genomics, molecule strings, and discretized images. We find that discrete diffusion models are more controllable than AR when paired with classifier-free guidance. Moreover, our proposed classifier-based method improves upon previous guidance mechanisms for diffusion (Gruver et al., 2024) and AR (Dathathri et al., 2019; Yang & Klein, 2021). Our language modeling experiments also reveal that contrary to a widely-held belief (Austin et al., 2021; Lou et al., 2023), uniform noise diffusion can attain state-of-the-art performance on small vocabulary datasets (e.g,. molecules, DNA) and that UDLM attains a new state-of-the-art in perplexity among uniform noise diffusion models.

In summary, our contributions are as follows:

- We provide simple and effective discrete classifier-based and classifier-free guidance.
- We introduce UDLM, a class of discrete diffusion models particularly amenable to guidance, for which we derive a tightened ELBO that significantly improves their performance.
- Across three domains, we demonstrate that discrete guidance yields better controllable generation compared to strong AR baselines and previous diffusion guidance methods.

## 2 BACKGROUND

**Notation** Let $\mathcal{V}$ be the space of all one-hot tokens over some vocabulary consisting of $N$ unique characters: $\mathcal{V} = \{\mathbf{z} \in \{0,1\}^N : \sum_i \mathbf{z}_i = 1\} \subset \Delta^N$, where $\Delta^N$ represents the simplex over $N$ categories. Let $\mathbf{1}$ be a $N$-dimensional column vector of all ones, and denote the Hadamard product between two vectors as $\odot$. We define $\mathbf{z}^{(1:L)}$ as a sequence of $L$ tokens, where $\mathbf{z}^{(\ell)} \in \mathcal{V}$, for all tokens $\ell \in \{1, \ldots, L\}$, and use $\mathcal{V}^L$ to denote the set of all such sequences. Finally, let $\mathrm{Cat}(\cdot\,; p)$ denote the categorical distribution with probability vector $p \in \Delta^N$.

### 2.1 DISCRETE DIFFUSION MODELS

Diffusion models are a class of generative models defined by a denoising network $p_\theta$ that is trained to remove noise from latent variables $\mathbf{z}_t$. These latents are generated by a fixed corruption process $q$ that, starting from clean data $\mathbf{x}$ drawn from the data distribution $q(\mathbf{x})$, increasingly adds more noise to $\mathbf{z}_t$, as $t$ increases (Sohl-Dickstein et al., 2015; Song & Ermon, 2019; Ho et al., 2020).

In discrete denoising diffusion probabilistic models (D3PM; Austin et al. (2021)), the noising process is defined in terms of a transition matrix $Q_{t|s}$ whose $(i, j)^{\text{th}}$ entry is the probability of transitioning to the $i$-th state at time $t$ given the $j$-th state at time $s$. This induces a Markov corruption process where we have $q(\mathbf{z}_t \mid \mathbf{z}_s) = \mathrm{Cat}(\mathbf{z}_t; Q_{t|s}\mathbf{z}_s)$.

Sahoo et al. (2024a) build off this framework to introduce specialized algorithms that are both simpler and more effective than the general D3PM framework. They focus on a specific class of forward processes from D3PM that can be defined as interpolations between clean data and a noisy prior $\boldsymbol{\pi}$, and we adopt their notation below:

$$q(\mathbf{z}_t|\mathbf{x}) = \text{Cat}(\mathbf{z}_t; \alpha_t \mathbf{x} + (1 - \alpha_t)\boldsymbol{\pi}), \qquad (1)$$

where $\alpha_t = \alpha(t)$ is a noise schedule monotonically decreasing in $t$. Defining $\alpha_{t|s} = \alpha_t/\alpha_s$, this class of processes admit the following posteriors

$$q(\mathbf{z}_s|\mathbf{z}_t, \mathbf{x}) = \text{Cat}\left(\mathbf{z}_s; \frac{[\alpha_{t|s}\mathbf{z}_t + (1 - \alpha_{t|s})\mathbf{1}\boldsymbol{\pi}^\top \mathbf{z}_t] \odot [\alpha_s \mathbf{x} + (1 - \alpha_s)\boldsymbol{\pi}]}{\alpha_t \mathbf{z}_t^\top \mathbf{x} + (1 - \alpha_t)\mathbf{z}_t^\top \boldsymbol{\pi}}\right). \qquad (2)$$

Of note, for absorbing-state diffusion, where $\boldsymbol{\pi} = [\text{MASK}]$, a one-hot vector at the special $[\text{MASK}]$ token index, Sahoo et al. (2024a) show that if $\mathbf{z}_t \neq [\text{MASK}]$ then $q(\mathbf{z}_s|\mathbf{z}_t, \mathbf{x}) = \text{Cat}(\mathbf{z}_s; \mathbf{z}_t)$, which reflects the fact that unasked tokens at time $t$ must remain unmasked for all time $s < t$.

## 2.2 Diffusion Guidance

For continuous data, diffusion models have demonstrated state-of-the-art controllable generation by means of classifier-based (Sohl-Dickstein et al., 2015; Dhariwal & Nichol, 2021a) and classifier-free guidance (Nichol et al., 2021; Ho & Salimans, 2022; Saharia et al., 2022). These approaches rely on different ways of expressing the score of a distribution conditioned on $y$.

**Classifier-based** Classifier-based generation employs a diffusion model to iteratively sample from a tempered distribution $p^\gamma(\mathbf{z}_s \mid y, \mathbf{z}_t) \propto p(y \mid \mathbf{z}_s)^\gamma p_\theta(\mathbf{z}_s \mid \mathbf{z}_t)$, where $\gamma$ represents an inverse temperature parameter, $p_\theta(\mathbf{z}_s \mid \mathbf{z}_t)$ is a pre-trained diffusion model, and $p(y \mid \mathbf{z}_s)$ is a classifier:

$$\nabla_{\mathbf{z}_s} \log p^\gamma(\mathbf{z}_s \mid y, \mathbf{z}_t) = \gamma \nabla_{\mathbf{z}_s} \log p(y \mid \mathbf{z}_s) + \nabla_{\mathbf{z}_s} \log p_\theta(\mathbf{z}_s \mid \mathbf{z}_t). \qquad (3)$$

**Classifier-free** We can also observe that appyling Bayes' rule to $p(y \mid \mathbf{x})$ and differentiating with respect to the input yields $\nabla_{\mathbf{x}} \log p(y \mid x) = \nabla_{\mathbf{x}} \log p(x \mid y) - \nabla_{\mathbf{x}} \log p(x)$. Applying this to $p(y \mid \mathbf{z}_s)$ and plugging into (3) gives us the formulation for classifier-free guidance:

$$\nabla_{\mathbf{z}_s} \log p^\gamma(\mathbf{z}_s \mid y, \mathbf{z}_t) = \gamma \cdot [\nabla_{\mathbf{z}_s} \log p_\theta(\mathbf{z}_s \mid y, \mathbf{z}_t) - \nabla_{\mathbf{z}_s} \log p_\theta(\mathbf{z}_s \mid \mathbf{z}_t)] + \nabla_{\mathbf{z}_s} \log p_\theta(\mathbf{z}_s \mid \mathbf{z}_t)$$
$$= \gamma \nabla_{\mathbf{z}_s} \log p_\theta(\mathbf{z}_s \mid y, \mathbf{z}_t) + (1 - \gamma) \nabla_{\mathbf{z}_s} \log p_\theta(\mathbf{z}_s \mid \mathbf{z}_t). \qquad (4)$$

with $p_\theta(\mathbf{z}_s \mid y, \mathbf{z}_t)$ conditionally and $p_\theta(\mathbf{z}_s \mid \mathbf{z}_t)$ unconditionally trained diffusion models.

## 3 Guidance Algorithms for Discrete Diffusion

Applying guidance to discrete diffusion is challenging because guidance terms are not differentiable with respect to the discrete representations $\mathbf{z}_t$ (Gruver et al., 2024). Here, we introduce two guidance algorithms for discrete diffusion that circumvent the issue of non-differentiability. As in Section 2.2, we formalize the guidance term as a probability $p(y|\mathbf{z}) \in \Delta^K$, where $y \in \{1, \dots, K\}$ is one of $K$ possible classes, and we begin with the distribution that controls the strength of the guidance term via the $\gamma$ temperature parameter (with $c$ denoting the logarithm of the normalization constant):

$$\log p^\gamma(\mathbf{z}_s \mid y, \mathbf{z}_t) = \gamma \log p(y \mid \mathbf{z}_s, \mathbf{z}_t) + \log p(\mathbf{z}_s \mid \mathbf{z}_t) + c. \qquad (5)$$

### 3.1 Classifier-free Guidance

**Guidance for a Single Token** To derive classifier-free guidance, we apply Bayes' rule to the first term on the right-hand side of (5):

$$\log p^\gamma(\mathbf{z}_s \mid y, \mathbf{z}_t) = \gamma \log p(\mathbf{z}_s \mid y, \mathbf{z}_t) - \gamma \log p(\mathbf{z}_s \mid \mathbf{z}_t) + \gamma \log p(y \mid \mathbf{z}_t) + \log p(\mathbf{z}_s \mid \mathbf{z}_t) + c.$$

Absorbing $\gamma \log p(y \mid \mathbf{z}_t)$ into the constant $c$ and grouping terms yields:

$$\log p^\gamma(\mathbf{z}_s \mid y, \mathbf{z}_t) = \gamma \log p(\mathbf{z}_s \mid y, \mathbf{z}_t) + (1 - \gamma) \log p(\mathbf{z}_s \mid \mathbf{z}_t) + c. \qquad (6)$$

**Extension to Sequences** In the derivation for (6), we can replace the single-token latent variables with sequences of multiple tokens:

$$
\log p^\gamma(\mathbf{z}_s^{(1:L)} \mid y, \mathbf{z}_t^{(1:L)}) = \gamma \log p(\mathbf{z}_s^{(1:L)} \mid y, \mathbf{z}_t^{(1:L)}) + (1 - \gamma) \log p(\mathbf{z}_s^{(1:L)} \mid \mathbf{z}_t^{(1:L)}) + c,
$$
$$
\implies p^\gamma(\mathbf{z}_s^{(1:L)} \mid y, \mathbf{z}_t^{(1:L)}) \propto p(\mathbf{z}_s^{(1:L)} \mid y, \mathbf{z}_t^{(1:L)})^\gamma p(\mathbf{z}_s^{(1:L)} \mid \mathbf{z}_t^{(1:L)})^{(1-\gamma)} \tag{7}
$$

**Discrete Classifier-Free Guidance** We can model the probabilities in (7) by training a conditional denoising diffusion network $p_\theta(\mathbf{z}_s^{(1:L)} \mid y, \mathbf{z}_t^{(1:L)})$ and an unconditional one $p_\theta(\mathbf{z}_s^{(1:L)} \mid \mathbf{z}_t^{(1:L)})$. In practice, we train these models in tandem by randomly 'dropping out' or 'masking' the conditioner during training to simulate the unconditional diffusion model. Each of these distributions is thus modeled to factorize independently along the sequence length dimension of $\mathbf{z}_s^{(1:L)}$ when conditioned on $\mathbf{z}_t^{(1:L)}$, allowing us to efficiently sample from the tempered guidance distribution as follows:

$$
p_\theta^\gamma(\mathbf{z}_s^{(1:L)} \mid \mathbf{z}_t^{(1:L)}, y) = \prod_{\ell=1}^L \frac{1}{Z^{(\ell)}} p_\theta(\mathbf{z}_s^{(\ell)} \mid y, \mathbf{z}_t^{(1:L)})^\gamma p_\theta(\mathbf{z}_s^{(\ell)} \mid \mathbf{z}_t^{(1:L)})^{(1-\gamma)}, \tag{8}
$$

where $Z^{(\ell)} = \sum_{\mathbf{z}_s'} p_\theta(\mathbf{z}_s' \mid \mathbf{z}_t^{(1:L)}, y)^\gamma p_\theta(\mathbf{z}_s' \mid \mathbf{z}_t^{(1:L)})^{(1-\gamma)}$ is the per-token partition function.

We refer to this method as **D-CFG** for **D**iscrete **C**lassifier-**F**ree **G**uidance.

## 3.2 CLASSIFIER-BASED GUIDANCE

**Guidance for a Single Token** Exponentiating (5) lets us draw classifier-guided samples from the following tempered distribution:

$$
p^\gamma(\mathbf{z}_s \mid \mathbf{z}_t, y) = \frac{p(y \mid \mathbf{z}_s, \mathbf{z}_t)^\gamma p(\mathbf{z}_s \mid \mathbf{z}_t)}{\sum_{\mathbf{z}_s'} p(y \mid \mathbf{z}_s', \mathbf{z}_t)^\gamma p(\mathbf{z}_s' \mid \mathbf{z}_t)}, \tag{9}
$$

which we can tractably normalize, as we only sum over the $N$ possible values of a single token $\mathbf{z}_s$.

**Extension to Sequences** Unfortunately, extending (9) to sequences is not trivial, since the classifier $p(y \mid \mathbf{z}_s^{(1:L)}, \mathbf{z}_t^{(1:L)})$ does not necessarily factorize independently across the sequence, which would potentially force us to compute a normalization constant with $N^L$ terms, accounting for every possible value of $\mathbf{z}_s^{(1:L)}$. To alleviate this issue, we require the assumption that conditioned on $\mathbf{z}_t^{(1:L)}$, the tempered distribution $p^\gamma(\mathbf{z}_s^{(1:L)} \mid \mathbf{z}_t^{1:L}, y)$ factorizes independently across tokens. Therefore, we can focus on the tempered distirbution of each token $\mathbf{z}_s^{(\ell)}$, for $\ell \in 1, \ldots, L$:

$$
p^\gamma(\mathbf{z}_s^{(\ell)} \mid \mathbf{z}_t^{(1:L)}, y) \propto p(y \mid \mathbf{z}_s^{(\ell)}, \mathbf{z}_t^{(1:L)})^\gamma p(\mathbf{z}_s^{(\ell)} \mid \mathbf{z}_t^{(1:L)}). \tag{10}
$$

**Discrete Classifier-Based Guidance** We first introduce the following additional notation: given $\mathbf{z}^{(1:L)}$, let $\tilde{\mathcal{Z}}_\ell(\mathbf{z}^{(1:L)})$ be the set of sequences $\tilde{\mathbf{z}}^{(1:L)}$ for which $\tilde{\mathbf{z}}^{(\ell')} = \mathbf{z}^{(\ell')}$ for all $\ell' \neq \ell$, i.e., the set of sequences that are either the same as or only differ in position $\ell$ relative to $\mathbf{z}^{(1:L)}$. We can sample from $p^\gamma(\mathbf{z}_s^{(\ell)} \mid \mathbf{z}_t^{(1:L)}, y)$ by training a classifier $p_\phi : \mathcal{V}^L \to \Delta^K$ on noised latents $\mathbf{z}_t^{(1:L)}$ for $t \in [0, 1]$ and use this to model the first term on the right-hand side of (10) by only evaluating $p_\phi$ on sequences for which $\mathbf{z}_s^{(1:L)}$ and $\mathbf{z}_t^{(1:L)}$ differ by at most the token at position $\ell$:

$$
p(y \mid \mathbf{z}_s^{(\ell)}, \mathbf{z}_t^{(1:L)}) \approx p_\phi(y \mid \tilde{\mathbf{z}}^{(1:L)}), \quad \text{for} \quad \tilde{\mathbf{z}}^{(1:L)} = \left[ \mathbf{z}_t^{(1:\ell-1)}, \mathbf{z}_s^{(\ell)}, \mathbf{z}_t^{(\ell+1:L)} \right] \in \tilde{\mathcal{Z}}_\ell(\mathbf{z}_t^{(1:L)}).
$$

We additionally train an unconditional denoising network $p_\theta(\mathbf{z}_s^{(\ell)} \mid \mathbf{z}_t^{(1:L)})$ and then sample from the re-normalized distribution:

$$
p_{\phi,\theta}^\gamma(\mathbf{z}_s^{(1:L)} \mid \mathbf{z}_t^{(1:L)}, y) = \prod_{\ell=1}^L \frac{p_\phi(y \mid \tilde{\mathbf{z}}^{(1:L)})^\gamma p_\theta(\mathbf{z}_s^{(\ell)} \mid \mathbf{z}_t^{(1:L)})}{\sum_{\tilde{\mathbf{z}}^{(1:L)}} p_\phi(y \mid \tilde{\mathbf{z}}^{(1:L)})^\gamma p_\theta(\mathbf{z}_s^{(\ell)} \mid \mathbf{z}_t^{(1:L)})}. \tag{11}
$$

Restricting the summation in the denominators of (11) to $\tilde{\mathcal{Z}}_\ell(\mathbf{z}_t^{(1:L)})$ makes normalization tractable, as we are only summing over $N$ terms.

We refer to our method as **D-CBG** for **D**iscrete **C**lassifier-**B**ased **G**uidance. Our method can be thought of as an adaptation of the successful FUDGE (Yang & Klein, 2021) approach, which guides AR generation, to discrete diffusion, similar to how NOS (Gruver et al., 2024) extended the AR guidance mechanism of PPLM (Dathathri et al., 2019) to diffusion models.

**First Order Approximation** While tractable, this formulation suffers from the drawback that at each denoising step we must perform $\mathcal{O}(L \cdot N)$ forward passes through the classifier model, which can quickly become impractical for larger vocabularies and longer sequences. Similarly to Grathwohl et al. (2021), Vignac et al. (2022), we can treat the classifier $p_\phi$ as a continuous function of the one-hot inputs $\tilde{\mathbf{z}}^{(1:L)} \in \mathbb{R}^{L \times N}$ and use the first order Taylor approximation of $\log p_\phi$ to efficiently compute $p_\phi(y \mid \tilde{\mathbf{z}}^{(1:L)})$ with only a single forward and backward pass through the classifier model:

$$
p_\phi(y \mid \tilde{\mathbf{z}}^{(1:L)}) = \exp\left(\log \frac{p_\phi(y \mid \tilde{\mathbf{z}}^{(1:L)})}{p_\phi(y \mid \mathbf{z}_t^{(1:L)})} + \log p_\phi(y \mid \mathbf{z}_t^{(1:L)})\right)
$$

$$
\approx \exp\left((\tilde{\mathbf{z}}^{(1:L)} - \mathbf{z}_t^{(1:L)})^T \nabla_{\mathbf{z}_t^{(1:L)}} \log p_\phi(y \mid \mathbf{z}_t^{(1:L)}) + \log p_\phi(y \mid \mathbf{z}_t^{(1:L)})\right). \quad (12)
$$

## 4 UNIFORM DIFFUSION LANGUAGE MODELS

While masked diffusion models have demonstrated better language modeling compared to other discrete diffusion (Austin et al., 2021; Lou et al., 2023), we argue that they are less amenable to guidance, since once a token is unmasked at some time $t$ it remains so for all $s < t$. In contrast, for uniform noising, intermediate latents can be refined multiple times during the denoising process. We therefore revisit categorical uniform noise discrete diffusion, where $\boldsymbol{\pi} = \boldsymbol{u} := \mathbf{1}/N$. Our aim is that by analyzing this class of diffusion models more carefully, we can reduce the gap to absorbing-state and yield performant models that are more easily steered by the guidance tools we developed above.

### 4.1 UNIFORM NOISE DIFFUSION

**Discrete Time Likelihood Bound** We start with the variational lower bound that is defined by a general diffusion process. For discrete-time diffusion, i.e., some finite steps $T$, we define $t(i) = (i+1)/T$ and $s(i) = i/T$ for $i$ in $0, \ldots, T$. Denoting the Kullback-Leibler divergence as $D_{\mathrm{KL}}$, the denoising network $p_\theta$ is trained to minimize a variational upper bound (NELBO), which is given by:

$$
\mathbb{E}_q\left[\underbrace{-\log p_\theta(\mathbf{x}|\mathbf{z}_{t(0)})}_{\mathcal{L}_{\mathrm{recons}}} + \underbrace{\sum_{i=1}^{T} D_{\mathrm{KL}}[q(\mathbf{z}_{s(i)}|\mathbf{z}_{t(i)}, \mathbf{x})\|p_\theta(\mathbf{z}_{s(i)}|\mathbf{z}_{t(i)})]}_{\mathcal{L}_{\mathrm{diffusion}}}\right] + \underbrace{D_{\mathrm{KL}}[q(\mathbf{z}_{t(T)}|\mathbf{x})\|p_\theta(\mathbf{z}_{t(T)})]}_{\mathcal{L}_{\mathrm{prior}}},
$$

$$(13)$$

When clear, we drop the explicit dependence of $t$ and $s$ on the discrete step $i$.

**Uniform Noise Forward Process** We focus on uniform noise diffusion using the interpolating discrete diffusion framework (Sahoo et al., 2024a; Zhao et al., 2024; Zheng et al., 2023). When letting $\boldsymbol{\pi} = \boldsymbol{u}$, the input $\mathbf{x}$ transitions to a random state with some probability at each time step. Crucially, after $\mathbf{x}$ changes once, it can do so again. When $\boldsymbol{\pi} = \boldsymbol{u}$, the posterior from (2) becomes

$$
q(\mathbf{z}_s \mid \mathbf{z}_t, \mathbf{x}) = \mathrm{Cat}\left(\mathbf{z}_s; \frac{N\alpha_t \mathbf{z}_t \odot \mathbf{x} + (\alpha_{t|s} - \alpha_t)\mathbf{z}_t + (\alpha_s - \alpha_t)\mathbf{x} + \frac{(\alpha_s - \alpha_t)(1 - \alpha_s)}{N\alpha_s}\mathbf{1}}{N\alpha_t \langle \mathbf{z}_t, \mathbf{x}\rangle + 1 - \alpha_t}\right) \quad (14)
$$

**Denoising Process** The optimal form for the reverse diffusion process $p_\theta$ matches (14): in fact setting $p_\theta$ to (14) reduces the KL terms in (13) to zero. However, setting $p_\theta$ to exactly (14) is not possible because it cannot be a function $\mathbf{x}$ (which $p_\theta$ is generating). Therefore, we introduce a predictive model $\mathbf{x}_\theta(\mathbf{z}_t, t) : \mathcal{V} \times [0, 1] \to \Delta^N$ of the 'clean' data given a noisy latent $\mathbf{z}_t$ at time $t$. We use $\mathbf{x}_\theta$ to parameterize the denoising process as $p_\theta(\mathbf{z}_s \mid \mathbf{z}_t) = q(\mathbf{z}_s \mid \mathbf{z}_t, \mathbf{x} = \mathbf{x}_\theta)$, yielding:

$$
p_\theta(\mathbf{z}_s \mid \mathbf{z}_t) = \mathrm{Cat}\left(\mathbf{z}_s; \frac{N\alpha_t \mathbf{z}_t \odot \mathbf{x}_\theta + (\alpha_{t|s} - \alpha_t)\mathbf{z}_t + (\alpha_s - \alpha_t)\mathbf{x}_\theta + \frac{(\alpha_s - \alpha_t)(1 - \alpha_s)}{N\alpha_s}\mathbf{1}}{N\alpha_t \langle \mathbf{z}_t, \mathbf{x}_\theta\rangle + 1 - \alpha_t}\right), \quad (15)
$$

Note that this minimizes the $\mathcal{L}_{\mathrm{diffusion}}$ term in (13) precisely when $\mathbf{x}_\theta = \mathbf{x}$, as desired. To simplify notation, we omit below the explicit dependence of $\mathbf{x}_\theta$ on $t$.

## 4.2 IMPROVED LIKELIHOOD BOUNDS IN CONTINUOUS TIME

We now present our contribution to uniform noise discrete diffusion modeling. We leverage the above formulation to develop an improved NELBO by taking $T \to \infty$ and analyzing each term $\mathcal{L}_{\text{recons}}, \mathcal{L}_{\text{diffusion}}, \mathcal{L}_{\text{prior}}$ in (13). This yields three improvements: (1) a simple and elegant closed-form expression for the NELBO that is easier to reason about; (2) an analytical reduction of $\mathcal{L}_{\text{recons}}, \mathcal{L}_{\text{prior}}$ to zero, which tightens the NELBO; (3) a further tightening via the continuous-time extension of $\mathcal{L}_{\text{diffusion}}$, as in Kingma et al. (2021) and Sahoo et al. (2024a;b).

**Prior Loss ($\mathcal{L}_{\textbf{prior}}$)** Given that we define our corruption process as an interpolation between clean data and a limiting distribution, for any noise schedules where $\alpha_{t(T)} = 0$, the distribution $q(\mathbf{z}_{t(T)}) = \boldsymbol{\pi}$. Therefore, we can simply define $p_\theta(\mathbf{z}_{t(T)}) = \boldsymbol{\pi}$, and the KL divergence in $\mathcal{L}_{\text{prior}}$ evaluates to zero.

**Reconstruction Loss ($\mathcal{L}_{\textbf{recons}}$)** For $\mathcal{L}_{\text{recons}}$, if our noise schedule is such that $T \to \infty \implies \alpha_{t(0)} = 1$ (i.e., $t(0) = 1/T$), then the marginal $q(\mathbf{z}_{\frac{1}{T}} \mid \mathbf{x}) \to \text{Cat}(\mathbf{z}_{\frac{1}{T}}; \mathbf{x})$. That is, in the limit, the first latent vector is identically equal to the clean data. We can thus parameterize our denoising network such that at time $t(0)$ the function simply copies its inputs: $\mathbf{x}_\theta(\mathbf{z}_{\frac{1}{T}}, 1/T) = \mathbf{z}_{\frac{1}{T}}$. Additionally, we note that our choice of parameterization for $p_\theta$ implies that $p_\theta(\mathbf{x} \mid \mathbf{z}_{\frac{1}{T}}) = \mathbf{x}_\theta(\mathbf{z}_{\frac{1}{T}}, 1/T)$. Thus in the continuous time limit, we have:

$$\lim_{T\to\infty} \mathbb{E}_q[\mathcal{L}_{\text{recons}}] = \lim_{T\to\infty} \mathbb{E}_q[\log p_\theta(\mathbf{x} \mid \mathbf{z}_{\frac{1}{T}})] = \lim_{T\to\infty} \mathbb{E}_q[\log(\langle \mathbf{x}, \mathbf{x}_\theta(\mathbf{z}_{\frac{1}{T}}, 1/T)\rangle)] = 0.$$

**Diffusion Loss ($\mathcal{L}_{\textbf{diffusion}}$)** Turning finally to the diffusion loss term $\mathcal{L}_{\text{diffusion}}$, we first define the shorthand $D_{\text{KL}}[q_t||p_\theta] = D_{\text{KL}}[[q(\mathbf{z}_s \mid \mathbf{z}_t, \mathbf{x})||p_\theta(\mathbf{z}_s \mid \mathbf{z}_t)]]$ and then re-write this loss term as an expectation over $t$ uniformly sampled from $1/T, 2/T, \ldots, T$:

$$\mathcal{L}_{\text{diffusion}} = \sum_{i=1}^{T} D_{\text{KL}}[q_{i/T}||p_\theta] = T \cdot \mathbb{E}_{t\sim\{1/T, 2/T, \ldots, T\}} D_{\text{KL}}[q_t||p_\theta]. \tag{16}$$

Plugging in our expressions for the true and predicted posteriors from (14) and (15) into (16), then taking $T \to \infty$, we get (see Appendix A for details):

$$\lim_{T\to\infty} \mathcal{L}_{\text{diffusion}} = \lim_{T\to\infty} \mathbb{E}_{t\sim\{1/T, 2/T, \ldots, T\}} T \cdot D_{\text{KL}}[q_t||p_\theta] = \int_{t=0}^{t=1} \lim_{T\to\infty} T \cdot D_{\text{KL}}[q_t||p_\theta] \mathrm{d}t$$
$$= \int_{t=0}^{t=1} \left[ \frac{\alpha'_t}{N\alpha_t} \left[ \frac{N}{\bar{\mathbf{x}}_i} - \frac{N}{(\bar{\mathbf{x}}_\theta)_i} - \sum_{j \text{ s.t. } (\mathbf{z}_t)_j = 0} \left( \frac{\bar{\mathbf{x}}_j}{\bar{\mathbf{x}}_i} \right) \log\left[ \left( \frac{(\bar{\mathbf{x}}_\theta)_i \cdot \bar{\mathbf{x}}_j}{(\bar{\mathbf{x}}_\theta)_j \cdot \bar{\mathbf{x}}_i} \right) \right] \right] \right] \mathrm{d}t, \tag{17}$$

where $\mathbf{x}_j$ denotes the $j^{\text{th}}$ index of a vector $\mathbf{x}$, $\bar{\mathbf{x}} = N\alpha_t\mathbf{x} + (1-\alpha_t)\mathbf{1}$, $\bar{\mathbf{x}}_\theta = N\alpha_t\mathbf{x}_\theta + (1-\alpha_t)\mathbf{1}$, and we define $i = \arg\max_{j\in[N]}(\mathbf{z}_t)_j$ to be the non-zero entry of $\mathbf{z}_t$.

Combining our arguments regarding $\mathcal{L}_{\text{prior}}$ and $\mathcal{L}_{\text{recons}}$ with (17), yields our final tight bound:

$$\mathcal{L}^\infty = \int_{t=0}^{t=1} \mathbb{E}_q\left[ \frac{\alpha'_t}{N\alpha_t} \left[ \frac{N}{\bar{\mathbf{x}}_i} - \frac{N}{(\bar{\mathbf{x}}_\theta)_i} - \sum_{j \text{ s.t. } (\mathbf{z}_t)_j = 0} \left( \frac{\bar{\mathbf{x}}_j}{\bar{\mathbf{x}}_i} \right) \log\left[ \left( \frac{(\bar{\mathbf{x}}_\theta)_i \cdot \bar{\mathbf{x}}_j}{(\bar{\mathbf{x}}_\theta)_j \cdot \bar{\mathbf{x}}_i} \right) \right] \right] \right] \mathrm{d}t. \tag{18}$$

**Extension to Sequences** Extending training with (18) from $\mathbf{x} \in \mathcal{V}$ to sequences $\mathbf{x}^{(1:L)} \in \mathcal{V}^L$, we make the assumption that the denoising process factorizes independently across tokens when conditioned on a sequence of noisy latents $\mathbf{z}_t^{(1:L)}$. In this case, we use a single model $\mathbf{x}_\theta^{(\ell)}(\mathbf{z}_t^{(1:L)}, t)$ for predicting each token $\ell \in \{1, \ldots, L\}$ in a sequence, and we train with the sequence-level objective:

$$\mathcal{L}^\infty = \int_{t=0}^{t=1} \mathbb{E}_q \sum_\ell \left[ \frac{\alpha'_t}{N\alpha_t} \left[ \frac{N}{\bar{\mathbf{x}}_i^{(\ell)}} - \frac{N}{(\bar{\mathbf{x}}_\theta^{(\ell)})_i} - \sum_{j \text{ s.t. } (\mathbf{z}_t^{(\ell)})_j = 0} \left( \frac{\bar{\mathbf{x}}_j^{(\ell)}}{\bar{\mathbf{x}}_i^{(\ell)}} \right) \log\left[ \left( \frac{(\bar{\mathbf{x}}_\theta^{(\ell)})_i \cdot \bar{\mathbf{x}}_j^{(\ell)}}{(\bar{\mathbf{x}}_\theta^{(\ell)})_j \cdot \bar{\mathbf{x}}_i^{(\ell)}} \right) \right] \right] \right] \mathrm{d}t. \tag{19}$$

We dub models trained with our refined objective **U**niform **D**iffusion **L**anguage **M**odels (UDLM).

## 5 EXPERIMENTS

**Datasets** For our language modeling experiments, we examine several discrete domains: reference genomes from tens species (Species10), the QM9 small molecule dataset (Ruddigkeit et al., 2012; Ramakrishnan et al., 2014), where molecules are represented by SMILES strings (Weininger, 1988), CIFAR10 discretized images (Krizhevsky et al., 2009), and three NLP datasets consisting of text8 (Mahoney, 2011), Amazon Review (McAuley & Leskovec, 2013; Zhang et al., 2015), and the one billion words dataset (LM1B; Chelba et al. (2014)). These datasets cover a range of domains and vocabularies of varying sizes (see Table 1). For guidance experiments, we explore species-specific sequence generation, molecular property maximization, and class-conditional image generation.

### 5.1 LANGUAGE MODELING WITH UNIFORM NOISE DISCRETE DIFFUSION

Our language modeling experiments show that (1) contrary to a widely-held belief, **uniform noise diffusion can attain state-of-the-art performance** on small vocabulary datasets (Table 1), and that (2) our **UDLM are state-of-the-art among uniform noise diffusion** (Tables 2 and 3).

In Table 1, despite previous evidence indicating that absorbing-state discrete diffusion greatly outperforms uniform noise, we find a more nuanced story. Namely, for smaller vocabulary regimes, the gap between MDLM and UDLM is negligible, with UDLM even outperforming absorbing-state on certain datasets. Even within a single domain, we find a trend between vocabulary size and performance gap, with the text8 results of UDLM on par with MDLM, and a more persistent gap for larger vocabularies. Intuitively, for larger vocabulary regimes, uniform noise diffusion models need to predict over a combinatorially larger set of potential clean data sequences compared to absorbing-state; smaller vocabularies reduce this complexity.

Table 1: UDLM performs best with smaller vocabs. Perplexity (↓) on various datasets. Best values are **bolded**. [*] indicates values reported from early stopping on the validation set; otherwise validation performance at the end of training is used. [†]From Sahoo et al. (2024a). [$]From Lou et al. (2023).

|  | \|Vocab.\| | AR | MDLM | UDLM |
|---|---|---|---|---|
| Species10 | 12 | **2.88** | $3.17_{\leq}$ | $3.15_{\leq}$ |
| QM9[*] | 40 | 2.19 | $2.12_{\leq}$ | **$2.02_{\leq}$** |
| CIFAR10 | 256 | - | **$9.14_{\leq}$** | $11.21_{\leq}$ |
| text8 | 35 | **2.35**[$] | $2.62_{\leq}$ | $2.71_{\leq}$ |
| Amazon[*] | 30,522 | **21.67** | $24.93_{\leq}$ | $27.27_{\leq}$ |
| LM1B | 30,522 | **22.32**[†] | $27.04_{\leq}^{†}$ | $31.28_{\leq}$ |

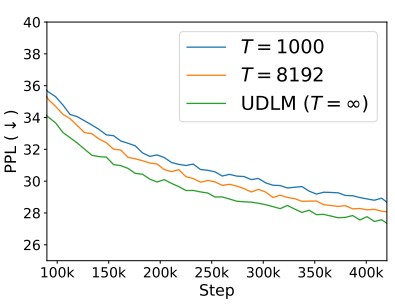

Figure 2: $T \to \infty$ improves validation PPL (↓) on Amazon dataset.

Secondly, despite the persisting gap between absorbing and uniform noise discrete diffusion for larger vocabulary NLP datasets, we note that models trained with UDLM close this gap, as evidenced in Tables 2 and 3, where we show that UDLM attains the best reported uniform noise discrete diffusion language modeling performance.

**Ablating Continuous Time Formulation** In Figure 2, we see the effect of increasing $T$ and using our continuous time NELBO. The curves represent validation perplexity on the Amazon Polarity dataset at various training steps, and we observe that increasing $T$ indeed improves language modeling with UDLM, i.e., $T = \infty$, performing best.

### 5.2 GUIDED DISCRETE DIFFUSION

Our guidance results indicate that (1) **classifier-free guidance is more useful when paired with diffusion models compared to AR** (Table 4) and that (2) **our proposed D-CBG is the best classifier-based method** for discrete guidance, especially when combined with UDLM (Table 5 & Figure 3).

**Baselines** For guidance experiments, our primary baseline is the dominant AR approach. We compare to three flavors of guided AR. The first is applying D-CFG to AR models. We also use the established control mechanisms of Plug-and-play language models (PPLM; Dathathri et al. (2019)) and FUDGE (Yang & Klein, 2021). To demonstrate the better performance of our D-CBG method, we compare to Gruver et al. (2024) (NOS), an extension of PPLM to discrete diffusion.

Table 2: UDLM outperforms other uniform discrete diffusion on text8. Best value is **bolded** & best uniform diffusion value is underlined. [†]From Lou et al. (2023). [*]From Shi et al. (2024).

| Method | BPC ($\downarrow$) |
|---|---|
| *Autoregressive* | |
| IAF/SCF[†] (Ziegler & Rush, 2019) | 1.88 |
| Argmax Flow[†] (Hoogeboom et al., 2021) | 1.39 |
| Discrete Flow[†] (Tran et al., 2019) | **1.23** |
| Autoregressive[†] | **1.23** |
| *Non-autoregressive* | |
| Mult. Diffusion[†] (Hoogeboom et al., 2021) | $1.72_{\leq}$ |
| MAC[†] (Shih et al., 2022) | $1.40_{\leq}$ |
| BFN[†] (Graves et al., 2023) | $1.41_{\leq}$ |
| D3PM Absorb[†] (Austin et al., 2021) | $1.45_{\leq}$ |
| SEDD Absorb[†] (Lou et al., 2023) | $1.39_{\leq}$ |
| MDLM (Sahoo et al., 2024a) *(Retrained)* | $1.38_{\leq}$ |
| MD4[*] (Shi et al., 2024) | $1.37_{\leq}$ |
| GenMD4[*] (Shi et al., 2024) | $1.34_{\leq}$ |
| *Discrete Uniform Diffusion* | |
| D3PM Uniform[†] (Austin et al., 2021) | $1.61_{\leq}$ |
| SEDD Uniform[†] (Lou et al., 2023) | $1.47_{\leq}$ |
| UDLM (*Ours*) | $\underline{1.44}_{\leq}$ |

Table 3: UDLM outperforms other uniform discrete diffusion on LM1B. Best value is **bolded** & best discrete uniform diffusion value is underlined. [†]From Sahoo et al. (2024a). [*]From Lou et al. (2023). [$]From Austin et al. (2021).

| Method | PPL ($\downarrow$) |
|---|---|
| *Autoregressive*[†] | |
| Transformer-X (Dai et al., 2019) | 23.5 |
| OmniNetT (Tay et al., 2021) | **21.5** |
| Transformer | 22.32 |
| *Diffusion*[†] | |
| BERT-Mouth (Wang & Cho, 2019) | $142.89_{\leq}$ |
| D3PM Absorb (Austin et al., 2021) | $77.50_{\leq}$ |
| Diffusion-LM (Li et al., 2022) | $118.62_{\leq}$ |
| DiffusionBert (Wang & Cho, 2019) | $63.78_{\leq}$ |
| SEDD Absorb (Lou et al., 2023) | $32.79_{\leq}$ |
| MDLM (Sahoo et al., 2024a) | $27.04_{\leq}$ |
| *Discrete Uniform Diffusion* | |
| D3PM Uniform[$] (Austin et al., 2021) | $137.9_{\leq}$ |
| SEDD Uniform[*] (Lou et al., 2023) | $40.25_{\leq}$ |
| UDLM (*Ours*) | $\underline{31.28}_{\leq}$ |

**Hyperparameters** For both D-CFG and D-CBG, we vary the strength of the $\gamma$ parameter. Although the original FUDGE formulation simply uses $\gamma = 1$, we also perform a search for this baseline. For PPLM and NOS, we vary the parameters of the Langevin sampling that updates models' hidden representation, i.e., number of update steps $n$, step size $\eta$, and fluency KL weight $\gamma_{kl}$ (see Dathathri et al. (2019) and Gruver et al. (2024) for more details). For all methods, we display the best performing hyperparameter configuration in the main table results, and defer the the full sweep to Appendix E.

**Species-specific Genome Generation** For genomic sequences, we evaluate D-CFG with diffusion compared to with AR. We train on sequences of 32,768 nucleotides, using base-pair level tokenization and conditionally generate 64 sequences for each class. We train a small classifier to distinguish between generated and validation set sequences and report the area under the receiver operator curve for this classifier (Disc. AUROC). Values closer to 0.5 indicate that the classifier is unable to distinguish between synthetic and true sequences (Sarkar et al., 2024). To measure the controllability, we train a separate classifier on the Species10 dataset and measure macro F1 score of this 'oracle' classifier on the generated sequences. Results of this experiment are presented in Table 4. For reference, we also provide metrics

Table 4: Diffusion decoding with D-CFG is more controllable than AR for genomic sequences. Mean $\pm$ standard deviation reported from five random seeds. Best values are **bolded**.

| Model | D-CFG $\gamma$ | Disc. AUROC ($\downarrow$) | F1 ($\uparrow$) |
|---|---|---|---|
| Random | – | 1.00 | 0.07 |
| AR | 1 | $0.53_{\pm 0.075}$ | $0.87_{\pm 0.011}$ |
| AR | 2 | $0.90_{\pm 0.051}$ | $0.81_{\pm 0.009}$ |
| AR | 3 | $0.97_{\pm 0.018}$ | $0.74_{\pm 0.022}$ |
| MDLM | 1 | $\mathbf{0.51}_{\pm \mathbf{0.059}}$ | $0.88_{\pm 0.011}$ |
| MDLM | 2 | $0.74_{\pm 0.037}$ | $0.91_{\pm 0.009}$ |
| MDLM | 3 | $0.93_{\pm 0.031}$ | $0.78_{\pm 0.007}$ |
| UDLM | 1 | $0.52_{\pm 0.044}$ | $0.91_{\pm 0.01}$ |
| UDLM | 2 | $0.61_{\pm 0.043}$ | $0.93_{\pm 0.009}$ |
| UDLM | 3 | $0.87_{\pm 0.051}$ | $\mathbf{0.94}_{\pm \mathbf{0.007}}$ |

for randomly generating sequences with nucleotide frequencies proportional to species representation in the data. We find that both MDLM and UDLM are able to better generate sequences that match the desired control parameter, with higher F1 scores relative to AR. Moreover, UDLM is able to outperform MDLM in satisfying this control. Importantly, we find that only UDLM is amenable to increasing the guidance parameter $\gamma$, where its metrics improves while AR and MDLM metrics degrade. Finally, of note, the diffusion model generation for this experiment is accomplished with far fewer function evaluations compared to AR. Whereas AR must decode each of the 32,768 tokens, because MDLM and UDLM can decode multiple tokens in parallel, we generate with $T = 512$.

Table 5: Guidance with discrete diffusion models better balances the generation of valid and novel molecules with maximizing the property of interest, drug likeness (QED) / ring count, compared to AR. Validity, novelty, and mean QED / ring count for novel sequences are measured for generated sequences from each method. Mean $\pm$ standard deviation reported from five random seeds. Best values are **bolded**.

| Method | Guidance | *QED* Valid (↑) | Novel (↑) | Mean (↑) | *Ring Count* Valid (↑) | Novel (↑) | Mean (↑) |
|---|---|---|---|---|---|---|---|
| Original Data | – | 133k | 133k | 0.47 | 133k | 133k | 1.74 |
| *Classifier-free* | | | | | | | |
| AR | D-CFG | $946.4_{\pm 9.0}$ | $79.4_{\pm 6.4}$ | $0.60_{\pm 0.00}$ | $441.4_{\pm 11.1}$ | $77.8_{\pm 2.3}$ | $4.83_{\pm 0.08}$ |
| MDLM | D-CFG | $317.4_{\pm 11.5}$ | $\mathbf{95.8}_{\pm 9.0}$ | $0.60_{\pm 0.01}$ | $90.0_{\pm 8.2}$ | $60.0_{\pm 7.5}$ | $\mathbf{5.26}_{\pm 0.15}$ |
| UDLM | D-CFG | $\mathbf{1013.6}_{\pm 2.5}$ | $64.0_{\pm 5.1}$ | $\mathbf{0.62}_{\pm 0.00}$ | $998.2_{\pm 4.5}$ | $216.2_{\pm 13.0}$ | $4.88_{\pm 0.04}$ |
| *Classifier-based* | | | | | | | |
| AR | FUDGE | $924.4_{\pm 12.1}$ | $53.0_{\pm 3.5}$ | $0.61_{\pm 0.00}$ | $281.2_{\pm 5.0}$ | $103.6_{\pm 4.8}$ | $4.89_{\pm 0.10}$ |
| AR | PPLM | $1007.2_{\pm 2.3}$ | $142.0_{\pm 14.2}$ | $0.45_{\pm 0.00}$ | $\mathbf{1009.8}_{\pm 1.9}$ | $140.6_{\pm 16.5}$ | $1.92_{\pm 0.09}$ |
| MDLM | D-CBG | $417.6_{\pm 19.7}$ | $116.6_{\pm 8.9}$ | $0.58_{\pm 0.00}$ | $113.0_{\pm 8.8}$ | $85.6_{\pm 8.8}$ | $4.75_{\pm 0.23}$ |
| MDLM | NOS | $506.0_{\pm 29.5}$ | $\mathbf{240.4}_{\pm 16.0}$ | $0.45_{\pm 0.01}$ | $247.8_{\pm 14.0}$ | $193.6_{\pm 13.1}$ | $3.51_{\pm 0.22}$ |
| UDLM | D-CBG | $994.8_{\pm 2.9}$ | $63.8_{\pm 8.1}$ | $0.61_{\pm 0.00}$ | $897.2_{\pm 11.6}$ | $\mathbf{432.0}_{\pm 19.1}$ | $4.84_{\pm 0.02}$ |
| UDLM | NOS | $547.8_{\pm 10.6}$ | $158.8_{\pm 11.0}$ | $0.47_{\pm 0.00}$ | $573.8_{\pm 13.0}$ | $244.4_{\pm 13.1}$ | $3.96_{\pm 0.07}$ |

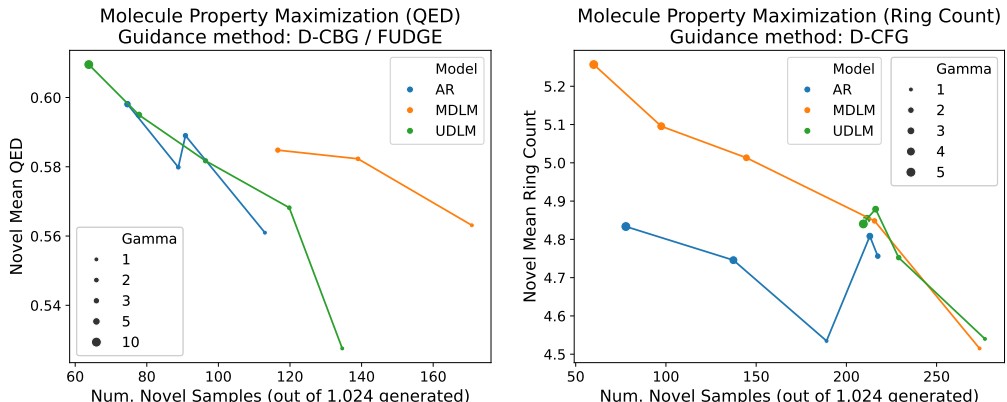

Figure 3: Diffusion models extend the steer-ability Pareto frontier. *(Left)* D-CBG outperforms FUDGE classifier guidance when maximizing drug-likeness (QED). *(Right)* D-CFG with diffusion better trades-off novel generation and ring-count maximization compared to AR.

Table 6: Guidance improves FID & IS on CIFAR10. Finite- (D3PM) vs. continuous-time (MDLM / UDLM). Best values are **bolded**. [†]From Austin et al. (2021).

| | FID (↓) | IS (↑) |
|---|---|---|
| D3PM Absorb[†] | 41.28 | 6.26 |
| MDLM | 33.75 | 6.74 |
| MDLM D-CFG | **15.56** | **9.02** |
| D3PM Uniform[†] | 51.27 | 5.99 |
| UDLM | 33.65 | 6.86 |
| UDLM D-CFG | 23.21 | 8.66 |

**Molecular Property Maximization** For QM9, we investigate novel generation of sequences that maximize either drug-likeness (QED; Bickerton et al. (2012)) or number of rings present in the molecule. We only report values for which we generated at least 50 novel sequences (out of 1,024). In Table 5, for D-CFG, we see that all methods perform comparably when maximizing drug likeness, but that MDLM and UDLM are better suited for the structural property of ring count. For classifier-based guidance mechanisms, we again find that both QED and ring count, D-CBG with discrete diffusion best trades-off maximizing these properties while generating valid and novel sequences compared to AR or diffusion with NOS.

**Class-conditional Image Generation** In Table 6, we see that both MDLM and UDLM outperform finite-time counterparts (in the form of D3PM (Austin et al., 2021)) with improved image quality

metrics of Fréchet inception distance (FID; Heusel et al. (2017)) and Inception Score (IS; Salimans et al. (2016)). This is especially true when we add guidance using D-CFG. MDLM / UDLM samples are generated with $T = 1000$.

**Ablation: Faster Sampling** In Table 7, we explore using faster inference settings (smaller $T$) where diffusion models predict multiple pixels in parallel. These results focus on the CIFAR10 dataset; see Appendix E.3 for additional experiments on other datasets. MDLM's performance deteriorates with smaller $T$, whereas UDLM is robust to this setting. This validates a key motivation behind UDLM: in settings where MDLM 'locks in' certain predictions that it cannot change, UDLM is more resilient given that all tokens can change throughout the decoding process.

# 6 RELATED WORKS, DISCUSSION, AND CONCLUSION

**Discrete Diffusion** Recent works have examined interpolating discrete diffusion (Ou et al., 2024; Sahoo et al., 2024a; Shi et al., 2024; Zhao et al., 2024; Zheng et al., 2023), a special case of the general framework from D3PM (Austin et al., 2021). Our UDLM method most closely aligns to the state-of-the-art discrete diffusion of Ou et al. (2024), Sahoo et al. (2024a), and Shi et al. (2024) that focus on absorbing state diffusion. Similar to these works, we provide a continuous time NELBO leading to performance gains, but we focus on uniform noise. Of note, other extensions of D3PM that do not start from the variational perspective, instead rely on the formalisms of continuous time Markov chains (CTMC) (Campbell et al., 2022) and concrete score matching (Lou et al., 2023), but

Table 7: UDLM is robust to faster sampling. CIFAR10 images sampled from a conditional model (D-CFG$_{\gamma=1}$). Best values per $T$ are **bolded**.

| Model | FID ($\downarrow$) | IS ($\uparrow$) |
|---|---|---|
| $T = 128$ | | |
| MDLM | 64.09 | 5.81 |
| UDLM | **30.48** | **7.30** |
| $T = 1024$ | | |
| MDLM | 27.94 | 7.14 |
| UDLM | **26.70** | **7.43** |

are less performant than works such as MDLM and our method. Also stemming from CTMC are extensions of flow-based (Lipman et al., 2022) approaches to discrete data (Campbell et al., 2024; Gat et al., 2024). In Appendix C, we discuss connections of our work to CTMC in more detail.

**Guidance** Leveraging continuous embeddings of discrete data, Diffusion-LM (Li et al., 2022) uses Langevin sampling with CBG. Similarly, SSD-LM (Han et al., 2022) perform Gaussian noising on the logits of a bi-directional model, which they combine with pre-trained classifiers to perform CBG with Langevin dynamics. LD4LG (Lovelace et al., 2024) implement CFG on continuous embeddings. Stark et al. (2024) use flow-matching on the simplex to perform CFG and CBG on continuous representations. Wang et al. (2023) perform guidance using auxiliary semantic latent variables. In contrast, to these continuous formulations for discrete data, our work adapts guidance mechanisms directly to the discrete domain.

FUDGE (Yang & Klein, 2021) can be viewed as the AR analog to our D-CBG. DiGress (Vignac et al., 2022) uses a similar first-order approximation to resolve the intractability of the normalizing constant. In our work, we derive a tractable expression for classifier-based guidance and simply use the Taylor approximation to speed up computation in large sequence length and vocabulary size regimes. Sanchez et al. (2023) derives an equivalent formulation for D-CFG and use it to better enforce AR models' adherence to prefix prompts. FreeGress (Ninniri et al., 2024) also offer a comparable method to D-CFG, but focus on graph diffusion models. Most similar to our method, is the concurrent work of Nisonoff et al. (2024), which derives classifier-based and classifier-free guidance for discrete diffusion and flow models. However, this work is highly tailored to models that leverage the formalism of CTMC, and guidance is applied to the rate matrices. See Appendix B for details on relating our guidance methods to Nisonoff et al. (2024).

**Conclusion** In search of a more controllable diffusion process, in this work, we derived a tight variational bound for uniform noise discrete diffusion, closing the gap to state-of-the-art absorbing-state diffusion models. We also highlighted that contrary to previous findings, in small vocabulary regimes, uniform noise is on par or better than absorbing state. We then demonstrated that straightforward adaptations of classifier-based and classifier-free guidance can offer improved guided generation relative to AR models. We found that with classifier-free mechanisms, diffusion models are more amenable to control without sacrificing quality of generated sequences. We also demonstrated that our classifier-based method is better than previous ones for both AR and diffusion models.

## ACKNOWLEDGMENTS AND DISCLOSURE OF FUNDING

We thank Nate Gruver for useful discussions regarding NOS. This work was supported by NSF CAREER grants (#2145577 and #2037519) and an NIH MIRA grant (#1R35GM151243-01).

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

CONTENTS

## A  CONTINUOUS TIME DISCRETE UNIFORM DIFFUSION

Here, we derive a continuous time formulation ($T \to \infty$) for the diffusion loss term $\mathcal{L}_{\text{diffusion}}$ when using the uniform distribution as the limiting distribution of the diffusion process.

For uniform noise diffusion, we define a limiting distribution $\boldsymbol{u} = \mathbf{1}/N$, where $\mathbf{1}$ represents the column vector of all ones, and $N$ the size of the vocabulary. We adopt the assumption from MDLM (Sahoo et al., 2024a) that our diffusion process interpolates between clean data and noise:

$$q(\mathbf{z}_t \mid \mathbf{x}) = \text{Cat}(\mathbf{z}_t; \alpha_t \mathbf{x} + (1 - \alpha_t)\boldsymbol{\pi}) \tag{20}$$

and the marginals are given as

$$q(\mathbf{z}_t \mid \mathbf{z}_s) = \text{Cat}(\mathbf{z}_t; \alpha_{t|s}\mathbf{z}_s + (1 - \alpha_{t|s})\boldsymbol{\pi}), \tag{21}$$

where $\alpha_{t|s} = \alpha_t/\alpha_s$.

Following Austin et al. (2021), we can derive the posterior:

$$q(\mathbf{z}_s \mid \mathbf{z}_t, \mathbf{x}) = \text{Cat}\left(\mathbf{z}_s; \frac{[\alpha_{t|s}\mathbf{z}_t + (1 - \alpha_{t|s})\mathbf{1}\boldsymbol{\pi}^\top \mathbf{z}_t] \odot [\alpha_s \mathbf{x} + (1 - \alpha_s)\boldsymbol{\pi}]}{\alpha_t \langle \mathbf{z}_t, \mathbf{x} \rangle + (1 - \alpha_t)\mathbf{z}_t^\top \boldsymbol{\pi}}\right). \tag{22}$$

Using $\boldsymbol{\pi} = \boldsymbol{u}$ as defined above, we get the following:

$$q(\mathbf{z}_s \mid \mathbf{z}_t, \mathbf{x}) = \text{Cat}\left(\mathbf{z}_s; \frac{\alpha_t \mathbf{z}_t \odot \mathbf{x} + \frac{(\alpha_{t|s} - \alpha_t)}{N}\mathbf{z}_t + \frac{(\alpha_s - \alpha_t)}{N}\mathbf{x} + \frac{(1 - \alpha_{t|s})(1 - \alpha_s)}{N^2}\mathbf{1}}{\alpha_t \langle \mathbf{z}_t, \mathbf{x} \rangle + \frac{(1 - \alpha_t)}{N}}\right) \tag{23}$$

$$= \text{Cat}\left(\mathbf{z}_s; \frac{N\alpha_t \mathbf{z}_t \odot \mathbf{x} + (\alpha_{t|s} - \alpha_t)\mathbf{z}_t + (\alpha_s - \alpha_t)\mathbf{x} + \frac{(\alpha_s - \alpha_t)(1 - \alpha_s)}{N\alpha_s}\mathbf{1}}{N\alpha_t \langle \mathbf{z}_t, \mathbf{x} \rangle + 1 - \alpha_t}\right). \tag{24}$$

For the denoising distribution, we replace $\mathbf{x}$ by $\mathbf{x}_\theta$:

$$p_\theta(\mathbf{z}_s \mid \mathbf{z}_t) = \text{Cat}\left(\mathbf{z}_s; \frac{N\alpha_t \mathbf{z}_t \odot \mathbf{x}_\theta + (\alpha_{t|s} - \alpha_t)\mathbf{z}_t + (\alpha_s - \alpha_t)\mathbf{x}_\theta + \frac{(\alpha_s - \alpha_t)(1 - \alpha_s)}{\alpha_s N}\mathbf{1}}{N\alpha_t \langle \mathbf{z}_t, \mathbf{x}_\theta \rangle + 1 - \alpha_t}\right). \tag{25}$$

Let us now look at the diffusion loss term in the NELBO:

$$T \cdot D_{\text{KL}}(q(\mathbf{z}_s \mid \mathbf{z}_t, \mathbf{x}) \| p_\theta(\mathbf{z}_s \mid \mathbf{z}_t)) = T \cdot \sum_{j \in [N]} q(\mathbf{z}_s \mid \mathbf{z}_t, \mathbf{x})_j \log\left(\frac{q(\mathbf{z}_s \mid \mathbf{z}_t, \mathbf{x})_j}{p_\theta(\mathbf{z}_s \mid \mathbf{z}_t)_j}\right). \tag{26}$$

Letting $i = \arg\max_{j \in [N]}(\mathbf{z}_t)_j$ be the non-zero entry of $\mathbf{z}_t$, we can break up this KL into two terms:

$$T \cdot D_{\text{KL}}(q(\mathbf{z}_s \mid \mathbf{z}_t, \mathbf{x}) \| p_\theta(\mathbf{z}_s \mid \mathbf{z}_t)) = \underbrace{T \cdot q(\mathbf{z}_s \mid \mathbf{z}_t, \mathbf{x})_i \log\left(\frac{q(\mathbf{z}_s \mid \mathbf{z}_t, \mathbf{x})_i}{p_\theta(\mathbf{z}_s \mid \mathbf{z}_t)_i}\right)}_{\text{Term 1}}$$

$$+ \underbrace{T \cdot \sum_{\substack{j \in [N] \\ \text{s.t. } (\mathbf{z}_t)_j = 0}} q(\mathbf{z}_s \mid \mathbf{z}_t, \mathbf{x})_j \log\left(\frac{q(\mathbf{z}_s \mid \mathbf{z}_t, \mathbf{x})_j}{p_\theta(\mathbf{z}_s \mid \mathbf{z}_t)_j}\right)}_{\text{Term 2}}. \tag{27}$$

We now examine each of these terms taking $T \to \infty$, or equivalently, using $s = t - \frac{1}{T} \implies T = \frac{1}{t-s}$, taking $s \to t$.

**Term 1:**

$$\lim_{s \to t} \frac{1}{t-s} \cdot \frac{N\alpha_t \mathbf{x}_i + \alpha_{t|s} - \alpha_t + (\alpha_s - \alpha_t)\mathbf{x}_i + \frac{(\alpha_s - \alpha_t)(1-\alpha_s)}{\alpha_s N}}{N\alpha_t \mathbf{x}_i + 1 - \alpha_t}$$

$$\cdot \log\left[\frac{\frac{N\alpha_t \mathbf{x}_i + \alpha_{t|s} - \alpha_t + (\alpha_s - \alpha_t)\mathbf{x}_i + \frac{(\alpha_s - \alpha_t)(1-\alpha_s)}{\alpha_s N}}{N\alpha_t \mathbf{x}_i + 1 - \alpha_t}}{\frac{N\alpha_t(\mathbf{x}_\theta)_i + \alpha_{t|s} - \alpha_t + (\alpha_s - \alpha_t)(\mathbf{x}_\theta)_i + \frac{(\alpha_s - \alpha_t)(1-\alpha_s)}{\alpha_s N}}{N\alpha_t(\mathbf{x}_\theta)_i + 1 - \alpha_t}}\right] \tag{28}$$

As $s \to t$, the coefficient on the $\log$ term will approach $1$. Additionally both the numerator and the denominator inside the $\log$ term will approach $1$, thus the entire $\log$ term will approach $0$. Combining this with the fact that $t - s$ will approach $0$, gives us the indeterminate form of $0/0$, hence we apply L'Hôpital's's rule to (28). Writing $q$ and $p_\theta$ as functions of $s$, when we differentiate the $\log$ term we get:

$$\frac{d}{ds} \log\left[\frac{q(s)}{p_\theta(s)}\right] = \frac{d}{ds} \log q(s) - \frac{d}{ds} \log p_\theta(s)$$

$$= \frac{\frac{d}{ds} q(s)}{q(s)} - \frac{\frac{d}{ds} p_\theta(s)}{p_\theta(s)} \tag{29}$$

Let's look at each derivative term in (29):

$$\lim_{s \to t} \frac{d}{ds} q(s) = \lim_{s \to t} \frac{\frac{-\alpha'_s \alpha_t}{\alpha_s^2} + \alpha'_s(\mathbf{x}_\theta)_i + \frac{N\alpha_s[\alpha'_s(1-\alpha_s) - \alpha'_s(\alpha_s - \alpha_t)] - [N\alpha'_s(\alpha_s - \alpha_t)(1-\alpha_s)]}{N\alpha_s^2}}{N\alpha_t \mathbf{x}_i + 1 - \alpha_t}$$

$$= \frac{\frac{-\alpha'_t}{\alpha_t} + \alpha'_t \mathbf{x}_i + \frac{\alpha'_t(1-\alpha_t)}{N\alpha_t}}{N\alpha_t \mathbf{x}_i + 1 - \alpha_t}$$

$$= \frac{\alpha'_t}{N\alpha_t}\left[\frac{-N + N\alpha_t \mathbf{x}_i + 1 - \alpha_t}{N\alpha_t \mathbf{x}_i + 1 - \alpha_t}\right]$$

$$= \frac{\alpha'_t}{N\alpha_t}\left[1 - \frac{N}{N\alpha_t \mathbf{x}_i + 1 - \alpha_t}\right] \tag{30}$$

Similarly,

$$\lim_{s \to t} \frac{d}{ds} p_\theta(s) = \frac{\alpha'_t}{N\alpha_t}\left[1 - \frac{N}{N\alpha_t(\mathbf{x}_\theta)_i + 1 - \alpha_t}\right] \tag{31}$$

Note that when taking $s \to t$, both $q(s)$ and $p_\theta(s)$ evaluate to $1$. Additionally, differentiating $\frac{1}{t-s}$ with respect to $s$ evaluates to $-1$. When combining these facts and plugging (30) and (31) into (29), (28) becomes

$$\lim_{s \to t} \text{Term 1} = \frac{-\alpha'_t}{N\alpha_t}\left[1 - \frac{N}{N\alpha_t \mathbf{x}_i + 1 - \alpha_t}\right] + \frac{\alpha'_t}{N\alpha_t}\left[1 - \frac{N}{N\alpha_t(\mathbf{x}_\theta)_i + 1 - \alpha_t}\right]$$

$$= \boxed{\frac{\alpha'_t}{N\alpha_t}\left[\frac{N}{N\alpha_t \mathbf{x}_i + 1 - \alpha_t} - \frac{N}{N\alpha_t(\mathbf{x}_\theta)_i + 1 - \alpha_t}\right]} \tag{32}$$

**Term 2:**

$$\lim_{s \to t} \frac{1}{t-s} \cdot \sum_{\substack{j \in [N] \\ \text{s.t. } (\mathbf{z}_t)_j = 0}} \frac{(\alpha_s - \alpha_t)\mathbf{x}_j + \frac{(\alpha_s - \alpha_t)(1-\alpha_s)}{N\alpha_s}}{N\alpha_t \mathbf{x}_i + 1 - \alpha_t} \log\left[\frac{\frac{(\alpha_s - \alpha_t)\mathbf{x}_j + \frac{(\alpha_s - \alpha_t)(1-\alpha_s)}{N\alpha_s}}{N\alpha_t \mathbf{x}_i + 1 - \alpha_t}}{\frac{(\alpha_s - \alpha_t)(\mathbf{x}_\theta)_j + \frac{(\alpha_s - \alpha_t)(1-\alpha_t)}{N\alpha_s}}{N\alpha_t(\mathbf{x}_\theta)_i + 1 - \alpha_t}}\right] \tag{33}$$

For each term in the summation, $\frac{1}{t-s}$ times the coefficient on the $\log$ term will have an indeterminate form of $0/0$ as $s \to t$. We therefore apply L'Hôpital's rule to this coefficient:

$$\lim_{s \to t} \frac{1}{t-s} \cdot \frac{(\alpha_s - \alpha_t)\mathbf{x}_j + \frac{(\alpha_s - \alpha_t)(1-\alpha_s)}{N\alpha_s}}{N\alpha_t \mathbf{x}_i + 1 - \alpha_t} = \frac{-\alpha'_t}{N\alpha_t}\left[\frac{N\alpha_t \mathbf{x}_j + 1 - \alpha_t}{N\alpha_t \mathbf{x}_i + 1 - \alpha_t}\right] \tag{34}$$

Now, for the $\log$ term, we exchange the limit with the continuous $\log$ function and have that both the numerator and the denominator go to zero as $s \to t$. We therefore apply L'Hôpital's rule here as well:

$$\log \lim_{s \to t} \left[ \frac{\frac{(\alpha_s - \alpha_t)\mathbf{x}_j + \frac{(\alpha_s - \alpha_t)(1 - \alpha_s)}{N\alpha_s}}{N\alpha_t \mathbf{x}_i + 1 - \alpha_t}}{\frac{(\alpha_s - \alpha_t)(\mathbf{x}_\theta)_j + \frac{(\alpha_s - \alpha_t)(1 - \alpha_t)}{N\alpha_s}}{N\alpha_t(\mathbf{x}_\theta)_i + 1 - \alpha_t}} \right] = \log \lim_{s \to t} \left[ \frac{\frac{d}{ds} \frac{(\alpha_s - \alpha_t)\mathbf{x}_j + \frac{(\alpha_s - \alpha_t)(1 - \alpha_s)}{N\alpha_s}}{N\alpha_t \mathbf{x}_i + 1 - \alpha_t}}{\frac{d}{ds} \frac{(\alpha_s - \alpha_t)(\mathbf{x}_\theta)_j + \frac{(\alpha_s - \alpha_t)(1 - \alpha_t)}{N\alpha_s}}{N\alpha_t(\mathbf{x}_\theta)_i + 1 - \alpha_t}} \right]$$

$$= \log \left[ \left( \frac{N\alpha_t(\mathbf{x}_\theta)_i + 1 - \alpha_t}{N\alpha_t \mathbf{x}_i + 1 - \alpha_t} \right) \left( \frac{\alpha_t' \mathbf{x}_j + \frac{\alpha_t'(1 - \alpha_t)}{N\alpha_t}}{\alpha_t'(\mathbf{x}_\theta)_j + \frac{\alpha_t'(1 - \alpha_t)}{N\alpha_t}} \right) \right]$$

$$= \log \left[ \left( \frac{N\alpha_t(\mathbf{x}_\theta)_i + 1 - \alpha_t}{N\alpha_t \mathbf{x}_i + 1 - \alpha_t} \right) \left( \frac{\frac{\alpha_t'}{N\alpha_t}(N\alpha_t \mathbf{x}_j + 1 - \alpha_t)}{\frac{\alpha_t'}{N\alpha_t}(N\alpha_t(\mathbf{x}_\theta)_j + 1 - \alpha_t)} \right) \right]$$

$$= \log \left[ \left( \frac{N\alpha_t(\mathbf{x}_\theta)_i + 1 - \alpha_t}{N\alpha_t \mathbf{x}_i + 1 - \alpha_t} \right) \left( \frac{N\alpha_t \mathbf{x}_j + 1 - \alpha_t}{N\alpha_t(\mathbf{x}_\theta)_j + 1 - \alpha_t} \right) \right]$$

$$= \log \left[ \left( \frac{N\alpha_t(\mathbf{x}_\theta)_i + 1 - \alpha_t}{N\alpha_t(\mathbf{x}_\theta)j + 1 - \alpha_t} \right) \left( \frac{N\alpha_t \mathbf{x}_j + 1 - \alpha_t}{N\alpha_t \mathbf{x}_i + 1 - \alpha_t} \right) \right] \tag{35}$$

Multiplying (34) by (35), we get:

$$\lim_{s \to t} \text{Term 2} = \boxed{ \frac{-\alpha_t'}{N\alpha_t} \sum_{\substack{j \in [N] \\ \text{s.t. } (\mathbf{z}_t)_j = 0}} \left( \frac{N\alpha_t \mathbf{x}_j + 1 - \alpha_t}{N\alpha_t \mathbf{x}_i + 1 - \alpha_t} \right) \log \left[ \left( \frac{N\alpha_t(\mathbf{x}_\theta)_i + 1 - \alpha_t}{N\alpha_t(\mathbf{x}_\theta)j + 1 - \alpha_t} \right) \left( \frac{N\alpha_t \mathbf{x}_j + 1 - \alpha_t}{N\alpha_t \mathbf{x}_i + 1 - \alpha_t} \right) \right] }$$

$$\tag{36}$$

**Combining Terms 1 and 2:** Using (32) and (36), the final KL term in the continuous time limit is:

$$\lim_{T \to \infty} T \cdot D_{\text{KL}}(q || p_\theta) = \frac{\alpha_t'}{N\alpha_t} \left[ \frac{N}{N\alpha_t \mathbf{x}_i + 1 - \alpha_t} - \frac{N}{N\alpha_t(\mathbf{x}_\theta)_i + 1 - \alpha_t} \right.$$

$$\left. - \sum_{\substack{j \in [N] \\ \text{s.t. } (\mathbf{z}_t)_j = 0}} \left( \frac{N\alpha_t \mathbf{x}_j + 1 - \alpha_t}{N\alpha_t \mathbf{x}_i + 1 - \alpha_t} \right) \log \left[ \left( \frac{N\alpha_t(\mathbf{x}_\theta)_i + 1 - \alpha_t}{N\alpha_t(\mathbf{x}_\theta)j + 1 - \alpha_t} \right) \left( \frac{N\alpha_t \mathbf{x}_j + 1 - \alpha_t}{N\alpha_t \mathbf{x}_i + 1 - \alpha_t} \right) \right] \right].$$

Defining $\bar{\mathbf{x}} = N\alpha_t \mathbf{x} + (1 - \alpha_t)\mathbf{1}$ and $\bar{\mathbf{x}}_\theta = N\alpha_t \mathbf{x}_\theta + (1 - \alpha_t)\mathbf{1}$, as in Section 4.2, yields the desired result.

## B  RELATING GUIDANCE TO CTMC

Below we demonstrate how our proposed guidance mechanisms from Section 3 admit straightforward continuous-time extensions. In concurrent work, Nisonoff et al. (2024) use the formalisms of continuous time Markov chains (CTMC) to derive similar guidance formulas, which we show are equivalent to the continuous-time extensions of our work.

**Background on CTMC**   In the CTMC formulation, the key quantity of interest is the **rate matrix** $R_t \in \mathbb{R}^{N \times N}$. In analogy to the discrete time transition matrices $Q_t$, which define the transition probabilities between states, these rate matrices define the instantaneous rate of change between states in continuous time. More formally, letting $\delta(\mathbf{x}, \mathbf{y})$ be the Kronecker delta function that equals 1 if $\mathbf{x} = \mathbf{y}$ and 0 otherwise, and using $\mathbf{z}$ and $\mathbf{z}'$ to denote observed values of the latents, we can define the forward noising process using $R_t$ as follows:

$$q(\mathbf{z}_t = \mathbf{z}' \mid \mathbf{z}_s = \mathbf{z}) = \delta_{\mathbf{z}, \mathbf{z}'} + R_t(\mathbf{z}, \mathbf{z}') \frac{1}{T} + o\left( \frac{1}{T} \right) \tag{37}$$

where $o(1/T)$ indicates terms that vanish more quickly than $1/T$. Note that when clear from the context, we use the shorthand $\delta_{\mathbf{z}, \mathbf{z}'} := \delta(\mathbf{z}, \mathbf{z}')$. Recall that we defined $s = t - \frac{1}{T}$. In continuous

time, as $T \to \infty$, $o(1/T)$ terms are ignored, and we have

$$R_t(\mathbf{z}, \mathbf{z}') = \lim_{T \to \infty} \frac{q(\mathbf{z}_t = \mathbf{z}' \mid \mathbf{z}_{t-(1/T)} = \mathbf{z}) - \delta_{\mathbf{z}, \mathbf{z}'}}{1/T}, \tag{38}$$

hence our treatment of $R_t$ as the 'instantaneous' rate of state transitions.

Importantly, processes defined as in (37) have known time reversals given by (Kelly, 2011; Sun et al., 2022):

$$q(\mathbf{z}_s = \mathbf{z} \mid \mathbf{z}_t = \mathbf{z}') = \delta_{\mathbf{z}', \mathbf{z}} + \check{R}_t(\mathbf{z}', \mathbf{z}) \frac{1}{T} + o\left(\frac{1}{T}\right), \tag{39}$$

where $\check{R}_t$ represents the reverse rate matrix which is related to the forward matrix as follows:

$$\check{R}_t(\mathbf{z}', \mathbf{z}) = \frac{q(\mathbf{z}_t = \mathbf{z}')}{q(\mathbf{z}_t = \mathbf{z})} R_t(\mathbf{z}, \mathbf{z}'), \quad \text{if } \mathbf{z}' \neq \mathbf{z}, \tag{40}$$

with $\check{R}_t(\mathbf{z}, \mathbf{z}) = -\sum_{\mathbf{z}' \neq \mathbf{z}} \check{R}_t(\mathbf{z}', \mathbf{z})$, which ensures that the rows of $\check{R}_t$ sum to zero (i.e., mass cannot be created or destroyed). Note that in (40), we are scaling the forward rate matrix by a ratio of the unconditional marginals: $q(\mathbf{z}_t = \mathbf{z}')/q(\mathbf{z}_t = \mathbf{z})$.

We remark that in the derivations for guidance in Nisonoff et al. (2024), they slightly change the form of (39) to:

$$q(\mathbf{z}_s = \mathbf{z} \mid \mathbf{z}_t = \mathbf{z}') = \delta_{\mathbf{z}', \mathbf{z}}(1 + \check{R}_t(\mathbf{z}', \mathbf{z}'))(1/T) + (1 - \delta_{\mathbf{z}', \mathbf{z}})\check{R}_t(\mathbf{z}', \mathbf{z})(1/T) + o(1/T). \tag{41}$$

**Sampling with Learned Reverse Rate Matrices**   Assuming we have a learned model $R_{t,\theta}(\mathbf{z}', \mathbf{z})$ that approximates the true reverse rates (see Campbell et al. (2022) or Campbell et al. (2024) for details), we can generate from this model by drawing samples from the limiting distribution $\mathbf{z}_1 \sim \boldsymbol{\pi}$ and using a forward Euler discretization to sample a chain of latents that end in samples $\mathbf{x} \sim q(\mathbf{x})$ (Campbell et al., 2024):

$$\mathbf{z}_s \sim \text{Cat}(\mathbf{z}_s; \delta(\mathbf{z}_t = \mathbf{z}', \mathbf{z}_s) + (1/T) \cdot R_{t,\theta}(\mathbf{z}_t = \mathbf{z}', \mathbf{z}_s)), \tag{42}$$

where $1/T$ represents the discretization step size and, as above, $s = t - 1/T$. Additionally, note that here we follow the convention of diffusion models and let the time variable go from $t = 1 \to 0$ to indicate moving from noise towards signal, in contrast with the flow matching literature, where the reverse time convention is used (Lipman et al., 2022; Campbell et al., 2024).

## B.1   CTMC FORMULATION OF D-CFG

In Nisonoff et al. (2024), guidance is achieved by scaling the parametrized reverse rate matrix $R_{t,\theta}$. Predictor-free guidance (the analogous method to D-CFG) is presented as:

$$R_{t,\theta}^{\gamma}(\mathbf{z}_t = \mathbf{z}', \mathbf{z}_s \mid y) = R_{t,\theta}(\mathbf{z}_t = \mathbf{z}', \mathbf{z}_s \mid y)^{\gamma} \cdot R_{t,\theta}(\mathbf{z}_t = \mathbf{z}', \mathbf{z}_s)^{(1-\gamma)}, \tag{43}$$

where $R_{t,\theta}(\cdot \mid y)$ is the learned reverse rate when conditioned on $y$.

We now present the continuous-time extension of D-CFG and show that it is proportional to using the predictor-free guided reverse rate from (43) in conjunction with the forward Euler sampling from (42). Starting from the formulation of D-CFG from our work (where we make more explicit which variables are free / set)

$$p^{\gamma}(\mathbf{z}_s \mid \mathbf{z}_t = \mathbf{z}', y) \propto p_{\theta}(\mathbf{z}_s \mid \mathbf{z}_t = \mathbf{z}', y)^{\gamma} \cdot p_{\theta}(\mathbf{z}_s \mid \mathbf{z}_t = \mathbf{z}')^{(1-\gamma)},$$

when using the forward Euler sampling scheme, we can plug in the formulation from (41) (with the true reverse rate replaced with the approximate one) into the right-hand side of this equation to get

$$p^\gamma(\mathbf{z}_s \mid \mathbf{z}_t = \mathbf{z}', y) \propto$$

$$\left[ \Big( \delta(\mathbf{z}_t = \mathbf{z}', \mathbf{z}_s)(1 + R_{t,\theta}(\mathbf{z}_t = \mathbf{z}', \mathbf{z}_s = \mathbf{z}' \mid y))(1/T) + (1 - \delta(\mathbf{z}_t = \mathbf{z}', \mathbf{z}_s))R_{t,\theta}(\mathbf{z}_t = \mathbf{z}', \mathbf{z}_s \mid y)(1/T) \Big)^\gamma \right.$$

$$\left. + \Big( \delta(\mathbf{z}_t = \mathbf{z}', \mathbf{z}_s)(1 + R_{t,\theta}(\mathbf{z}_t = \mathbf{z}', \mathbf{z}_s = \mathbf{z}'))(1/T) + (1 - \delta(\mathbf{z}_t = \mathbf{z}', \mathbf{z}_s))R_{t,\theta}(\mathbf{z}_t = \mathbf{z}', \mathbf{z}_s)(1/T) \Big)^{(1-\gamma)} \right]$$

$$= \left[ \delta(\mathbf{z}_t = \mathbf{z}', \mathbf{z}_s)(1 + R_{t,\theta}(\mathbf{z}_t = \mathbf{z}', \mathbf{z}_s = \mathbf{z}' \mid y))^\gamma \cdot (1 + R_{t,\theta}(\mathbf{z}_t = \mathbf{z}', \mathbf{z}_s = \mathbf{z}'))^{(1-\gamma)}(1/T) \right.$$

$$\left. + (1 - \delta(\mathbf{z}_t = \mathbf{z}', \mathbf{z}_s))R_{t,\theta}(\mathbf{z}_t = \mathbf{z}', \mathbf{z}_s \mid y)^\gamma \cdot R_{t,\theta}(\mathbf{z}_t = \mathbf{z}', \mathbf{z}_s)^{(1-\gamma)}(1/T) \right]$$

$$= \left[ \delta(\mathbf{z}_t = \mathbf{z}', \mathbf{z}_s)(1 + R_{t,\theta}^\gamma(\mathbf{z}_t = \mathbf{z}', \mathbf{z}_s = \mathbf{z}' \mid y))(1/T) \right.$$

$$\left. + (1 - \delta(\mathbf{z}_t = \mathbf{z}', \mathbf{z}_s))R_{t,\theta}^\gamma(\mathbf{z}_t = \mathbf{z}', \mathbf{z}_s \mid y)(1/T) \right],$$

where the first equality comes from the fact that all cross terms involving $\delta(\mathbf{z}_t = \mathbf{z}', \mathbf{z}) \cdot (1 - \delta(\mathbf{z}_t = \mathbf{z}', \mathbf{z}))$ evaluate to zero, allowing us to aggregate terms under the exponents $\gamma$ and $1 - \gamma$, and the second equality comes from (43).

## B.2 CTMC Formulation of D-CBG

While for classifier-free guidance, we did not need to distinguish between latent variables with a single token and those representing sequences of tokens (as argued in Section 3.1), below we make this distinction explicit by including superscripts $(\ell)$ and $(1 : L)$ on variables $\mathbf{z}_s, \mathbf{z}_t$. However, for notational simplicity we omit theses subscripts on variables that denote realized values, such as $\mathbf{z}'$.

In Nisonoff et al. (2024), predictor guided rates (the analog to our D-CBG) are attained by scaling $R_{t,\theta}$ as follows:

$$R_{t,\theta}^\gamma(\mathbf{z}_t^{(1:L)} = \mathbf{z}', \mathbf{z}_s^{(1:L)} \mid y) = \left( \frac{p_\phi(y \mid \mathbf{z}_s^{(1:L)}, \mathbf{z}_t^{(1:L)} = \mathbf{z}')}{p_\phi(y \mid \mathbf{z}_s^{(1:L)} = \mathbf{z}', \mathbf{z}_t^{(1:L)} = \mathbf{z}')} \right)^\gamma R_{t,\theta}(\mathbf{z}_t^{(1:L)} = \mathbf{z}', \mathbf{z}_s^{(1:L)}), \tag{44}$$

where $p_\phi$ is an external classifier. Given the factorization assumptions of the forward and reverse processes, the reverse rate matrix in the CTMC formulation is only non-zero at entries where its arguments differ in (at most) one dimension (Campbell et al., 2022; 2024; Nisonoff et al., 2024). Thus, the classifier only needs to be evaluated on candidates for which latents differ in at most one token, which coincides with the argument / classifier we introduce in Section 3.2. The formulation in (44) can be further simplified to (see Nisonoff et al. (2024) for details):

$$R_{t,\theta}^\gamma(\mathbf{z}_t^{(1:L)} = \mathbf{z}', \mathbf{z}_s^{(1:L)} \mid y) = \frac{p_\phi(y \mid \mathbf{z}_s^{(1:L)})}{p_\phi(y \mid \mathbf{z}_s^{(1:L)} = \mathbf{z}_t^{(1:L)} = \mathbf{z}')} R_{t,\theta}(\mathbf{z}_t^{(1:L)} = \mathbf{z}', \mathbf{z}_s^{(1:L)}), \tag{45}$$

where for each $\ell$, using the notation introduced in Section 3.2, we only evaluate $p_\phi$ on $\mathbf{z}_s^{(1:L)} \in \tilde{\mathcal{Z}}_\ell(\mathbf{z}_t^{(1:L)})$.

As in Appendix B.1, we present the continuous-time extension of D-CBG and show that it is proportional to sampling using the reverse rate matrix in (45) (with Euler discretization for sampling).

Let $R_{t,\theta}^{(\ell)}$ represent the rate matrix corresponding to transitions where all dimensions are kept the same except for potentially the one corresponding to the $\ell^{\text{th}}$ token. We follow a similar derivation to that used in Appendix B.1 and start with our formulation for D-CBG:

$$p^\gamma(\mathbf{z}_s^{(\ell)} \mid \mathbf{z}_t^{(1:L)}, y) \propto p_\phi(y \mid \tilde{\mathbf{z}}^{(1:L)})^\gamma p_\theta(\mathbf{z}_s^{(\ell)} \mid \mathbf{z}_t^{(1:L)}),$$

where recall that $\tilde{\mathbf{z}}^{(1:L)} = \left[ \mathbf{z}_t^{(1:\ell-1)}, \mathbf{z}_s^{(\ell)}, \mathbf{z}_t^{(\ell+1:L)} \right]$. We then plug in the formulation for the approximate reverse rate to get:

$$p^\gamma(\mathbf{z}_s^{(\ell)} \mid \mathbf{z}_t^{(1:L)} = \mathbf{z}', y) \propto$$

$$p_\phi(y \mid \tilde{\mathbf{z}}^{(1:L)})^\gamma \cdot \left( \delta(\mathbf{z}_t^{(1:L)} = \mathbf{z}', \mathbf{z}_s^{(1:L)})(1 + R_{t,\theta}^{(\ell)}(\mathbf{z}_t^{(1:L)} = \mathbf{z}', \mathbf{z}_s^{(1:L)} = \mathbf{z}'))(1/T) \right.$$

$$\left. + (1 - \delta(\mathbf{z}_t^{(1:L)} = \mathbf{z}', \mathbf{z}_s^{(1:L)}))R_{t,\theta}^{(\ell)}(\mathbf{z}_t^{(1:L)} = \mathbf{z}', \mathbf{z}_s^{(1:L)} = \mathbf{z}')(1/T) \right)$$

$$\propto \left( \frac{p_\phi(y \mid \tilde{\mathbf{z}}^{(1:L)})}{p_\phi(y \mid \mathbf{z}_t^{(1:L)} = \mathbf{z}')} \right)^\gamma \cdot \left( \delta(\mathbf{z}_t = \mathbf{z}', \mathbf{z}_s)(1 + R_{t,\theta}^{(\ell)}(\mathbf{z}_t = \mathbf{z}', \mathbf{z}_s = \mathbf{z}'))(1/T) \right.$$

$$\left. + (1 - \delta(\mathbf{z}_t = \mathbf{z}', \mathbf{z}_s))R_{t,\theta}^{(\ell)}(\mathbf{z}_t^{(1:L)} = \mathbf{z}', \mathbf{z}_s^{(1:L)} = \mathbf{z}')(1/T) \right),$$

where the second proportionality comes from the fact that $p_\phi(y \mid \mathbf{z}_t^{(1:L)} = \mathbf{z}')$ is simply a constant. Therefore, we have shown the equivalence between our D-CBG method and the predictor guidance method introduced in Nisonoff et al. (2024).

## C  RELATING UDLM TO CTMC

Similar to Sahoo et al. (2024a), our work tackles the problem of discrete diffusion from the variational perspective and analyzes the ELBO in the continuous time limit. In contrast, other works, such as Campbell et al. (2022) and Lou et al. (2023), have extended the discrete diffusion framework proposed in Austin et al. (2021) using the formalisms of CTMC. In this section, we relate these two approaches.

**Forward Rate Matrices for Uniform Noise**   Recall from the analysis in Appendix A that we can use L'Hospital's rule to evaluate $\lim_{T\to\infty} T \cdot (1 - \alpha_{t|s}) = -\alpha_t'/\alpha_t$. Now, using (21) from above, for $\mathbf{z}' \neq \mathbf{z}$ we have

$$q(\mathbf{z}_t = \mathbf{z}' \mid \mathbf{z}_s = \mathbf{z}) = \frac{1 - \alpha_{t|s}}{N}. \tag{46}$$

Combining this with (38), we have that:

$$R_t(\mathbf{z}, \mathbf{z}') = \lim_{T\to\infty} T \cdot \frac{1 - \alpha_{t|s}}{N} = -\frac{\alpha_t'}{N\alpha_t}. \tag{47}$$

Now for $\mathbf{z}' = \mathbf{z}$, again from (21), we have

$$q(\mathbf{z}_t = \mathbf{z} \mid \mathbf{z}_s = \mathbf{z}) = \alpha_{t|s} + \frac{1 - \alpha_{t|s}}{N}, \tag{48}$$

which combines with (38) to yield:

$$R_t(\mathbf{z}, \mathbf{z}) = \lim_{T\to\infty} T \cdot \left( \alpha_{t|s} + \frac{1 - \alpha_{t|s}}{N} - 1 \right) = \frac{1 - N}{N} \lim_{T\to\infty} T \cdot (1 - \alpha_{t|s}) = \frac{-\alpha_t'}{N\alpha_t}(1 - N). \tag{49}$$

Alternatively, for (49), we could have relied on the property that $R_t(\mathbf{z}, \mathbf{z}) = -\sum_{\mathbf{z}' \neq \mathbf{z}} R_t(\mathbf{z}, \mathbf{z}')$.

Writing (47) and (49) as a single expression, gives:

$$R_t(\mathbf{z}, \mathbf{z}') = -\frac{\alpha_t'}{N\alpha_t}[\mathbf{1}\mathbf{1}^\top - N\mathbf{I}]. \tag{50}$$

### C.1 EQUIVALENCE OF UDLM ELBO AND SEDD ELBO

Below, we demonstrate that the variational lower bound from (18) used to train UDLM is equivalent to the lower bound derived in SEDD (Lou et al., 2023), for a specific parameterization of the denoising score matching.

**Notation**  To facilitate the discussion, we introduce a notational shorthand $q_t(\mathbf{z}') = q(\mathbf{z}_t = \mathbf{z}')$ and $q_t(\mathbf{z}' \mid \mathbf{x}) = q(\mathbf{z}_t = \mathbf{z}' \mid \mathbf{x})$.

In SEDD, the quantity of interest is the ratio of probabilities in the reverse rate matrix equation given in (40), and they train a parametric model to learn this so-called concrete score (Meng et al., 2022):

$$s_\theta(\mathbf{z})_{\mathbf{z}'} \approx \frac{q_t(\mathbf{z}')}{q_t(\mathbf{z})}. \tag{51}$$

Since the unconditional marginals in this ratio are intractable, SEDD proposes a tractable denoising score-based objective. Importantly, they show that the denoising score-based objective they use for training serves as variational bound and derive the following expression for Negative Evidence Lower Bound (NELBO):

NELBO$_{\text{SEDD}}$

$$= \mathbb{E}_{t\in[0,1],\mathbf{z}\sim q_t(.|\mathbf{x})} \left[ \sum_{\mathbf{z}'\neq\mathbf{z}_t} R_t(\mathbf{z},\mathbf{z}') \left( \mathbf{s}_\theta(\mathbf{z})_{\mathbf{z}'} - \frac{q_t(\mathbf{z}'|\mathbf{x})}{q_t(\mathbf{z}|\mathbf{x})} \log \mathbf{s}_\theta(\mathbf{z})_{\mathbf{z}'} + K\left( \frac{q_t(\mathbf{z}'|\mathbf{x})}{q_t(\mathbf{z}|\mathbf{x})} \right) \right) \right], \tag{52}$$

where $K(a) = a(\log a - 1)$, for $a \in \mathbb{R}^+$.

**SEDD with Interpolating Uniform Noise**  From (1) we have $q_t(\mathbf{z}|\mathbf{x}) = \alpha_t \mathbf{x}_i + (1-\alpha_t)/N$ where $i$ is the non-zero index of the one-hot vector $\mathbf{z}$, i.e., $\mathbf{z}_i = 1$. Thus, the 'true' conditional score in (52) can be written as

$$\frac{q_t(\mathbf{z}'|\mathbf{x})}{q_t(\mathbf{z}|\mathbf{x})} = \frac{\alpha_t \mathbf{x}_j + (1-\alpha_t)/N}{\alpha_t \mathbf{x}_i + (1-\alpha_t)/N} = \frac{N\alpha_t \mathbf{x}_j + (1-\alpha_t)}{N\alpha_t \mathbf{x}_i + (1-\alpha_t)}, \tag{53}$$

where we use $i$ and $j$ to denote the non-zero indices of the one-hot vectors $\mathbf{z}$ and $\mathbf{z}'$, respectively.

**Using Mean Parameterization in SEDD NELBO**  In our work, we use the mean parameterization: we predict the 'clean' data given noisy observations using the model that we denote as $\mathbf{x}_\theta$. Note that (52) is minimized if

$$\mathbf{s}_\theta(\mathbf{z})_{\mathbf{z}'} = \frac{q_t(\mathbf{z}'|\mathbf{x})}{q_t(\mathbf{z}|\mathbf{x})}.$$

Thus, we can replace $\mathbf{x}$ in (53) to extract a score model from our parameterization:

$$\mathbf{s}_\theta(\mathbf{z})_{\mathbf{z}'} = \frac{\alpha_t(\mathbf{x}_\theta)_j + (1-\alpha_t)/N}{\alpha_t(\mathbf{x}_\theta)_i + (1-\alpha_t)/N} = \frac{N\alpha_t(\mathbf{x}_\theta)_j + (1-\alpha_t)}{N\alpha_t(\mathbf{x}_\theta)_i + (1-\alpha_t)}. \tag{54}$$

We now show two useful identities that come from this parameterization. First,

$$
\begin{aligned}
\sum_{\mathbf{z}' \neq \mathbf{z}} \mathbf{s}_\theta(\mathbf{z})_{\mathbf{z}'} &= \sum_{j \neq i} \frac{\alpha_t(\mathbf{x}_\theta)_j + (1 - \alpha_t)/N}{\alpha_t(\mathbf{x}_\theta)_i + (1 - \alpha_t)/N} \\
&= \frac{\alpha_t[\sum_{j \neq i}(\mathbf{x}_\theta)_j] + \sum_{j \neq i}(1 - \alpha_t)/N}{\alpha_t(\mathbf{x}_\theta)_i + (1 - \alpha_t)/N} \\
&= \frac{\alpha_t[\sum_{j \neq i}(\mathbf{x}_\theta)_j] + (1 - \alpha_t)(N - 1)/N}{\alpha_t(\mathbf{x}_\theta)_i + (1 - \alpha_t)/N} \qquad \because \sum_{j=1}^{N} 1 = N \implies \sum_{j \neq i} 1 = N - 1 \\
&= \frac{\alpha_t[1 - (\mathbf{x}_\theta)_i] + (1 - \alpha_t) - (1 - \alpha_t)/N}{\alpha_t(\mathbf{x}_\theta)_i + (1 - \alpha_t)/N} \qquad \because \sum_{j}(\mathbf{x}_\theta)_j = 1 \implies \sum_{j \neq i}(\mathbf{x}_\theta)_j = 1 - (\mathbf{x}_\theta)_i \\
&= \frac{\alpha_t + (1 - \alpha_t) - \alpha_t(\mathbf{x}_\theta)_i - (1 - \alpha_t)/N}{\alpha_t(\mathbf{x}_\theta)_i + (1 - \alpha_t)/N} \\
&= \frac{1 - \alpha_t(\mathbf{x}_\theta)_i - (1 - \alpha_t)/N}{\alpha_t(\mathbf{x}_\theta)_i + (1 - \alpha_t)/N} \\
&= \frac{1}{\alpha_t(\mathbf{x}_\theta)_i + (1 - \alpha_t)/N} - 1 \\
&= \frac{N}{N\alpha_t(\mathbf{x}_\theta)_i + (1 - \alpha_t)} - 1 \qquad\qquad\qquad (55)
\end{aligned}
$$

The same logic can be applied to show that

$$
\sum_{\mathbf{z}' \neq \mathbf{z}} \frac{q_t(\mathbf{z}'|\mathbf{x})}{q_t(\mathbf{z}|\mathbf{x})} = \sum_{j \neq i} \frac{\alpha_t \mathbf{x}_j + (1 - \alpha_t)/N}{\alpha_t \mathbf{x}_i + (1 - \alpha_t)/N} = \frac{N}{N\alpha_t \mathbf{x}_i + (1 - \alpha_t)} - 1. \qquad (56)
$$

**Equivalence Between NELBO$_{\text{SEDD}}$ and NELBO$_{\text{UDLM}}$**   We can now use the identities from (55) and (56) to demonstrate that NELBO$_{\text{SEDD}}$ (52) is equivalent to NELBO$_{\text{UDLM}}$ (18).

NELBO$_{\text{SEDD}}$

$$
= \mathbb{E}_{t\in[0,1],\mathbf{z}\sim q_t(.|\mathbf{x})} \left[ \sum_{\mathbf{z}'\neq\mathbf{z}} R_t(\mathbf{z},\mathbf{z}') \left( \mathbf{s}_\theta(\mathbf{z})_{\mathbf{z}'} - \frac{q_t(\mathbf{z}'|\mathbf{x})}{q_t(\mathbf{z}|\mathbf{x})} \log \mathbf{s}_\theta(\mathbf{z})_{\mathbf{z}'} + K\left(\frac{q_t(\mathbf{z}'|\mathbf{x})}{q_t(\mathbf{z}|\mathbf{x})}\right) \right) \right]
$$

$$
= \mathbb{E}_{t\in[0,1],\mathbf{z}\sim q_t(.|\mathbf{x})} \left[ \sum_{\mathbf{z}'\neq\mathbf{z}} -\frac{\alpha'_t}{N\alpha_t} \left( \mathbf{s}_\theta(\mathbf{z})_{\mathbf{z}'} - \frac{q_t(\mathbf{z}'|\mathbf{x})}{q_t(\mathbf{z}|\mathbf{x})} \log \mathbf{s}_\theta(\mathbf{z})_{\mathbf{z}'} + K\left(\frac{q_t(\mathbf{z}'|\mathbf{x})}{q_t(\mathbf{z}|\mathbf{x})}\right) \right) \right] \quad \text{from (47)}
$$

$$
= \mathbb{E}_{t\in[0,1],\mathbf{z}\sim q_t(.|\mathbf{x})} \left[ \frac{\alpha'_t}{N\alpha_t} \left( -\sum_{\mathbf{z}'\neq\mathbf{z}} \mathbf{s}_\theta(\mathbf{z})_{\mathbf{z}'} + \sum_{\mathbf{z}'\neq\mathbf{z}} \frac{q_t(\mathbf{z}'|\mathbf{x})}{q_t(\mathbf{z}|\mathbf{x})} \log \mathbf{s}_\theta(\mathbf{z})_{\mathbf{z}'} - \sum_{\mathbf{z}'\neq\mathbf{z}} K\left(\frac{q_t(\mathbf{z}'|\mathbf{x})}{q_t(\mathbf{z}|\mathbf{x})}\right) \right) \right]
$$

Recall that $K(a) = a\log a - a$

$$
= \mathbb{E}_{t\in[0,1],\mathbf{z}\sim q_t(.|\mathbf{x})} \frac{\alpha'_t}{N\alpha_t} \left[ -\sum_{\mathbf{z}'\neq\mathbf{z}} \mathbf{s}_\theta(\mathbf{z})_{\mathbf{z}'} + \sum_{\mathbf{z}'\neq\mathbf{z}} \frac{q_t(\mathbf{z}'|\mathbf{x})}{q_t(\mathbf{z}|\mathbf{x})} \log \mathbf{s}_\theta(\mathbf{z})_{\mathbf{z}'} - \sum_{\mathbf{z}'\neq\mathbf{z}} \frac{q_t(\mathbf{z}'|\mathbf{x})}{q_t(\mathbf{z}|\mathbf{x})} \log \frac{q_t(\mathbf{z}'|\mathbf{x})}{q_t(\mathbf{z}|\mathbf{x})} + \sum_{\mathbf{z}'\neq\mathbf{z}} \frac{q_t(\mathbf{z}'|\mathbf{x})}{q_t(\mathbf{z}|\mathbf{x})} \right]
$$

Using (55) and (56) we get,

$$
= \mathbb{E}_{t\in[0,1],\mathbf{z}\sim q_t(.|\mathbf{x})} \frac{\alpha'_t}{N\alpha_t} \left[ -\frac{N}{\alpha_t N(\mathbf{x}_\theta)_i + (1-\alpha_t)} + \frac{N}{\alpha_t N\mathbf{x}_i + (1-\alpha_t)} \right.
$$
$$
\left. + \sum_{\mathbf{z}'\neq\mathbf{z}} \frac{q_t(\mathbf{z}'|\mathbf{x})}{q_t(\mathbf{z}|\mathbf{x})} \log \mathbf{s}_\theta(\mathbf{z})_{\mathbf{z}'} - \sum_{\mathbf{z}'\neq\mathbf{z}} \frac{q_t(\mathbf{z}'|\mathbf{x})}{q_t(\mathbf{z}|\mathbf{x})} \log \frac{q_t(\mathbf{z}'|\mathbf{x})}{q_t(\mathbf{z}|\mathbf{x})} \right]
$$

$$
= \mathbb{E}_{t\in[0,1],\mathbf{z}\sim q_t(.|\mathbf{x})} \frac{\alpha'_t}{N\alpha_t} \left[ -\frac{N}{\alpha_t N(\mathbf{x}_\theta)_i + (1-\alpha_t)} + \frac{N}{\alpha_t N\mathbf{x}_i + (1-\alpha_t)} \right.
$$
$$
\left. - \sum_{\mathbf{z}'\neq\mathbf{z}} \frac{q_t(\mathbf{z}'|\mathbf{x})}{q_t(\mathbf{z}|\mathbf{x})} \log \left( \frac{1}{\mathbf{s}_\theta(\mathbf{z})_{\mathbf{z}'}} \frac{q_t(\mathbf{z}'|\mathbf{x})}{q_t(\mathbf{z}|\mathbf{x})} \right) \right]
$$

Using (53) and (54) we get,

$$
= \mathbb{E}_{t\in[0,1],\mathbf{z}\sim q_t(.|\mathbf{x})} \frac{\alpha'_t}{N\alpha_t} \left[ \frac{N}{\alpha_t N\mathbf{x}_i + (1-\alpha_t)} - \frac{N}{\alpha_t N(\mathbf{x}_\theta)_i + (1-\alpha_t)} \right.
$$
$$
\left. - \sum_{\mathbf{z}'\neq\mathbf{z}} \frac{\alpha_t N\mathbf{x}_j + (1-\alpha_t)}{\alpha_t N\mathbf{x}_i + (1-\alpha_t)} \log \left( \frac{\alpha_t N(\mathbf{x}_\theta)_i + (1-\alpha_t)}{\alpha_t N(\mathbf{x}_\theta)_j + (1-\alpha_t)} \cdot \frac{\alpha_t N\mathbf{x}_j + (1-\alpha_t)}{\alpha_t N\mathbf{x}_i + (1-\alpha_t)} \right) \right]
$$

$$
= \text{NELBO}_{\text{UDLM}}
$$

# D   EXPERIMENTAL DETAILS

## D.1   DATASET DETAILS

In this section, we provide more details, e.g., source, train/validation splits, etc., for the the datasets used in this work. For an overview, please see Table 8.

**Species10**   This dataset is a composite of reference genomes from ten diverse species: *Arabidopsis thaliana*, *Caenorhabditis elegans*, *Danio rerio*, *Drosophila melanogaster*, *Felis catus*, *Gallus gallus*, *Gorilla gorilla*, *Homo sapiens*, *Mus musculus*, and *Salmo trutta*, which were downloaded from NCBI refseq database (O'Leary et al., 2016). In Table 9, we provide the assembly accession IDs used to download the reference genomes. Genomes were chunked into non-overlapping segments of 32,768 nucleotides and were tokenized using base-pair-level tokenization.

Training and validation sets were randomly split using 95% / 5%. In Table 10, we specify the size and relative species composition of each split. For guidance, we use the species label. This

Table 8: Relevant details for datasets used in this work.

| Dataset | Tokenizer | Vocab. Size | Input size | Padding? |
|---|---|---|---|---|
| Species10 | Base-pair | 12 | 32,768 | No |
| QM9 | Regex (Schwaller et al., 2019) | 40 | 32 | Yes |
| CIFAR10 | Binned Pixel Intensity | 256 | $32{\times}32{\times}3$ | No |
| text8 | Character | 35 | 256 | No |
| Amazon | `bert-base-uncased` | 30,522 | 128 | Yes |
| LM1B | `bert-base-uncased` | 30,522 | 128 | Yes |

Table 9: Species genomes accession IDs

| Species | Assembly Accession |
|---|---|
| *Arabidopsis thaliana* | `GCF_000001735.4_TAIR10.1` |
| *Caenorhabditis elegans* | `GCF_000002985.6_WBcel235` |
| *Danio rerio* | `GCF_000002035.6_GRCz11` |
| *Drosophila melanogaster* | `GCF_000001215.4_Release_6_plus_ISO1_MT` |
| *Felis catus* | `GCF_018350175.1_F.catus_Fca126_mat1.0` |
| *Gallus gallus* | `GCF_016699485.2_bGalGal1.mat.broiler.GRCg7b` |
| *Gorilla gorilla* | `GCF_029281585.2_NHGRI_mGorGor1-v2.0_pri` |
| *Homo sapiens* | `GCF_000001405.40_GRCh38.p14` |
| *Mus musculus* | `GCF_000001635.27_GRCm39` |
| *Salmo trutta* | `GCF_901001165.1_fSalTru1.1` |

dataset is made available here: `https://huggingface.co/datasets/yairschiff/ten_species`.

**QM9 Molecules** The QM9 dataset comes from Ruddigkeit et al. (2012) and Ramakrishnan et al. (2014). The dataset is comprised of ∼133k small molecules. We process the dataset using the RDKit library (Landrum et al., 2013) to extract 'canonical' SMILES string representations and add the annotations for drug-likeness (QED) and ring-counts. The data are tokenized using a regular expression from Schwaller et al. (2019), which is available here: `https://huggingface.co/yairschiff/qm9-tokenizer`. We used input lengths of 32 tokens (with right-sided padding) for pre-training and generation experiments.

Training and validation sets were randomly split using 95% / 5%. For guidance, we use a cutoff of 90[th] percentile for QED / ring count to generate binary labels. This dataset is made available here: `https://huggingface.co/datasets/yairschiff/qm9`.

**CIFAR10** This widely-used image dataset contains RGB images with $32 \times 32$ pixels per channel. We use the provided train and test splits of 50,000 training images and 10,000 validation images. The dataset contains 10 classes of roughly equal proportion. We tokenize the dataset by rounding to integer pixel intensity values in the range $[0, 255]$. For guidance, we use the image class label.

**text8** The dataset was downloaded from `http://mattmahoney.net/dc/text8.zip`. Data was tokenized at the character level using the lower case letters ['a' - 'z'] and a white-space character. The data was broken into non-overlapping chunks of 256 tokens.

The first 90M characters were used for the training set and the final 5M characters were used as a validation set.

**Amazon Review** The Amazon Review dataset was downloaded from `https://huggingface.co/datasets/fancyzhx/amazon_polarity`. We tokenize using the `bert-base-uncased` tokenizer. Sequences were padded to a max input length of 128 tokens.

Table 10: Species10 train and validation splits statistics. Proportion of each species in the training and validation sets. Overall split size is indicated in parentheses in the column header.

| Species | Train (95% ≈ 16.5B bps) | Validation (5% ≈ 869M bps) |
|---|---|---|
| *Arabidopsis thaliana* | 0.68 | 0.77 |
| *Caenorhabditis elegans* | 0.58 | 0.58 |
| *Danio rerio* | 9.50 | 9.29 |
| *Drosophila melanogaster* | 0.08 | 0.73 |
| *Felis catus* | 13.94 | 14.05 |
| *Gallus gallus* | 6.04 | 5.93 |
| *Gorilla gorilla* | 20.40 | 20.19 |
| *Homo sapiens* | 18.89 | 19.22 |
| *Mus musculus* | 15.69 | 15.60 |
| *Salmo trutta* | 13.49 | 13.64 |

Train and validation splits were used from the downloaded data, with 3.6 M sequences in the training data and 400k sequences in the validation set.

**LM1B** This dataset was downloaded from `https://huggingface.co/datasets/billion-word-benchmark/lm1b`. We tokenize using the `bert-base-uncased` tokenizer. Sequences were padded to a max input length of 128 tokens.

We use the train and validation splits provided in the downloaded data. After chunking the data, our training set consisted of 7M sequences and our validation set consisted of 72k sequences.

## D.2 ARCHITECTURAL DETAILS

In Table 11, we provide an overview of the architectures and parameter counts of the models used for each dataset.

Table 11: Architectures used when training on various datasets.

| Dataset | Architecture | Parameter Count |
|---|---|---|
| Species10 | Mamba (AR) / Caduceus (Diffusion) | 3.5 M (AR) / 4.8 M (Diffusion) |
| QM9 | Transformer | 92.4 M |
| CIFAR10 | UNet | 35.8 M |
| Text8 | Transformer | 92.4 M |
| Amazon | Transformer | 139 M |
| LM1B | Transformer | 139 M |

**Genomics Caduceus** For the Species10 experiments, we use Mamba-based models (Gu & Dao, 2023). The AR model is a standard next-token-prediction Mamba backbone with 8 blocks and hidden dimension 256. For the diffusion models we train with a variant of the Caduceus architecture from Schiff et al. (2024), also with 8 blocks and hidden dimension 256. Specifically, we use the non-reverse complementary equivariant version of Caduceus, dubbed Caduceus-Ph in Schiff et al. (2024), which is similar to the AR Mamba, but with bi-directional Mamba blocks that use strategic weight tying to limit the parameter count: 3.5 M parameters of AR vs. 4.8 M for diffusion models.

**CIFAR10 UNet** We adopt the UNet (Ronneberger et al., 2015) as a main backbone for both MDLM and UDLM, following (Ho et al., 2020). Network configurations are presented in Table 12. Specifically, we follow Austin et al. (2021) and Campbell et al. (2022) and use the original UNet backbone from DDPM (Ho et al., 2020), adding an extra discretized truncated logistic transformation to network outputs. To enable conditioning on labels, we add a label embedding layer which is added to the time embeddings, inspired by ADM network (Dhariwal & Nichol, 2021b).

Table 12: Architecture details for CIFAR10 UNet.

|  | MDLM | UDLM |
|---|---|---|
| Vocab size | 258 | 256 |
| Number of ResNet blocks per scale | 2 | 2 |
| Base channels | 128 | 128 |
| Channel multiplier per scale | (1,2,2,2) | (1,2,2,2) |
| Attention resolutions | 16 | 16 |
| Time conditioning | False | True |
| Conditional embedding dimension | 128 | 128 |
| Number of Params | 35.8M | 35.8M |

**QM9 and NLP Transformer** For both the QM9 and NLP datasets (Amazon, text8, and LM1B), we use the same Diffusion Transformer (Peebles & Xie, 2023) with adaLN for conditioning on time (for uniform noise diffusion models) and class (for guided generation). The Transformer for each dataset differs only in the word-embedding size, which is determined by the vocabulary size of the datasets' corresponding tokenizer. Our Transformer consists of 12 layers, a hidden dimension of 768, and 12 attention heads. We use RoPE (Su et al., 2021) as the positional embeddings.

## D.3 TRAINING CONFIGURATIONS

In Table 13, we detail the hyperparameter setup for each of the language modeling experiments in Section 5.1.

All diffusion models were trained and evaluated using a log-linear noise schedule.

Of note, for CIFAR10, our models are trained for 300K iterations as opposed to 1.5M iterations, as in D3PM (Austin et al., 2021).

Table 13: Training hyper-parameters for all included experiments.

|  | Species10 | QM9 | CIFAR10 | text8 | Amazon | LM1B |
|---|---|---|---|---|---|---|
| Train steps | 30K | 25K | 300K | 1000K | 184K | 1000K |
| Context size | 32,768 | 32 | 32×32×3 | 256 | 128 | 128 |
| Batch size | 32 | 2048 | 512 | 512 | 512 | 512 |
| LR | $2e^{-3}$ | $3e^{-4}$ | $2e^{-4}$ | $3e^{-4}$ | $3e^{-4}$ | $3e^{-4}$ |
| Optim. | ADAM (0.9, 0.999) | ADAM (0.9, 0.999) | ADAM (0.9, 0.999) | ADAM (0.9, 0.999) | ADAM (0.9, 0.999) | ADAM (0.9, 0.999) |
| LR sched. | Cosine decay $3e^{-6}$ min. | Cosine decay $3e^{-6}$ min. | - | - | - | - |
| LR warmup steps | 3K | 1K | 5K | 2.5K | 2.5K | 2.5K |
| GPU count | 8 | 4 | 8 | 8 | 8 | 8 |
| GPU type | A5000 | A6000 | A100 | A100 | A5000 | A100 |

## D.4 GUIDANCE DETAILS

### D.4.1 BASELINES

**FUDGE Implementation** FUDGE is a classifier-based autoregressive guidance method proposed by Yang & Klein (2021). FUDGE first trains a classifier on all possible prefixes. Instead of directly sampling from $p(\mathbf{x}^{(\ell)} \mid \mathbf{x}^{(1:\ell-1)})$, FUDGE samples from the perturbed conditional distribution $p(y \mid \mathbf{x}^{(1:\ell)})p(\mathbf{x}^{(\ell)} \mid \mathbf{x}^{(1:\ell-1)})$ where $y$ denotes the classifier label, $\mathbf{x}^{(1:\ell-1)}$ denotes the already

decoded tokens and $\mathbf{x}^{(\ell)}$ stands for possible token to be generated. For efficiency, FUDGE truncates $\mathbf{x}^{(\ell)}$ to the $topk$ tokens with the largest unconditional log-likelihood. Although not in the original FUDGE formulation, following the classifier-based guidance in diffusion models (Dhariwal & Nichol (2021b)), we introduce the temperature variable $\gamma$ and sample from the perturbed distribution:

$$\frac{p(\mathbf{x}^{(\ell)} \mid \mathbf{x}^{(1:\ell-1)})p^{\gamma}(y \mid \mathbf{x}^{(1:\ell)})}{\sum_{\mathbf{x}^{(\ell)}} p(\mathbf{x}^{(\ell)} \mid \mathbf{x}^{(1:\ell-1)})p^{\gamma}(y \mid \mathbf{x}^{(1:\ell)})}$$

In our QM9 experiments, $topk$ is set as the vocabulary size, which is 40. When training the classifier, we use a smaller classification model using the same dataset specified in Appendix D.1. The smaller backbone DiT consists of 8 layers, 8 attention heads, and a hidden dimension of 512. We use the same hyperparameters as those specified in Table 13.

**PPLM Implementation** Inspired by the approximate Metropolis-adjusted Langevin (MALA), Plug and Play language model (PPLM; Dathathri et al. (2019)) introduces guidance by conducting gradient updates on the hidden representations of a language model. PPLM first rewrites the autoregressive language models' decoding process using the KV-cache mechanism (Wolf et al., 2020): $\mathbf{x}_{t+1}, H_{t+1} = LM(\mathbf{x}_t, H_t)$, where $\mathbf{x}_t$ denotes the token at time step $t$ and $H_t$ is defined as $[(K_t^{(1)}, V_t^{(1)}), ..., (K_t^{(l)}, V_t^{(l)}]$ in which $l$ is the number of layers. When decoding the token $\mathbf{x}_{t+1}$ at time step $t + 1$, PPLM initializes the perturbation term $\Delta H_t$ as 0, and then performs $n$ updates, following the gradient ascend formula:

$$\Delta H_t \leftarrow \Delta H_t + \eta \frac{\nabla_{\Delta H_t}[\log(p(y \mid H_t + \Delta H_t)) - \gamma_{KL} D_{KL}(p(\mathbf{x}_{t+1} \mid H_t + \Delta H_t) || p(\mathbf{x}_{t+1} \mid H_t))]}{\|\nabla_{\Delta H_t}[log(p(y \mid H_t + \Delta H_t)) - \gamma_{KL} D_{KL}(p(\mathbf{x}_{t+1} \mid H_t + \Delta H_t) || p(\mathbf{x}_{t+1} \mid H_t))]\|}.$$

PPLM then generates the perturbed probability distribution $p_{perb}(\mathbf{x}_{t+1} \mid H_t + \Delta H_t)$ using $H_t + \Delta H_t$ and decodes $\mathbf{x}_{t+1}$ using the fused probability distribution $\frac{1}{\beta} p_{perb}^{\gamma_{gm}}(\mathbf{x}_{t+1} \mid H_t + \Delta H_t) p_{unperb}^{1-\gamma_{gm}}(\mathbf{x}_{t+1} \mid H_t)$, where $\beta$ is the normalization factor. Following Dathathri et al. (2019), in our experiments $\gamma_{kl}$ is set as 0.01 and $\gamma_{gm}$ is set as 0.95. $\eta$ and $n$ are decided by grid search.

For the QM9 experiments, when training the classifier for PPLM, we use the same dataset specified in Appendix D.1. We use the same hyperparameters as those specified in Table 13, except the LR peak is reduced to $3e^{-5}$ and minimum is reduced to $3e^{-7}$.

**NOS Implementation** To implement the NOS baseline from Gruver et al. (2024), we train a classifier where we use the same backbone as the unconditional diffusion model (see Appendix D.2 for architecture details), and we initialize and freeze weights using the pre-trained unconditional diffusion model. We then mean pool the last hidden embeddings of the backbone and linearly project them to the classification logits. This final projection layer represents the only trainable parameters of the guidance model.

At inference, we perform Langevin sampling following Algorithm 2 from Gruver et al. (2024). In our experiments, we denote step-size for the Langevin sampling by $\eta$, the number of steps by $n$, and the weight on the 'stability' / 'fluency' KL penalty by $\gamma_{kl}$.

For the QM9 experiments, when training the classifier for NOS, we use the same dataset specified in Appendix D.1. We use the same hyperparameters as those specified in Table 13, except the LR peak is reduced to $3e^{-5}$ and minimum is reduced to $3e^{-7}$.

### D.4.2 D-CFG DETAILS

When implementing D-CFG we train a single AR / discrete diffusion model (see Appendix D.2 for architecture details) where we randomly drop out the class condition by replacing it with a class [MASK] token. The class condition is fused into models using the implementation of adaptive layer norm from Peebles & Xie (2023). We use 10% rate for masking / dropping-out the condition.

At inference time, we perform two forward passes through the model, one with condition provided to compute the conditional probability $p_\theta(\mathbf{z}_s \mid \mathbf{z}_t, y)$ and one where the condition is masked to compute the unconditional probability $p_\theta(\mathbf{z}_s \mid \mathbf{z}_t)$. These values are then used as described in Section 3.1.

### D.4.3 D-CBG Details

For D-CBG in the QM9 experiment, we train a smaller classification model using the same dataset specified in Appendix D.1. The smaller backbone DiT consists of 8 layers, 8 attention heads, and a hidden dimension of 512. We apply mean pooling on the final hidden representations before linearly projecting to the classification logits. We use the same hyperparameters as those specified in Table 13.

We train the model on noised inputs using a log-linear schedule, where the type of corruption applied corresponds to the diffusion model to which guidance is applied.

## D.5 Guidance Evaluation Details

### D.5.1 Genomic Sequences Metrics

Below we describe the quality and control metrics used in the species-specific genome generation experiment.

$k$**-mer JS**    To compute the $k$-mer distirbution shift, for each species, we create counts for each of the unique 3mers and 6mers for that species in the validation set and 64 generated sequences for that species. We then compute the Jensen-Shannon divergence between those categorical histograms. Finally we take a weighted average of these distances across species where the weights are given by the relative species proportion in the validation dataset. See Table 10 for the relative proportions.

**Discriminator AUROC**    For the discriminator AUROC metric, we train a HyenaDNA model with 2 layers and hidden dimension 128. This model was downloaded from `https://huggingface.co/LongSafari/hyenadna-small-32k-seqlen-hf`, modified (we reduced the number of layers and hidden dimension), and initialized from scratch. We mean pool the final layer embeddings and linearly project them to the binary classification logits. The model is trained on the 640 generated sequences, which are labeled as the negative class, and 640 randomly selected sequences from the ground truth validation set (64 sequences per species), which are labeled as the positive class. This dataset of 1,280 sequences is randomly split into 95% train and 5% validation. The discriminator is trained with batch size of 8, learning rate of $1\mathrm{e}^{-4}$, and the ADAM optimizer for 5 epochs to minimize binary cross entropy loss on the classification of real vs. generated sequences. We report the AUROC from the final epoch on the 5% validation split of this classification dataset.

**Oracle F1**    Finally, controllability is measured by the macro-averaged F1 of an oracle model on the 640 generated sequences. Our oracle model is a separate HyenaDNA model with 8 layers and hidden dimension 256, which has 6.6M parameters. This model was downloaded and initialized from scratch from `https://huggingface.co/LongSafari/hyenadna-small-32k-seqlen-hf`. We mean pool the final layer embeddings and linearly project them to the ten category classification logits. This model was trained on the full Species10 dataset as described in Appendix D.1. For reference, in Table 14 we present the classification results of the this oracle model on the 5% valdiation set of the original data. We see that other than difficulty distinguishing between human and gorilla genomes, the model can serve as a near perfect oracle.

### D.5.2 CIFAR10 Quality Metrics

For evaluation, we randomly samples 50,000 images for each model and the tools provided here: `https://github.com/w86763777/pytorch-image-generation-metrics.git`, as described in Campbell et al. (2022).

**FID**    Fréchet inception distance (Heusel et al., 2017) is a common metric in image generation where the divergence between real and generated data is measured to reflect the alignment of two distributions. The metric uses features extracted from a pretrained Inception-v3 model on ImageNet-1K to estimate the mean and variance of the input data. The difference of two multi-dimensional Gaussian distributions is measured by Wasserstein-2 distance or Fréchet distance $d(.)$ as follows:

Table 14: Evaluation of HyenaDNA 'oracle' classifier on Species10 validation split.

| Species | Precision | Recall | F1 |
|---|---|---|---|
| *Arabidopsis thaliana* | 1.00 | 0.99 | 1.00 |
| *Caenorhabditis elegans* | 1.00 | 1.00 | 1.00 |
| *Danio rerio* | 1.00 | 1.00 | 1.00 |
| *Drosophila melanogaster* | 1.00 | 1.00 | 1.00 |
| *Felis catus* | 1.00 | 1.00 | 1.00 |
| *Gallus gallus* | 1.00 | 1.00 | 1.00 |
| *Gorilla gorilla* | 0.63 | 0.45 | 0.52 |
| *Homo sapiens* | 0.54 | 0.72 | 0.62 |
| *Mus musculus* | 1.00 | 0.97 | 0.98 |
| *Salmo trutta* | 1.00 | 0.98 | 0.99 |

$$FID = d(\mathcal{N}(\mu_{real}, \Sigma_{real}), \mathcal{N}(\mu_{fake}, \Sigma_{fake})) = \|\mu_{real} - \mu_{fake}\| +$$
$$\|\text{Tr}(\Sigma_{real} + \Sigma_{fake} - 2(\Sigma_{real}\Sigma_{fake})^{0.5})\| \quad (57)$$

**IS** Inception Score (Salimans et al., 2016) is an alternative measure of how well generated images are aligned with human judgement. IS also utilizes Inception-v3 model to compute label distribution $p(y|x)$ for each generated image. IS focuses on two criteria: (1) A generated image should contain a distinct class object, meaning its label distribution is expected to have low entropy; (2) The generated images should vary across multiple classes, so the marginal distribution, $p(y) = \int_z p(y|G(z))dz$, is expected to have high entropy, ideally approaching a uniform distribution. The formula is presented as below:

$$IS = \exp\left[\mathbb{E}_x \text{KL}(p(y|x)\|p(y))\right], \quad (58)$$

**F1** F1 is used as a proxy for satisfying the desired conditional generation of generated samples by a pre-trained classifier on CIFAR10. We use a pretrained Vision Transformer model downloaded from https://huggingface.co/edadaltocg/vit_base_patch16_224_in21k_ft_cifar10 and fine-tuned on CIFAR10.

### D.5.3 QM9 GUIDANCE METRICS

For the guidance experiments in QM9, we generate 2,048 sequences of length 32. The reported metrics are explained below.

**Validity** Validity is measured by whether the generated SMILES string can be parsed by the RDKit library (Landrum et al., 2013). Any strings that fail to be parsed are counted as invalid.

**Novelty** Novelty is measured by the number of valid and unique sequences that are not present in the original QM9 dataset.

**Property Mean / Median** Finally, we also report the mean and median of the novel generated sequence for the property of interest, QED or ring count. These quantities are computed using the RDKit library.

## E    GUIDANCE ABLATION RESULTS

### E.1    EFFECT OF VARYING GUIDANCE HYPERPARAMETERS

For each of the guidance experiments, we perform a hyerparameter search on the guidance parameters, e.g., $\gamma$ in D-CFG and D-CBG. Below we present results from these searches for the various guidance experiments.

**Species-specific Genome Generation**   In this experiment we vary $\gamma \in \{1, 2, 3\}$ for D-CFG applied to AR, MDLM, and UDLM. Additionally, for MDLM and UDLM we vary the number of sampling steps $T \in \{128, 256, 512\}$.

Results presented in Table 15 highlight that UDLM is more amenable to guidance than either AR or MDLM in this setting. UDLM achieves better quality and control metrics, and AR and MDLM performance degrades as $\gamma$, this is not the case for UDLM.

Table 15: Varying $\gamma$ for species-specific generation with AR, MDLM, UDLM using D-CFG. Mean $\pm$ standard deviation reported from repeated generation of 640 sequences (64 per species) using five different random seeds.

| D-CFG$_\gamma$ | $T$ | 3-mer JS ($\downarrow$) | 6-mer JS ($\downarrow$) | Disc. AUROC ($\downarrow$) | F1 ($\uparrow$) |
|---|---|---|---|---|---|
| *AR* | | | | | |
| 1 | 32,768 | $0.03 \pm 0.002$ | $0.07 \pm 0.004$ | $0.53 \pm 0.075$ | $0.87 \pm 0.011$ |
| 2 | 32,768 | $0.05 \pm 0.003$ | $0.12 \pm 0.004$ | $0.90 \pm 0.051$ | $0.81 \pm 0.009$ |
| 3 | 32,768 | $0.07 \pm 0.002$ | $0.15 \pm 0.002$ | $0.97 \pm 0.018$ | $0.74 \pm 0.022$ |
| *MDLM* | | | | | |
| 1 | 128 | $0.02 \pm 0.001$ | $0.06 \pm 0.001$ | $0.51 \pm 0.073$ | $0.88 \pm 0.01$ |
| 1 | 256 | $0.02 \pm 0.002$ | $0.06 \pm 0.003$ | $0.55 \pm 0.039$ | $0.88 \pm 0.006$ |
| 1 | 512 | $0.02 \pm 0.001$ | $0.06 \pm 0.001$ | $0.51 \pm 0.059$ | $0.88 \pm 0.011$ |
| 2 | 128 | $0.06 \pm 0.001$ | $0.11 \pm 0.003$ | $0.71 \pm 0.013$ | $0.90 \pm 0.011$ |
| 2 | 256 | $0.05 \pm 0.001$ | $0.10 \pm 0.002$ | $0.77 \pm 0.027$ | $0.91 \pm 0.009$ |
| 2 | 512 | $0.05 \pm 0.001$ | $0.11 \pm 0.003$ | $0.74 \pm 0.037$ | $0.91 \pm 0.009$ |
| 3 | 128 | $0.12 \pm 0.002$ | $0.20 \pm 0.003$ | $0.94 \pm 0.031$ | $0.78 \pm 0.009$ |
| 3 | 256 | $0.11 \pm 0.002$ | $0.19 \pm 0.005$ | $0.91 \pm 0.025$ | $0.78 \pm 0.01$ |
| 3 | 512 | $0.11 \pm 0.004$ | $0.20 \pm 0.005$ | $0.93 \pm 0.031$ | $0.78 \pm 0.007$ |
| *UDLM* | | | | | |
| 1 | 128 | $0.02 \pm 0.001$ | $0.05 \pm 0.001$ | $0.55 \pm 0.041$ | $0.90 \pm 0.006$ |
| 1 | 256 | $0.02 \pm 0.002$ | $0.06 \pm 0.002$ | $0.56 \pm 0.042$ | $0.91 \pm 0.006$ |
| 1 | 512 | $0.02 \pm 0.002$ | $0.06 \pm 0.003$ | $0.52 \pm 0.044$ | $0.91 \pm 0.01$ |
| 2 | 128 | $0.05 \pm 0.003$ | $0.12 \pm 0.004$ | $0.52 \pm 0.02$ | $0.92 \pm 0.006$ |
| 2 | 256 | $0.05 \pm 0.002$ | $0.13 \pm 0.004$ | $0.59 \pm 0.043$ | $0.91 \pm 0.005$ |
| 2 | 512 | $0.05 \pm 0.002$ | $0.13 \pm 0.002$ | $0.61 \pm 0.043$ | $0.93 \pm 0.009$ |
| 3 | 128 | $0.08 \pm 0.003$ | $0.19 \pm 0.004$ | $0.80 \pm 0.048$ | $0.93 \pm 0.009$ |
| 3 | 256 | $0.08 \pm 0.002$ | $0.20 \pm 0.004$ | $0.81 \pm 0.084$ | $0.92 \pm 0.005$ |
| 3 | 512 | $0.08 \pm 0.002$ | $0.20 \pm 0.003$ | $0.87 \pm 0.051$ | $0.94 \pm 0.007$ |

**QM9**   In this section, we list the hyperparameter grid search results for QM9 drug likeliness (QED) maximization using

- AR D-CFG (Table 16), AR FUDGE (Table 17), AR PPLM (Table 18),
- MDLM D-CFG (Table 19), MDLM D-CBG (Table 20), MDLM NOS (Table 21),
- UDLM D-CFG (Table 22), UDLM D-CBG (Table 23), and UDLM NOS (Table 24).

We also list the grid search results for QM9 ring count maximization guidance results with

- AR D-CFG (Table 25), AR FUDGE (Table 26) AR PPLM (Table 27),
- MDLM D-CFG (Table 28) MDLM D-CBG (Table 29), MDLM NOS (Table 30),
- UDLM D-CFG (Table 31), UDLM D-CBG (Table 32), UDLM NOS (Table 33).

**CIFAR10**   In Table 34, we explore the effect of $\gamma \in 1, 2, 3, 4, 5$ in conditional image generation. For this table, we use $T = 1000$. We find that increasing $\gamma$ generally leads to better IS and F1 scores for both MDLM and UDLM. For FID, MDLM achieves better scores as $\gamma$ increases. However, the impact of $\gamma$ is weaker for UDLM, as the model's FID score worsens when $\gamma$ exceeds 2. We also plot

Table 16: Varying $\gamma$ for maximizing drug-likeness (QED) guidance with AR D-CFG. Validity, novelty, and mean QED for novel sequences are reported. Mean $\pm$ standard deviation reported from repeated generation of 1,024 sequences using five different random seeds. The setting reported in the main paper is **bolded**.

| $\gamma$ | Num. Valid ($\uparrow$) | Num. Novel ($\uparrow$) | QED Mean ($\uparrow$) |
|---|---|---|---|
| 1 | $1013.0 \pm 4.9$ | $82.6 \pm 2.97$ | $0.57 \pm 0.00$ |
| 2 | $1001.4 \pm 5.37$ | $79.2 \pm 9.34$ | $0.59 \pm 0.00$ |
| **3** | $\mathbf{946.4} \pm \mathbf{8.99}$ | $\mathbf{79.4} \pm \mathbf{6.35}$ | $\mathbf{0.60} \pm \mathbf{0.00}$ |
| 4 | $777.2 \pm 15.39$ | $86.8 \pm 8.73$ | $0.59 \pm 0.00$ |
| 5 | $591.8 \pm 10.43$ | $77.8 \pm 9.12$ | $0.58 \pm 0.00$ |

Table 17: Varying $\gamma$ for maximizing drug-likeness (QED) guidance with AR FUDGE. Validity, novelty, and mean QED for novel sequences are reported. Mean $\pm$ standard deviation reported from repeated generation of 1,024 sequences using five different random seeds. The setting reported in the main paper is **bolded**.

| $\gamma$ | Num. Valid ($\uparrow$) | Num. Novel ($\uparrow$) | QED Mean ($\uparrow$) |
|---|---|---|---|
| 1 | $998.4 \pm 3.21$ | $113.0 \pm 7.18$ | $0.56 \pm 0.00$ |
| 2 | $985.2 \pm 6.46$ | $88.8 \pm 5.5$ | $0.58 \pm 0.01$ |
| 3 | $973.8 \pm 4.97$ | $90.8 \pm 7.66$ | $0.59 \pm 0.00$ |
| 4 | $963.4 \pm 8.53$ | $86.4 \pm 14.05$ | $0.59 \pm 0.00$ |
| 5 | $949.6 \pm 10.71$ | $74.6 \pm 9.07$ | $0.60 \pm 0.00$ |
| 6 | $935.0 \pm 9.33$ | $64.6 \pm 7.09$ | $0.60 \pm 0.00$ |
| **7** | $\mathbf{924.4} \pm \mathbf{12.05}$ | $\mathbf{53.0} \pm \mathbf{3.54}$ | $\mathbf{0.61} \pm \mathbf{0.00}$ |
| 8 | $902.8 \pm 16.08$ | $46.6 \pm 4.04$ | $0.62 \pm 0.00$ |
| 9 | $876.6 \pm 13.03$ | $34.8 \pm 3.63$ | $0.62 \pm 0.01$ |
| 10 | $873.8 \pm 11.21$ | $29.2 \pm 2.77$ | $0.62 \pm 0.01$ |

visual outputs of different $\gamma$ in Figure 4. As $\gamma$ is increased, the appearance and the shape of a class object become more refined and sharpened.

### E.2 EFFECT OF USING FIRST ORDER APPROXIMATION IN D-CBG

The data for this analysis is available in Tables 20, 23, 29, and 32. We present it graphically in Figure 5. In these plots, we use $T = 32$ for diffusion generation.

**QED** We see that for drug-likeness (QED) maximization, using the full D-CBG without the first order approximation leads to a boost in performance for both MDLM and UDLM.

**Ring Count** When maximizing ring count, we again find that for UDLM, not relying on the first order approximation improves performance with both more novel molecule generation and higher ring counts. For MDLM, however, we observe instability for $\gamma > 1$, as the model tends to decode the full sequence early on in the generation process and cannot recover from mistakes. The first order approximation for MDLM does not appear to suffer from this same instability.

### E.3 EFFECT OF VARYING $T$

**Species10** In Table 15, we see the effect of increasing $T$ on the genomic sequence generation. In this setting $T \in \{128, 256, 512\}$ is orders of magnitude smaller than sequence length $L = 32768$. We see a positive relationship between decoding steps and sample quality for both MDLM and UDLM.

**QM9** In Figure 6, we see that for both properties (QED and ring count), MDLM benefits significantly from increasing $T$. In contrast, UDLM appears to have more consistent results across varying

Table 18: Varying $n$ (the number of Langevin steps) and $\eta$ (the step size) for maximizing drug-likeness (QED) guidance with AR PPLM. Validity, novelty, and mean QED for novel sequences are measured for generated sequences from each method. Mean $\pm$ standard deviation reported from repeated generation of 1,024 sequences using five different random seeds. The setting reported in the main paper is **bolded**.

| $n$ | $\eta$ | Num. Valid ($\uparrow$) | Num. Novel ($\uparrow$) | Novel QED Mean ($\uparrow$) |
|---|---|---|---|---|
| 10 | 0.040 | $1009.6 \pm 2.61$ | $140.4 \pm 18.19$ | $0.46 \pm 0.01$ |
| 10 | 0.100 | $1008.8 \pm 2.17$ | $138.4 \pm 17.24$ | $0.46 \pm 0.01$ |
| 30 | 0.040 | $1008.8 \pm 2.77$ | $137.8 \pm 16.83$ | $0.45 \pm 0.00$ |
| **30** | **0.100** | $\mathbf{1007.2} \pm \mathbf{2.28}$ | $\mathbf{142.0} \pm \mathbf{14.2}$ | $\mathbf{0.45} \pm \mathbf{0.00}$ |

Table 19: Varying $\gamma$ for maximizing drug-likeness (QED) guidance with MDLM D-CFG. Validity, novelty, and mean QED for novel sequences are reported. Mean $\pm$ standard deviation reported from repeated generation of 1,024 sequences using five different random seeds. The setting reported in the main paper is **bolded**.

| $\gamma$ | Num. Valid ($\uparrow$) | Num. Novel ($\uparrow$) | Novel QED Mean ($\uparrow$) |
|---|---|---|---|
| 1 | $561.0 \pm 6.08$ | $182.8 \pm 10.52$ | $0.56 \pm 0.00$ |
| 2 | $449.8 \pm 8.7$ | $142.8 \pm 8.47$ | $0.59 \pm 0.01$ |
| **3** | $\mathbf{317.4} \pm \mathbf{11.5}$ | $\mathbf{95.8} \pm \mathbf{9.04}$ | $\mathbf{0.60} \pm \mathbf{0.01}$ |
| 4 | $226.0 \pm 10.34$ | $79.0 \pm 6.36$ | $0.59 \pm 0.01$ |
| 5 | $163.4 \pm 7.86$ | $68.4 \pm 3.58$ | $0.59 \pm 0.01$ |

number of decoding steps. Note that for the D-CBG results in this section we do not use the first order approximation for maximizing QED and we do use it for maximizing ring count.

**CIFAR10** In Table 35, we present additional $T$ beyond those presented in Table 7 in Section 5.2. As discussed in the main paper results, we find that UDLM is able to accommodate this faster inference setting better than MDLM, owing to the ability of UDLM to recover from 'mistakes.'

# F  UNCONDITIONAL SAMPLE GENERATION

In this section, we evaluate unconditional samples generated from models trained on LM1B. In Table 36, we report generative perplexity for 1,024 unconditionally generated sequences. We use GPT-2 Large (Radford et al., 2019) downloaded from https://huggingface.co/openai-community/gpt2-large to compute generative perplexity. In Table 37, we present random examples of sample generation.

Table 20: Varying $\gamma$ for maximizing drug-likeness (QED) guidance with MDLM D-CBG. We also report results for both using the first order approximation and not using it. Validity, novelty, and mean QED for novel sequences are reported. Mean $\pm$ standard deviation reported from repeated generation of 1,024 sequences using five different random seeds. The setting reported in the main paper is **bolded**.

| $\gamma$ | Use Approx. | Num. Valid ($\uparrow$) | Num. Novel ($\uparrow$) | Novel QED Mean ($\uparrow$) |
|---|---|---|---|---|
| 1 | False | $526.4 \pm 16.53$ | $170.8 \pm 14.69$ | $0.56 \pm 0.00$ |
| 1 | True | $476.0 \pm 15.23$ | $221.4 \pm 7.57$ | $0.46 \pm 0.01$ |
| 2 | False | $524.8 \pm 22.04$ | $139.0 \pm 11.79$ | $0.58 \pm 0.00$ |
| 2 | True | $363.4 \pm 17.57$ | $172.4 \pm 9.42$ | $0.47 \pm 0.01$ |
| **3** | **False** | $\mathbf{417.6} \pm \mathbf{19.69}$ | $\mathbf{116.6} \pm \mathbf{8.91}$ | $\mathbf{0.58} \pm \mathbf{0.00}$ |
| 3 | True | $244.0 \pm 13.38$ | $114.0 \pm 6.16$ | $0.47 \pm 0.01$ |
| 4 | False | $200.4 \pm 10.31$ | $66.4 \pm 7.7$ | $0.58 \pm 0.00$ |
| 4 | True | $174.6 \pm 11.26$ | $83.6 \pm 3.51$ | $0.49 \pm 0.02$ |
| 5 | False | $24.6 \pm 3.21$ | $11.6 \pm 2.3$ | $0.58 \pm 0.01$ |
| 5 | True | $120.6 \pm 6.69$ | $48.4 \pm 5.41$ | $0.50 \pm 0.01$ |
| 6 | False | $0.2 \pm 0.45$ | $0.2 \pm 0.45$ | $0.12 \pm 0.26$ |
| 6 | True | $96.4 \pm 4.04$ | $35.0 \pm 5.24$ | $0.50 \pm 0.02$ |
| 7 | False | $0.0 \pm 0.00$ | $0.0 \pm 0.00$ | $0.00 \pm 0.00$ |
| 7 | True | $85.4 \pm 9.81$ | $28.6 \pm 5.32$ | $0.51 \pm 0.01$ |
| 8 | False | $0.0 \pm 0.00$ | $0.00 \pm 0.00$ | $0.00 \pm 0.00$ |
| 8 | True | $77.0 \pm 8.63$ | $24.8 \pm 6.76$ | $0.55 \pm 0.01$ |
| 9 | False | $0.0 \pm 0.00$ | $0.0 \pm 0.00$ | $0.00 \pm 0.00$ |
| 9 | True | $71.2 \pm 10.94$ | $21.6 \pm 6.73$ | $0.55 \pm 0.01$ |
| 10 | False | $0.0 \pm 0.00$ | $0.0 \pm 0.00$ | $0.00 \pm 0.00$ |
| 10 | True | $72.6 \pm 7.13$ | $21.2 \pm 1.79$ | $0.55 \pm 0.02$ |

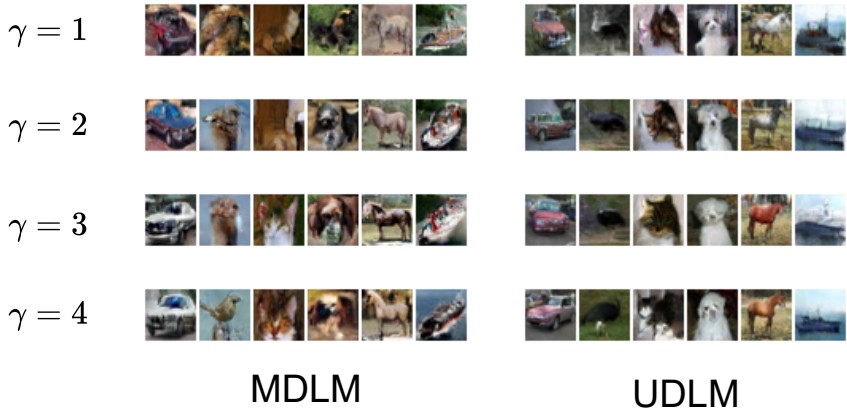

Figure 4: Illustration of varying $\gamma$ on CIFAR10.

Table 21: Varying $n$ (the number of Langevin steps), $\eta$ (the step size), and $\gamma_{kl}$ (the stability regularization coefficient) for maximizing drug-likeness (QED) guidance with MDLM NOS. Validity, novelty, and mean ring count for novel sequences are measured for generated sequences from each method. Mean $\pm$ standard deviation reported from repeated generation of 1,024 sequences using five different random seeds. The setting reported in the main paper is **bolded**.

| $n$ | $\eta$ | $\gamma_{kl}$ | Num. Valid ($\uparrow$) | Num. Novel ($\uparrow$) | Novel QED Mean ($\uparrow$) |
|---|---|---|---|---|---|
| 1 | 0.001 | 0.000 | $509.4 \pm {\scriptstyle 23.78}$ | $238.2 \pm {\scriptstyle 8.87}$ | $0.45 \pm {\scriptstyle 0.00}$ |
| 1 | 0.001 | 0.001 | $509.4 \pm {\scriptstyle 23.78}$ | $238.2 \pm {\scriptstyle 8.87}$ | $0.45 \pm {\scriptstyle 0.00}$ |
| 1 | 0.001 | 0.010 | $509.4 \pm {\scriptstyle 23.78}$ | $238.2 \pm {\scriptstyle 8.87}$ | $0.45 \pm {\scriptstyle 0.00}$ |
| 1 | 0.010 | 0.000 | $510.00 \pm {\scriptstyle 24.24}$ | $238.8 \pm {\scriptstyle 7.82}$ | $0.45 \pm {\scriptstyle 0.00}$ |
| 1 | 0.010 | 0.001 | $509.4 \pm {\scriptstyle 24.15}$ | $238.6 \pm {\scriptstyle 7.77}$ | $0.45 \pm {\scriptstyle 0.00}$ |
| 1 | 0.010 | 0.010 | $509.4 \pm {\scriptstyle 24.15}$ | $238.6 \pm {\scriptstyle 7.77}$ | $0.45 \pm {\scriptstyle 0.00}$ |
| 1 | 0.100 | 0.000 | $507.4 \pm {\scriptstyle 29.98}$ | $240.2 \pm {\scriptstyle 14.82}$ | $0.45 \pm {\scriptstyle 0.00}$ |
| 1 | 0.100 | 0.001 | $507.4 \pm {\scriptstyle 29.98}$ | $240.2 \pm {\scriptstyle 14.82}$ | $0.45 \pm {\scriptstyle 0.00}$ |
| 1 | 0.100 | 0.010 | $507.4 \pm {\scriptstyle 29.98}$ | $240.2 \pm {\scriptstyle 14.82}$ | $0.45 \pm {\scriptstyle 0.00}$ |
| 1 | 1.000 | 0.000 | $461.0 \pm {\scriptstyle 29.21}$ | $233.2 \pm {\scriptstyle 15.16}$ | $0.44 \pm {\scriptstyle 0.00}$ |
| 1 | 1.000 | 0.001 | $461.0 \pm {\scriptstyle 29.21}$ | $233.2 \pm {\scriptstyle 15.16}$ | $0.44 \pm {\scriptstyle 0.00}$ |
| 1 | 1.000 | 0.010 | $461.0 \pm {\scriptstyle 29.21}$ | $233.2 \pm {\scriptstyle 15.16}$ | $0.44 \pm {\scriptstyle 0.00}$ |
| 1 | 5.000 | 0.000 | $185.8 \pm {\scriptstyle 8.23}$ | $127.0 \pm {\scriptstyle 8.25}$ | $0.42 \pm {\scriptstyle 0.01}$ |
| 1 | 5.000 | 0.001 | $185.8 \pm {\scriptstyle 8.23}$ | $127.0 \pm {\scriptstyle 8.25}$ | $0.42 \pm {\scriptstyle 0.01}$ |
| 1 | 5.000 | 0.010 | $185.8 \pm {\scriptstyle 8.23}$ | $127.0 \pm {\scriptstyle 8.25}$ | $0.42 \pm {\scriptstyle 0.01}$ |
| 5 | 0.001 | 0.000 | $509.4 \pm {\scriptstyle 24.67}$ | $237.0 \pm {\scriptstyle 9.67}$ | $0.45 \pm {\scriptstyle 0.00}$ |
| 5 | 0.001 | 0.001 | $509.4 \pm {\scriptstyle 24.67}$ | $237.0 \pm {\scriptstyle 9.67}$ | $0.45 \pm {\scriptstyle 0.00}$ |
| 5 | 0.001 | 0.010 | $509.4 \pm {\scriptstyle 24.67}$ | $237.0 \pm {\scriptstyle 9.67}$ | $0.45 \pm {\scriptstyle 0.00}$ |
| 5 | 0.010 | 0.000 | $509.2 \pm {\scriptstyle 26.2}$ | $238.4 \pm {\scriptstyle 8.79}$ | $0.45 \pm {\scriptstyle 0.01}$ |
| 5 | 0.010 | 0.001 | $509.2 \pm {\scriptstyle 26.2}$ | $238.4 \pm {\scriptstyle 8.79}$ | $0.45 \pm {\scriptstyle 0.01}$ |
| 5 | 0.010 | 0.010 | $509.2 \pm {\scriptstyle 26.2}$ | $238.4 \pm {\scriptstyle 8.79}$ | $0.45 \pm {\scriptstyle 0.01}$ |
| 5 | 0.100 | 0.000 | $505.6 \pm {\scriptstyle 29.52}$ | $239.8 \pm {\scriptstyle 16.36}$ | $0.45 \pm {\scriptstyle 0.01}$ |
| 5 | 0.100 | 0.001 | $505.6 \pm {\scriptstyle 29.52}$ | $239.8 \pm {\scriptstyle 16.36}$ | $0.45 \pm {\scriptstyle 0.01}$ |
| **5** | **0.100** | **0.010** | $\mathbf{506.0} \pm {\scriptstyle \mathbf{29.45}}$ | $\mathbf{240.4} \pm {\scriptstyle \mathbf{16.01}}$ | $\mathbf{0.45} \pm {\scriptstyle \mathbf{0.01}}$ |
| 5 | 1.000 | 0.000 | $461.0 \pm {\scriptstyle 29.21}$ | $233.2 \pm {\scriptstyle 15.16}$ | $0.44 \pm {\scriptstyle 0.00}$ |
| 5 | 1.000 | 0.001 | $461.6 \pm {\scriptstyle 28.4}$ | $233.2 \pm {\scriptstyle 14.77}$ | $0.44 \pm {\scriptstyle 0.00}$ |
| 5 | 1.000 | 0.010 | $459.8 \pm {\scriptstyle 28.6}$ | $231.2 \pm {\scriptstyle 13.72}$ | $0.44 \pm {\scriptstyle 0.00}$ |
| 5 | 5.000 | 0.000 | $185.8 \pm {\scriptstyle 8.23}$ | $127.0 \pm {\scriptstyle 8.25}$ | $0.42 \pm {\scriptstyle 0.01}$ |
| 5 | 5.000 | 0.001 | $186.2 \pm {\scriptstyle 8.98}$ | $127.2 \pm {\scriptstyle 8.79}$ | $0.42 \pm {\scriptstyle 0.01}$ |
| 5 | 5.000 | 0.010 | $186.0 \pm {\scriptstyle 9.59}$ | $127.0 \pm {\scriptstyle 9.46}$ | $0.42 \pm {\scriptstyle 0.01}$ |
| 10 | 0.001 | 0.000 | $509.6 \pm {\scriptstyle 25.58}$ | $238.0 \pm {\scriptstyle 9.77}$ | $0.45 \pm {\scriptstyle 0.00}$ |
| 10 | 0.001 | 0.001 | $509.0 \pm {\scriptstyle 25.46}$ | $237.4 \pm {\scriptstyle 9.56}$ | $0.45 \pm {\scriptstyle 0.00}$ |
| 10 | 0.001 | 0.010 | $509.0 \pm {\scriptstyle 25.46}$ | $237.4 \pm {\scriptstyle 9.56}$ | $0.45 \pm {\scriptstyle 0.00}$ |
| 10 | 0.010 | 0.000 | $508.2 \pm {\scriptstyle 25.77}$ | $238.4 \pm {\scriptstyle 11.08}$ | $0.45 \pm {\scriptstyle 0.00}$ |
| 10 | 0.010 | 0.001 | $508.2 \pm {\scriptstyle 25.77}$ | $238.4 \pm {\scriptstyle 11.08}$ | $0.45 \pm {\scriptstyle 0.00}$ |
| 10 | 0.010 | 0.010 | $508.2 \pm {\scriptstyle 25.77}$ | $238.4 \pm {\scriptstyle 11.08}$ | $0.45 \pm {\scriptstyle 0.00}$ |
| 10 | 0.100 | 0.000 | $504.2 \pm {\scriptstyle 26.75}$ | $238.2 \pm {\scriptstyle 14.81}$ | $0.45 \pm {\scriptstyle 0.01}$ |
| 10 | 0.100 | 0.001 | $505.0 \pm {\scriptstyle 26.84}$ | $238.8 \pm {\scriptstyle 15.25}$ | $0.45 \pm {\scriptstyle 0.01}$ |
| 10 | 0.100 | 0.010 | $504.2 \pm {\scriptstyle 26.75}$ | $238.2 \pm {\scriptstyle 14.81}$ | $0.45 \pm {\scriptstyle 0.01}$ |
| 10 | 1.000 | 0.000 | $461.0 \pm {\scriptstyle 29.21}$ | $233.2 \pm {\scriptstyle 15.16}$ | $0.44 \pm {\scriptstyle 0.00}$ |
| 10 | 1.000 | 0.001 | $461.0 \pm {\scriptstyle 28.31}$ | $232.8 \pm {\scriptstyle 14.08}$ | $0.44 \pm {\scriptstyle 0.00}$ |
| 10 | 1.000 | 0.010 | $460.6 \pm {\scriptstyle 28.41}$ | $231.8 \pm {\scriptstyle 12.99}$ | $0.45 \pm {\scriptstyle 0.00}$ |
| 10 | 5.000 | 0.000 | $185.8 \pm {\scriptstyle 8.23}$ | $127.0 \pm {\scriptstyle 8.25}$ | $0.42 \pm {\scriptstyle 0.01}$ |
| 10 | 5.000 | 0.001 | $186.4 \pm {\scriptstyle 7.92}$ | $127.4 \pm {\scriptstyle 7.96}$ | $0.42 \pm {\scriptstyle 0.01}$ |
| 10 | 5.000 | 0.010 | $190.4 \pm {\scriptstyle 10.48}$ | $130.4 \pm {\scriptstyle 9.63}$ | $0.42 \pm {\scriptstyle 0.01}$ |

Table 22: Varying $\gamma$ for maximizing drug-likeness (QED) guidance with UDLM D-CFG. Validity, novelty, and mean QED for novel sequences are reported. Mean $\pm$ standard deviation reported from repeated generation of 1,024 sequences using five different random seeds. The setting reported in the main paper is **bolded**.

| $\gamma$ | Num. Valid ($\uparrow$) | Num. Novel ($\uparrow$) | Novel QED Mean ($\uparrow$) |
|---|---|---|---|
| 1 | $1003.4 \pm 5.46$ | $82.0 \pm 15.28$ | $0.57 \pm 0.00$ |
| 2 | $1017.6 \pm 1.82$ | $64.2 \pm 3.96$ | $0.60 \pm 0.00$ |
| 3 | $1017.2 \pm 0.84$ | $61.2 \pm 7.19$ | $0.61 \pm 0.00$ |
| 4 | $1019.0 \pm 1.41$ | $61.6 \pm 9.13$ | $0.61 \pm 0.00$ |
| **5** | **$1013.6 \pm 2.51$** | **$64.0 \pm 5.1$** | **$0.62 \pm 0.00$** |

Table 23: Varying $\gamma$ for maximizing drug-likeness (QED) guidance with UDLM D-CBG. We also report results for both using the first order approximation and not using it. Validity, novelty, and mean QED for novel sequences are reported. Mean $\pm$ standard deviation reported from repeated generation of 1,024 sequences using five different random seeds. The setting reported in the main paper is **bolded**.

| $\gamma$ | Use Approx. | Num. Valid ($\uparrow$) | Num. Novel ($\uparrow$) | Novel QED Mean ($\uparrow$) |
|---|---|---|---|---|
| 1 | False | $933.4 \pm 7.5$ | $134.6 \pm 7.33$ | $0.53 \pm 0.01$ |
| 1 | True | $996.4 \pm 5.86$ | $132.2 \pm 8.87$ | $0.47 \pm 0.01$ |
| 2 | False | $911.2 \pm 8.14$ | $119.8 \pm 16.47$ | $0.57 \pm 0.01$ |
| 2 | True | $974.8 \pm 8.26$ | $110.6 \pm 9.53$ | $0.49 \pm 0.01$ |
| 3 | False | $941.0 \pm 11.31$ | $96.4 \pm 8.82$ | $0.58 \pm 0.01$ |
| 3 | True | $925.0 \pm 8.09$ | $111.4 \pm 9.56$ | $0.51 \pm 0.01$ |
| 4 | False | $961.8 \pm 7.98$ | $77.0 \pm 7.55$ | $0.59 \pm 0.00$ |
| 4 | True | $866.4 \pm 17.67$ | $93.8 \pm 10.28$ | $0.52 \pm 0.00$ |
| 5 | False | $967.6 \pm 1.52$ | $77.8 \pm 10.83$ | $0.59 \pm 0.01$ |
| 5 | True | $828.2 \pm 17.85$ | $86.2 \pm 13.01$ | $0.53 \pm 0.01$ |
| 6 | False | $976.2 \pm 7.92$ | $78.6 \pm 12.92$ | $0.60 \pm 0.00$ |
| 6 | True | $821.0 \pm 11.22$ | $79.8 \pm 7.19$ | $0.54 \pm 0.00$ |
| 7 | False | $982.4 \pm 3.21$ | $73.8 \pm 6.91$ | $0.60 \pm 0.00$ |
| 7 | True | $798.6 \pm 5.64$ | $82.2 \pm 7.66$ | $0.55 \pm 0.00$ |
| 8 | False | $986.4 \pm 4.98$ | $66.6 \pm 8.53$ | $0.61 \pm 0.01$ |
| 8 | True | $789.8 \pm 15.09$ | $69.2 \pm 6.76$ | $0.55 \pm 0.01$ |
| 9 | False | $992.8 \pm 6.3$ | $68.0 \pm 1.22$ | $0.61 \pm 0.00$ |
| 9 | True | $798.4 \pm 11.06$ | $68.2 \pm 10.5$ | $0.57 \pm 0.01$ |
| **10** | **False** | **$994.8 \pm 2.86$** | **$63.8 \pm 8.11$** | **$0.61 \pm 0.00$** |
| 10 | True | $783.2 \pm 13.88$ | $67.0 \pm 3.08$ | $0.56 \pm 0.01$ |

Table 24: Varying $n$ (the number of Langevin steps), $\eta$ (the step size), and $\gamma_{kl}$ (the stability regularization coefficient) for maximizing drug-likeness (QED) guidance with UDLM NOS. Validity, novelty, and mean ring count for novel sequences are measured for generated sequences from each method. Mean $\pm$ standard deviation reported from repeated generation of 1,024 sequences using five different random seeds. The setting reported used in the main paper is **bolded**.

| $n$ | $\eta$ | $\gamma_{kl}$ | Num. Valid ($\uparrow$) | Num. Novel ($\uparrow$) | Novel QED Mean ($\uparrow$) |
|---|---|---|---|---|---|
| 1 | 0.001 | 0.000 | $1000.8 \pm 2.77$ | $145.6 \pm 8.44$ | $0.46 \pm 0.00$ |
| 1 | 0.001 | 0.001 | $1000.8 \pm 2.77$ | $145.6 \pm 8.44$ | $0.46 \pm 0.00$ |
| 1 | 0.001 | 0.010 | $1000.8 \pm 2.77$ | $145.8 \pm 8.04$ | $0.46 \pm 0.00$ |
| 1 | 0.010 | 0.000 | $1001.0 \pm 3.0$ | $147.0 \pm 7.81$ | $0.46 \pm 0.01$ |
| 1 | 0.010 | 0.001 | $1001.0 \pm 3.0$ | $147.0 \pm 7.81$ | $0.46 \pm 0.01$ |
| 1 | 0.010 | 0.010 | $1001.0 \pm 3.0$ | $147.0 \pm 7.81$ | $0.46 \pm 0.01$ |
| 1 | 0.100 | 0.000 | $999.2 \pm 4.6$ | $146.4 \pm 6.88$ | $0.46 \pm 0.01$ |
| 1 | 0.100 | 0.001 | $999.2 \pm 4.6$ | $146.4 \pm 6.88$ | $0.46 \pm 0.01$ |
| 1 | 0.100 | 0.010 | $999.2 \pm 4.6$ | $146.4 \pm 6.88$ | $0.46 \pm 0.01$ |
| 1 | 1.000 | 0.000 | $992.2 \pm 5.63$ | $152.0 \pm 7.25$ | $0.46 \pm 0.00$ |
| 1 | 1.000 | 0.001 | $992.2 \pm 5.63$ | $152.0 \pm 7.25$ | $0.46 \pm 0.00$ |
| 1 | 1.000 | 0.010 | $992.2 \pm 5.63$ | $152.0 \pm 7.25$ | $0.46 \pm 0.00$ |
| 1 | 5.000 | 0.000 | $357.4 \pm 20.84$ | $106.4 \pm 12.66$ | $0.47 \pm 0.01$ |
| 1 | 5.000 | 0.001 | $357.4 \pm 20.84$ | $106.4 \pm 12.66$ | $0.47 \pm 0.01$ |
| 1 | 5.000 | 0.010 | $357.4 \pm 20.84$ | $106.4 \pm 12.66$ | $0.47 \pm 0.01$ |
| 5 | 0.001 | 0.000 | $1000.8 \pm 2.77$ | $147.0 \pm 7.91$ | $0.46 \pm 0.00$ |
| 5 | 0.001 | 0.001 | $1000.8 \pm 2.77$ | $147.0 \pm 7.91$ | $0.46 \pm 0.00$ |
| 5 | 0.001 | 0.010 | $1000.8 \pm 2.77$ | $147.0 \pm 7.91$ | $0.46 \pm 0.00$ |
| 5 | 0.010 | 0.000 | $1000.4 \pm 2.3$ | $146.0 \pm 8.97$ | $0.46 \pm 0.01$ |
| 5 | 0.010 | 0.001 | $1000.4 \pm 2.3$ | $146.0 \pm 8.97$ | $0.46 \pm 0.01$ |
| 5 | 0.010 | 0.010 | $1000.4 \pm 2.3$ | $146.0 \pm 8.97$ | $0.46 \pm 0.01$ |
| 5 | 0.100 | 0.000 | $1000.4 \pm 5.68$ | $144.8 \pm 5.36$ | $0.46 \pm 0.01$ |
| 5 | 0.100 | 0.001 | $1000.2 \pm 5.63$ | $144.4 \pm 5.18$ | $0.46 \pm 0.01$ |
| 5 | 0.100 | 0.010 | $1000.2 \pm 5.63$ | $145.2 \pm 5.22$ | $0.46 \pm 0.01$ |
| 5 | 1.000 | 0.000 | $992.2 \pm 5.63$ | $152.0 \pm 7.25$ | $0.46 \pm 0.00$ |
| 5 | 1.000 | 0.001 | $991.8 \pm 4.21$ | $151.0 \pm 5.66$ | $0.46 \pm 0.01$ |
| 5 | 1.000 | 0.010 | $994.0 \pm 5.74$ | $151.4 \pm 5.68$ | $0.46 \pm 0.01$ |
| 5 | 5.000 | 0.000 | $357.4 \pm 20.84$ | $106.4 \pm 12.66$ | $0.47 \pm 0.01$ |
| 5 | 5.000 | 0.001 | $373.8 \pm 13.2$ | $109.0 \pm 8.09$ | $0.47 \pm 0.01$ |
| 5 | 5.000 | 0.010 | $456.6 \pm 10.29$ | $135.8 \pm 11.41$ | $0.47 \pm 0.01$ |
| 10 | 0.001 | 0.000 | $1000.8 \pm 2.77$ | $146.8 \pm 8.14$ | $0.46 \pm 0.01$ |
| 10 | 0.001 | 0.001 | $1000.8 \pm 2.77$ | $146.8 \pm 8.14$ | $0.46 \pm 0.01$ |
| 10 | 0.001 | 0.010 | $1001.0 \pm 2.74$ | $146.2 \pm 8.04$ | $0.46 \pm 0.01$ |
| 10 | 0.010 | 0.000 | $1000.8 \pm 2.77$ | $146.4 \pm 10.64$ | $0.46 \pm 0.00$ |
| 10 | 0.010 | 0.001 | $1001.0 \pm 3.46$ | $146.0 \pm 10.46$ | $0.46 \pm 0.00$ |
| 10 | 0.010 | 0.010 | $1000.6 \pm 3.13$ | $146.4 \pm 10.09$ | $0.46 \pm 0.00$ |
| 10 | 0.100 | 0.000 | $1001.2 \pm 5.26$ | $144.6 \pm 3.29$ | $0.46 \pm 0.01$ |
| 10 | 0.100 | 0.001 | $1000.6 \pm 5.68$ | $144.4 \pm 2.88$ | $0.46 \pm 0.01$ |
| 10 | 0.100 | 0.010 | $1000.6 \pm 6.31$ | $144.8 \pm 1.92$ | $0.46 \pm 0.01$ |
| 10 | 1.000 | 0.000 | $992.2 \pm 5.63$ | $152.0 \pm 7.25$ | $0.46 \pm 0.00$ |
| 10 | 1.000 | 0.001 | $991.0 \pm 3.81$ | $149.8 \pm 6.22$ | $0.46 \pm 0.01$ |
| 10 | 1.000 | 0.010 | $993.6 \pm 5.55$ | $150.6 \pm 3.97$ | $0.46 \pm 0.00$ |
| 10 | 5.000 | 0.000 | $357.4 \pm 20.84$ | $106.4 \pm 12.66$ | $0.47 \pm 0.01$ |
| 10 | 5.000 | 0.001 | $386.0 \pm 15.02$ | $111.4 \pm 6.35$ | $0.47 \pm 0.01$ |
| **10** | **5.000** | **0.010** | $\mathbf{547.8 \pm 10.57}$ | $\mathbf{158.8 \pm 10.99}$ | $\mathbf{0.47 \pm 0.00}$ |

Table 25: Varying $\gamma$ for maximizing ring count guidance with AR D-CFG. Validity, novelty, and mean ring count for novel sequences are reported. Mean $\pm$ standard deviation reported from repeated generation of 1,024 sequences using five different random seeds. The setting reported in the main paper is **bolded**.

| $\gamma$ | Num. Valid ($\uparrow$) | Num. Novel ($\uparrow$) | Novel Ring Count Mean ($\uparrow$) |
|---|---|---|---|
| 1 | $1003.4 \pm 4.72$ | $189.0 \pm 6.44$ | $4.53 \pm 0.07$ |
| 2 | $966.6 \pm 5.32$ | $217.4 \pm 14.5$ | $4.76 \pm 0.10$ |
| 3 | $856.4 \pm 10.21$ | $213.0 \pm 12.75$ | $4.81 \pm 0.08$ |
| 4 | $647.6 \pm 16.07$ | $137.4 \pm 9.21$ | $4.75 \pm 0.10$ |
| **5** | $\mathbf{441.4} \pm \mathbf{11.08}$ | $\mathbf{77.8} \pm \mathbf{2.28}$ | $\mathbf{4.83} \pm \mathbf{0.08}$ |

Table 26: Varying $\gamma$ for maximizing ring count guidance with AR FUDGE. Validity, novelty, and mean ring count for novel sequences are reported. Mean $\pm$ standard deviation reported from repeated generation of 1,024 sequences using five different random seeds. The setting reported in the main paper is **bolded**.

| $\gamma$ | Num. Valid ($\uparrow$) | Num. Novel ($\uparrow$) | Novel Ring Count Mean ($\uparrow$) |
|---|---|---|---|
| 1 | $956.8 \pm 3.11$ | $220.2 \pm 27.53$ | $4.50 \pm 0.10$ |
| 2 | $861.6 \pm 6.07$ | $239.4 \pm 20.89$ | $4.70 \pm 0.06$ |
| 3 | $847.2 \pm 8.44$ | $266.4 \pm 13.76$ | $4.80 \pm 0.06$ |
| 4 | $704.0 \pm 5.2$ | $241.0 \pm 18.37$ | $4.87 \pm 0.04$ |
| **5** | $\mathbf{281.2} \pm \mathbf{4.97}$ | $\mathbf{103.6} \pm \mathbf{4.83}$ | $\mathbf{4.89} \pm \mathbf{0.10}$ |
| 6 | $50.6 \pm 5.86$ | $19.4 \pm 4.34$ | $5.02 \pm 0.23$ |
| 7 | $8.6 \pm 4.34$ | $4.0 \pm 1.58$ | $4.89 \pm 0.44$ |
| 8 | $1.0 \pm 1.41$ | $0.4 \pm 0.55$ | $2.20 \pm 3.03$ |
| 9 | $0.0 \pm 0.0$ | $0.0 \pm 0.0$ | $0.00 \pm 0.00$ |
| 10 | $0.0 \pm 0.0$ | $0.0 \pm 0.0$ | $0.00 \pm 0.00$ |

Table 27: Varying $n$ (the number of Langevin steps) and $\eta$ (the step size) for maximizing ring count guidance with AR PPLM. Validity, novelty, and mean ring count for novel sequences are measured for generated sequences from each method. Mean $\pm$ standard deviation reported from repeated generation of 1,024 sequences using five different random seeds. The setting reported in the main paper is **bolded**.

| $n$ | $\eta$ | Num. Valid ($\uparrow$) | Num. Novel ($\uparrow$) | Novel Ring Count Mean ($\uparrow$) |
|---|---|---|---|---|
| 10 | 0.04 | $1009.8 \pm 1.92$ | $140.6 \pm 16.47$ | $1.92 \pm 0.09$ |
| **10** | **0.1** | $\mathbf{1009.6} \pm \mathbf{2.07}$ | $\mathbf{140.2} \pm \mathbf{16.25}$ | $\mathbf{1.92} \pm \mathbf{0.06}$ |
| 30 | 0.04 | $1010.0 \pm 2.35$ | $139.4 \pm 14.74$ | $1.91 \pm 0.09$ |
| 30 | 0.1 | $1010.0 \pm 1.58$ | $138.6 \pm 10.31$ | $1.88 \pm 0.12$ |

Table 28: Varying $\gamma$ for maximizing ring count guidance with MDLM D-CFG. Validity, novelty, and mean ring count for novel sequences are reported. Mean $\pm$ standard deviation reported from repeated generation of 1,024 sequences using five different random seeds. The setting reported in the main paper is **bolded**.

| $\gamma$ | Num. Valid ($\uparrow$) | Num. Novel ($\uparrow$) | Novel Ring Count Mean ($\uparrow$) |
|---|---|---|---|
| 1 | $465.0 \pm 18.56$ | $273.8 \pm 11.95$ | $4.52 \pm 0.05$ |
| 2 | $363.6 \pm 17.5$ | $215.6 \pm 11.1$ | $4.85 \pm 0.05$ |
| 3 | $242.4 \pm 15.85$ | $144.6 \pm 16.89$ | $5.01 \pm 0.08$ |
| 4 | $152.2 \pm 4.44$ | $97.4 \pm 5.08$ | $5.10 \pm 0.13$ |
| 5 | $\mathbf{90.0} \pm \mathbf{8.15}$ | $\mathbf{60.0} \pm \mathbf{7.52}$ | $\mathbf{5.26} \pm \mathbf{0.15}$ |

Table 29: Varying $\gamma$ for maximizing ring count guidance with MDLM D-CBG. We also report results for both using the first order approximation and not using it. Validity, novelty, and mean ring count for novel sequences are reported. Mean $\pm$ standard deviation reported from repeated generation of 1,024 sequences using five different random seeds. The setting reported in the main paper is **bolded**.

| $\gamma$ | Use Approx. | Num. Valid ($\uparrow$) | Num. Novel ($\uparrow$) | Novel Ring Count Mean ($\uparrow$) |
|---|---|---|---|---|
| 1 | False | $143.8 \pm 6.61$ | $121.2 \pm 7.01$ | $5.00 \pm 0.07$ |
| 1 | True | $455.4 \pm 24.81$ | $229.4 \pm 15.21$ | $2.52 \pm 0.11$ |
| 2 | False | $0.4 \pm 0.55$ | $0.4 \pm 0.55$ | $3.00 \pm 4.47$ |
| 2 | True | $327.6 \pm 15.82$ | $176.4 \pm 17.3$ | $2.82 \pm 0.15$ |
| 3 | False | $0.0 \pm 0.0$ | $0.0 \pm 0.0$ | $0.00 \pm 0.00$ |
| 3 | True | $223.8 \pm 15.01$ | $136.0 \pm 9.8$ | $3.25 \pm 0.26$ |
| 4 | False | $0.0 \pm 0.0$ | $0.0 \pm 0.0$ | $0.00 \pm 0.00$ |
| 4 | True | $158.0 \pm 9.27$ | $106.6 \pm 7.13$ | $3.73 \pm 0.19$ |
| 5 | False | $0.0 \pm 0.0$ | $0.0 \pm 0.0$ | $0.00 \pm 0.00$ |
| 5 | True | $135.0 \pm 9.9$ | $94.4 \pm 11.06$ | $4.11 \pm 0.26$ |
| 6 | False | $0.0 \pm 0.0$ | $0.0 \pm 0.0$ | $0.00 \pm 0.00$ |
| 6 | True | $122.8 \pm 11.3$ | $89.8 \pm 7.82$ | $4.41 \pm 0.11$ |
| 7 | False | $0.0 \pm 0.0$ | $0.0 \pm 0.0$ | $0.00 \pm 0.00$ |
| 7 | True | $116.4 \pm 12.5$ | $85.4 \pm 10.38$ | $4.59 \pm 0.10$ |
| 8 | False | $0.0 \pm 0.0$ | $0.0 \pm 0.0$ | $0.00 \pm 0.00$ |
| 8 | True | $121.2 \pm 12.91$ | $92.6 \pm 11.95$ | $4.54 \pm 0.16$ |
| 9 | False | $0.0 \pm 0.0$ | $0.0 \pm 0.0$ | $0.00 \pm 0.00$ |
| 9 | True | $112.2 \pm 12.28$ | $87.4 \pm 11.84$ | $4.70 \pm 0.17$ |
| 10 | False | $0.0 \pm 0.0$ | $0.0 \pm 0.0$ | $0.00 \pm 0.00$ |
| **10** | **True** | $\mathbf{113.0} \pm \mathbf{8.75}$ | $\mathbf{85.6} \pm \mathbf{8.82}$ | $\mathbf{4.75} \pm \mathbf{0.23}$ |

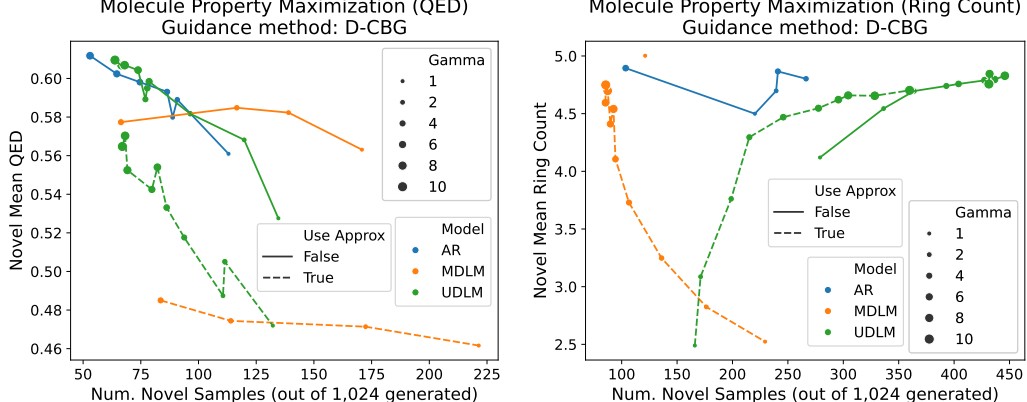

Figure 5: Ablating the use of first order approximation when applying D-CBG to discrete diffusion models. *(Left)* Maximizing drug-likeness (QED). *(Right)* Maximizing ring count.

Table 30: Varying $n$ (the number of Langevin steps), $\eta$ (the step size), and $\gamma_{kl}$ (the stability regularization coefficient) for maximizing ring count guidance with MDLM NOS. Validity, novelty, and mean ring count for novel sequences are measured for generated sequences from each method. Mean $\pm$ standard deviation reported from repeated generation of 1,024 sequences using five different random seeds. The setting reported used in the main paper is **bolded**.

| $n$ | $\eta$ | $\gamma_{kl}$ | Num. Valid ($\uparrow$) | Num. Novel ($\uparrow$) | Novel Ring Count Mean ($\uparrow$) |
|---|---|---|---|---|---|
| 1 | 0.001 | 0.000 | $508.8 \pm 24.18$ | $237.0 \pm 8.57$ | $2.41 \pm 0.16$ |
| 1 | 0.001 | 0.001 | $508.8 \pm 24.18$ | $237.0 \pm 8.57$ | $2.41 \pm 0.16$ |
| 1 | 0.001 | 0.010 | $508.8 \pm 24.18$ | $237.0 \pm 8.57$ | $2.41 \pm 0.16$ |
| 1 | 0.010 | 0.000 | $509.6 \pm 24.27$ | $237.0 \pm 10.37$ | $2.41 \pm 0.16$ |
| 1 | 0.010 | 0.001 | $509.6 \pm 24.27$ | $237.4 \pm 10.45$ | $2.41 \pm 0.16$ |
| 1 | 0.010 | 0.010 | $509.6 \pm 24.27$ | $237.0 \pm 10.37$ | $2.41 \pm 0.16$ |
| 1 | 0.100 | 0.000 | $505.2 \pm 25.71$ | $237.4 \pm 13.2$ | $2.46 \pm 0.15$ |
| 1 | 0.100 | 0.001 | $504.4 \pm 25.59$ | $236.4 \pm 12.66$ | $2.46 \pm 0.15$ |
| 1 | 0.100 | 0.010 | $505.2 \pm 25.71$ | $237.4 \pm 13.2$ | $2.46 \pm 0.15$ |
| 1 | 1.000 | 0.000 | $458.4 \pm 26.28$ | $250.00 \pm 12.35$ | $2.79 \pm 0.14$ |
| 1 | 1.000 | 0.001 | $458.4 \pm 26.28$ | $250.00 \pm 12.35$ | $2.79 \pm 0.14$ |
| 1 | 1.000 | 0.010 | $458.4 \pm 26.28$ | $250.00 \pm 12.35$ | $2.79 \pm 0.14$ |
| 1 | 5.000 | 0.000 | $187.0 \pm 11.6$ | $141.4 \pm 13.96$ | $3.08 \pm 0.24$ |
| 1 | 5.000 | 0.001 | $187.0 \pm 11.6$ | $141.4 \pm 13.96$ | $3.08 \pm 0.24$ |
| 1 | 5.000 | 0.010 | $187.0 \pm 11.6$ | $141.4 \pm 13.96$ | $3.08 \pm 0.24$ |
| 5 | 0.001 | 0.000 | $509.8 \pm 24.71$ | $237.6 \pm 9.79$ | $2.4 \pm 0.16$ |
| 5 | 0.001 | 0.001 | $510.00 \pm 24.66$ | $237.6 \pm 9.79$ | $2.4 \pm 0.16$ |
| 5 | 0.001 | 0.010 | $510.00 \pm 24.66$ | $237.6 \pm 9.79$ | $2.4 \pm 0.16$ |
| 5 | 0.010 | 0.000 | $509.4 \pm 24.44$ | $238.0 \pm 11.47$ | $2.42 \pm 0.17$ |
| 5 | 0.010 | 0.001 | $509.2 \pm 24.43$ | $237.8 \pm 11.21$ | $2.42 \pm 0.17$ |
| 5 | 0.010 | 0.010 | $509.4 \pm 24.42$ | $238.0 \pm 11.25$ | $2.42 \pm 0.17$ |
| 5 | 0.100 | 0.000 | $506.0 \pm 21.25$ | $238.6 \pm 11.74$ | $2.48 \pm 0.17$ |
| 5 | 0.100 | 0.001 | $506.4 \pm 20.21$ | $238.0 \pm 10.54$ | $2.48 \pm 0.14$ |
| 5 | 0.100 | 0.010 | $505.0 \pm 21.06$ | $236.6 \pm 10.06$ | $2.48 \pm 0.16$ |
| 5 | 1.000 | 0.000 | $458.4 \pm 26.28$ | $250.00 \pm 12.35$ | $2.79 \pm 0.14$ |
| 5 | 1.000 | 0.001 | $464.4 \pm 21.0$ | $252.4 \pm 11.95$ | $2.79 \pm 0.14$ |
| 5 | 1.000 | 0.010 | $466.4 \pm 22.04$ | $254.0 \pm 11.38$ | $2.78 \pm 0.1$ |
| 5 | 5.000 | 0.000 | $187.0 \pm 11.6$ | $141.4 \pm 13.96$ | $3.08 \pm 0.24$ |
| 5 | 5.000 | 0.001 | $213.6 \pm 17.36$ | $164.0 \pm 18.33$ | $3.31 \pm 0.19$ |
| **5** | **5.000** | **0.010** | $\mathbf{247.8 \pm 13.97}$ | $\mathbf{193.6 \pm 13.13}$ | $\mathbf{3.51 \pm 0.22}$ |
| 10 | 0.001 | 0.000 | $509.0 \pm 24.14$ | $237.2 \pm 8.7$ | $2.41 \pm 0.16$ |
| 10 | 0.001 | 0.001 | $509.8 \pm 24.28$ | $238.0 \pm 9.0$ | $2.4 \pm 0.17$ |
| 10 | 0.001 | 0.010 | $509.8 \pm 24.28$ | $238.0 \pm 9.0$ | $2.4 \pm 0.17$ |
| 10 | 0.010 | 0.000 | $510.4 \pm 24.42$ | $239.4 \pm 10.21$ | $2.43 \pm 0.14$ |
| 10 | 0.010 | 0.001 | $510.2 \pm 24.41$ | $239.6 \pm 10.36$ | $2.42 \pm 0.14$ |
| 10 | 0.010 | 0.010 | $510.00 \pm 24.4$ | $239.6 \pm 10.36$ | $2.43 \pm 0.14$ |
| 10 | 0.100 | 0.000 | $504.0 \pm 19.26$ | $238.4 \pm 10.74$ | $2.48 \pm 0.19$ |
| 10 | 0.100 | 0.001 | $505.6 \pm 18.15$ | $239.6 \pm 10.36$ | $2.48 \pm 0.16$ |
| 10 | 0.100 | 0.010 | $506.4 \pm 18.66$ | $240.4 \pm 9.79$ | $2.48 \pm 0.17$ |
| 10 | 1.000 | 0.000 | $458.4 \pm 26.28$ | $250.00 \pm 12.35$ | $2.79 \pm 0.14$ |
| 10 | 1.000 | 0.001 | $471.4 \pm 18.01$ | $257.8 \pm 10.89$ | $2.8 \pm 0.1$ |
| 10 | 1.000 | 0.010 | $473.2 \pm 18.86$ | $257.0 \pm 7.91$ | $2.76 \pm 0.12$ |
| 10 | 5.000 | 0.000 | $187.0 \pm 11.6$ | $141.4 \pm 13.96$ | $3.08 \pm 0.24$ |
| 10 | 5.000 | 0.001 | $225.6 \pm 17.47$ | $176.8 \pm 18.65$ | $3.32 \pm 0.17$ |
| 10 | 5.000 | 0.010 | $265.4 \pm 12.62$ | $206.6 \pm 10.88$ | $3.47 \pm 0.24$ |

Table 31: Varying $\gamma$ for maximizing ring count guidance with UDLM D-CFG. Validity, novelty, and mean ring count for novel sequences are reported. Mean $\pm$ standard deviation reported from repeated generation of 1,024 sequences using five different random seeds. The setting reported in the main paper is **bolded**.

| $\gamma$ | Num. Valid ($\uparrow$) | Num. Novel ($\uparrow$) | Novel Ring Count Mean ($\uparrow$) |
|---|---|---|---|
| 1 | 979.0 $\pm$ 8.83 | 276.8 $\pm$ 17.34 | 4.54 $\pm$ 0.05 |
| 2 | 998.2 $\pm$ 6.83 | 229.0 $\pm$ 8.94 | 4.75 $\pm$ 0.08 |
| **3** | **998.2** $\pm$ **4.49** | **216.2** $\pm$ **12.99** | **4.88** $\pm$ **0.04** |
| 4 | 989.6 $\pm$ 5.94 | 211.4 $\pm$ 2.88 | 4.85 $\pm$ 0.08 |
| 5 | 968.6 $\pm$ 7.89 | 209.4 $\pm$ 13.2 | 4.84 $\pm$ 0.06 |

Table 32: Varying $\gamma$ for maximizing ring count guidance with UDLM D-CBG. We also report results for both using the first order approximation and not using it. Validity, novelty, and mean ring count for novel sequences are reported. Mean $\pm$ standard deviation reported from repeated generation of 1,024 sequences using five different random seeds. The setting reported in the main paper is **bolded**.

| $\gamma$ | Use Approx. | Num. Valid ($\uparrow$) | Num. Novel ($\uparrow$) | Novel Ring Count Mean ($\uparrow$) |
|---|---|---|---|---|
| 1 | False | 797.4 $\pm$ 9.91 | 279.0 $\pm$ 23.37 | 4.12 $\pm$ 0.04 |
| 1 | True | 978.4 $\pm$ 3.91 | 166.0 $\pm$ 11.81 | 2.49 $\pm$ 0.07 |
| 2 | False | 829.4 $\pm$ 11.59 | 336.4 $\pm$ 10.55 | 4.54 $\pm$ 0.03 |
| 2 | True | 892.2 $\pm$ 12.19 | 171.2 $\pm$ 13.59 | 3.09 $\pm$ 0.08 |
| 3 | False | 862.6 $\pm$ 7.73 | 363.8 $\pm$ 12.76 | 4.70 $\pm$ 0.03 |
| 3 | True | 763.8 $\pm$ 11.52 | 198.8 $\pm$ 10.76 | 3.76 $\pm$ 0.09 |
| 4 | False | 880.6 $\pm$ 19.15 | 393.0 $\pm$ 12.39 | 4.74 $\pm$ 0.08 |
| 4 | True | 705.8 $\pm$ 14.69 | 215.2 $\pm$ 15.06 | 4.30 $\pm$ 0.14 |
| 5 | False | 889.2 $\pm$ 12.4 | 404.0 $\pm$ 9.64 | 4.76 $\pm$ 0.03 |
| 5 | True | 726.8 $\pm$ 15.64 | 245.8 $\pm$ 6.83 | 4.47 $\pm$ 0.07 |
| 6 | False | 898.6 $\pm$ 21.13 | 427.6 $\pm$ 13.81 | 4.79 $\pm$ 0.06 |
| 6 | True | 757.6 $\pm$ 10.38 | 277.6 $\pm$ 8.73 | 4.55 $\pm$ 0.04 |
| 7 | False | 898.2 $\pm$ 9.26 | 436.4 $\pm$ 8.79 | 4.80 $\pm$ 0.06 |
| 7 | True | 779.4 $\pm$ 17.29 | 295.6 $\pm$ 22.18 | 4.62 $\pm$ 0.06 |
| **8** | **False** | **897.2** $\pm$ **11.58** | **432.0** $\pm$ **19.07** | **4.84** $\pm$ **0.02** |
| 8 | True | 796.6 $\pm$ 15.84 | 304.4 $\pm$ 10.01 | 4.66 $\pm$ 0.06 |
| 9 | False | 903.2 $\pm$ 3.96 | 445.8 $\pm$ 13.14 | 4.83 $\pm$ 0.05 |
| 9 | True | 811.8 $\pm$ 7.82 | 328.4 $\pm$ 7.5 | 4.66 $\pm$ 0.02 |
| 10 | False | 891.6 $\pm$ 12.18 | 431.4 $\pm$ 11.46 | 4.76 $\pm$ 0.05 |
| 10 | True | 816.6 $\pm$ 15.14 | 359.8 $\pm$ 21.87 | 4.70 $\pm$ 0.05 |

Table 33: Varying $n$ (the number of Langevin steps), $\eta$ (the step size), and $\gamma_{kl}$ (the stability regularization coefficient) for maximizing ring count guidance with UDLM NOS. Validity, novelty, and mean ring count for novel sequences are measured for generated sequences from each method. Mean $\pm$ standard deviation reported from repeated generation of 1,024 sequences using five different random seeds. The setting reported used in the main paper is **bolded**.

| $n$ | $\eta$ | $\gamma_{kl}$ | Num. Valid ($\uparrow$) | Num. Novel ($\uparrow$) | Novel Ring Count Mean ($\uparrow$) |
|---|---|---|---|---|---|
| 1 | 0.001 | 0.000 | $1000.8 \pm 2.77$ | $146.2 \pm 7.26$ | $2.10 \pm 0.08$ |
| 1 | 0.001 | 0.001 | $1000.8 \pm 2.77$ | $146.2 \pm 7.26$ | $2.10 \pm 0.08$ |
| 1 | 0.001 | 0.010 | $1000.8 \pm 2.77$ | $146.2 \pm 7.26$ | $2.10 \pm 0.08$ |
| 1 | 0.010 | 0.000 | $1001.0 \pm 2.45$ | $147.2 \pm 8.44$ | $2.10 \pm 0.09$ |
| 1 | 0.010 | 0.001 | $1000.8 \pm 2.17$ | $147.4 \pm 8.56$ | $2.10 \pm 0.09$ |
| 1 | 0.010 | 0.010 | $1001.0 \pm 2.45$ | $147.2 \pm 8.44$ | $2.10 \pm 0.09$ |
| 1 | 0.100 | 0.000 | $1000.8 \pm 4.92$ | $148.2 \pm 7.66$ | $2.13 \pm 0.09$ |
| 1 | 0.100 | 0.001 | $1000.8 \pm 4.92$ | $148.2 \pm 7.66$ | $2.13 \pm 0.09$ |
| 1 | 0.100 | 0.010 | $1000.8 \pm 4.92$ | $148.2 \pm 7.66$ | $2.13 \pm 0.09$ |
| 1 | 1.000 | 0.000 | $988.6 \pm 5.5$ | $161.6 \pm 10.74$ | $2.47 \pm 0.11$ |
| 1 | 1.000 | 0.001 | $988.6 \pm 5.5$ | $161.6 \pm 10.74$ | $2.47 \pm 0.11$ |
| 1 | 1.000 | 0.010 | $988.6 \pm 5.5$ | $161.6 \pm 10.74$ | $2.47 \pm 0.11$ |
| 1 | 5.000 | 0.000 | $323.6 \pm 15.18$ | $109.4 \pm 12.24$ | $3.41 \pm 0.09$ |
| 1 | 5.000 | 0.001 | $323.6 \pm 15.18$ | $109.4 \pm 12.24$ | $3.41 \pm 0.09$ |
| 1 | 5.000 | 0.010 | $323.6 \pm 15.18$ | $109.4 \pm 12.24$ | $3.41 \pm 0.09$ |
| 5 | 0.001 | 0.000 | $1000.8 \pm 2.77$ | $146.6 \pm 7.5$ | $2.10 \pm 0.08$ |
| 5 | 0.001 | 0.001 | $1000.8 \pm 2.77$ | $146.6 \pm 7.5$ | $2.10 \pm 0.08$ |
| 5 | 0.001 | 0.010 | $1000.8 \pm 2.77$ | $146.6 \pm 7.5$ | $2.10 \pm 0.08$ |
| 5 | 0.010 | 0.000 | $1000.8 \pm 2.86$ | $148.2 \pm 6.94$ | $2.11 \pm 0.10$ |
| 5 | 0.010 | 0.001 | $1000.8 \pm 2.86$ | $148.0 \pm 7.04$ | $2.11 \pm 0.10$ |
| 5 | 0.010 | 0.010 | $1000.8 \pm 2.86$ | $147.8 \pm 7.4$ | $2.12 \pm 0.10$ |
| 5 | 0.100 | 0.000 | $1001.0 \pm 4.24$ | $149.0 \pm 5.29$ | $2.14 \pm 0.13$ |
| 5 | 0.100 | 0.001 | $1001.0 \pm 4.24$ | $149.0 \pm 5.29$ | $2.14 \pm 0.13$ |
| 5 | 0.100 | 0.010 | $1001.0 \pm 4.24$ | $149.0 \pm 6.0$ | $2.14 \pm 0.13$ |
| 5 | 1.000 | 0.000 | $988.6 \pm 5.5$ | $161.6 \pm 10.74$ | $2.47 \pm 0.11$ |
| 5 | 1.000 | 0.001 | $990.2 \pm 5.26$ | $162.2 \pm 9.96$ | $2.46 \pm 0.11$ |
| 5 | 1.000 | 0.010 | $989.4 \pm 4.83$ | $160.4 \pm 7.57$ | $2.44 \pm 0.11$ |
| 5 | 5.000 | 0.000 | $323.6 \pm 15.18$ | $109.4 \pm 12.24$ | $3.41 \pm 0.09$ |
| 5 | 5.000 | 0.001 | $413.0 \pm 16.17$ | $166.8 \pm 18.95$ | $3.92 \pm 0.05$ |
| **5** | **5.000** | **0.010** | $\mathbf{573.8} \pm \mathbf{13.03}$ | $\mathbf{244.4} \pm \mathbf{13.05}$ | $\mathbf{3.96} \pm \mathbf{0.07}$ |
| 10 | 0.001 | 0.000 | $1001.0 \pm 2.74$ | $146.8 \pm 7.33$ | $2.09 \pm 0.09$ |
| 10 | 0.001 | 0.001 | $1001.0 \pm 2.74$ | $146.8 \pm 7.33$ | $2.09 \pm 0.09$ |
| 10 | 0.001 | 0.010 | $1001.0 \pm 2.74$ | $146.8 \pm 7.33$ | $2.09 \pm 0.09$ |
| 10 | 0.010 | 0.000 | $1001.0 \pm 3.08$ | $147.8 \pm 7.46$ | $2.12 \pm 0.10$ |
| 10 | 0.010 | 0.001 | $1001.0 \pm 3.08$ | $147.8 \pm 7.46$ | $2.12 \pm 0.10$ |
| 10 | 0.010 | 0.010 | $1000.6 \pm 3.78$ | $147.2 \pm 6.98$ | $2.12 \pm 0.10$ |
| 10 | 0.100 | 0.000 | $1000.0 \pm 4.74$ | $150.4 \pm 7.09$ | $2.15 \pm 0.13$ |
| 10 | 0.100 | 0.001 | $1000.0 \pm 4.74$ | $151.0 \pm 6.78$ | $2.14 \pm 0.12$ |
| 10 | 0.100 | 0.010 | $1000.0 \pm 4.74$ | $150.6 \pm 6.15$ | $2.15 \pm 0.13$ |
| 10 | 1.000 | 0.000 | $988.6 \pm 5.5$ | $161.6 \pm 10.74$ | $2.47 \pm 0.11$ |
| 10 | 1.000 | 0.001 | $990.6 \pm 5.55$ | $162.0 \pm 9.64$ | $2.45 \pm 0.09$ |
| 10 | 1.000 | 0.010 | $993.0 \pm 3.46$ | $163.6 \pm 9.89$ | $2.47 \pm 0.09$ |
| 10 | 5.000 | 0.000 | $323.6 \pm 15.18$ | $109.4 \pm 12.24$ | $3.41 \pm 0.09$ |
| 10 | 5.000 | 0.001 | $454.6 \pm 12.78$ | $188.8 \pm 12.24$ | $3.96 \pm 0.05$ |
| 10 | 5.000 | 0.010 | $670.0 \pm 6.4$ | $276.8 \pm 20.62$ | $3.81 \pm 0.07$ |

Table 34: Ablation results of $\gamma$ values on CIFAR10. Best values for each model are **bolded**.

|  | FID ($\downarrow$) | IS ($\uparrow$) | F1 ($\uparrow$) |
|---|---|---|---|
| **MDLM D-CFG** | | | |
| $\gamma = 1$ | 27.94 | 7.14 | 0.76 |
| $\gamma = 2$ | 18.62 | 8.24 | 0.95 |
| $\gamma = 3$ | 16.19 | 8.78 | 0.99 |
| $\gamma = 4$ | **15.56** | 9.02 | 0.99 |
| $\gamma = 5$ | 15.73 | **9.19** | **1.00** |
| **UDLM D-CFG** | | | |
| $\gamma = 1$ | 26.70 | 7.43 | 0.81 |
| $\gamma = 2$ | **20.75** | 8.34 | 0.96 |
| $\gamma = 3$ | 21.31 | 8.52 | 0.98 |
| $\gamma = 4$ | 23.21 | **8.66** | **0.99** |
| $\gamma = 5$ | 26.15 | 8.60 | **0.99** |

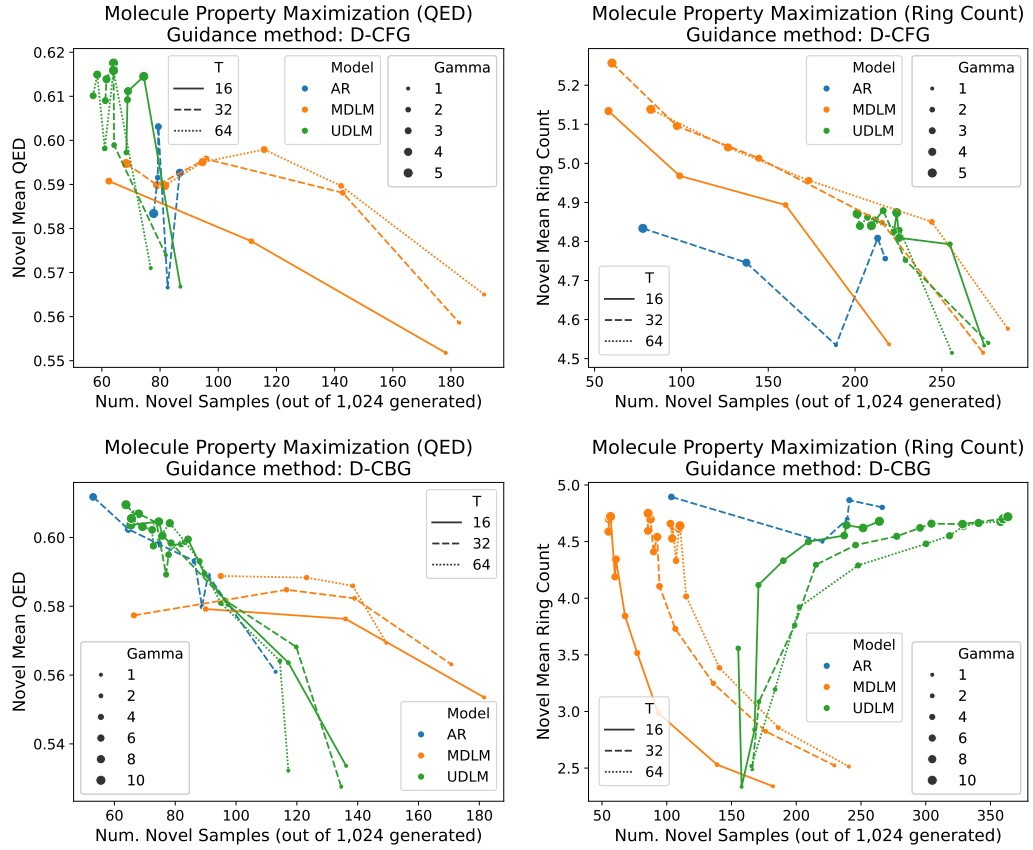

Figure 6: Effect of varying $T$ on QM9 guidance generation. *(Top Left)* Maximizing drug-likeness (QED) using D-CFG. *(Bottom Left)* Maximizing drug-likeness (QED) using D-CBG. *(Top Right)* Maximizing ring count using D-CFG. *(Bottom Right)* Maximizing ring count using D-CBG.

Table 35: Varying $T$ for CIFAR10 conditional generation. 50k images sampled from a conditional model (D-CFG $_{\gamma=1}$). F1 metric from a separate classifier trained to identify the class label used for the conditional generation. Best metric per $T$ is **bolded**.

| Model | FID ($\downarrow$) | IS ($\uparrow$) | F1 ($\uparrow$) |
|---|---|---|---|
| $T = 128$ | | | |
| MDLM | 64.09 | 5.81 | 0.63 |
| UDLM | **30.48** | **7.30** | **0.80** |
| $T = 256$ | | | |
| MDLM | 41.53 | 6.51 | 0.70 |
| UDLM | **28.30** | **7.40** | **0.81** |
| $T = 512$ | | | |
| MDLM | 31.97 | 6.89 | **0.74** |
| UDLM | **27.12** | **7.43** | **0.74** |
| $T = 1024$ | | | |
| MDLM | 27.94 | 7.14 | **0.81** |
| UDLM | **26.70** | **7.43** | **0.81** |

Table 36: Generative Perplexity (Gen. PPL; using GPT-2 Large) for unconditional sample generation from models trained on LM1B. Best value is **bolded**.

| Model | $T$ | Gen. PPL ($\downarrow$) |
|---|---|---|
| AR | 128 | **67.46** |
| MDLM | 16 | 170.64 |
| MDLM | 32 | 140.47 |
| MDLM | 64 | 130.62 |
| MDLM | 128 | 120.93 |
| MDLM | 256 | 119.96 |
| MDLM | 512 | 118.34 |
| MDLM | 1024 | 116.80 |
| UDLM | 16 | 82.09 |
| UDLM | 32 | 79.80 |
| UDLM | 64 | 79.93 |
| UDLM | 128 | 79.87 |
| UDLM | 256 | 79.07 |
| UDLM | 512 | 79.14 |
| UDLM | 1024 | 78.22 |

Table 37: Generated samples for unconditional generation from models trained on LM1B.

| Model - $T$ | Generated Text |
| --- | --- |
| AR - 128 | "[CLS] to move the global economy away from inappropriate trade practices and build an environment that rewards innovation. i. e. samples of s̈eed money by jim mellon, paul pompa jr., gil familier, vanessa thomas, george rothman, george whitacre, john ziemba, mark sweitzer, little george nowi, thayer evans and allison gallagher. i. e. yang if you don't wanna fly, the statistics means you are going to make your rocket arm - rolls. [CLS] savings that made the difference in someone's life cannot be made in another country without foreign assistance on their [CLS]" |
| MDLM - 16 | "[CLS] mezzo ev of dream spirits. [CLS] to naval cadets i didn't want mum and dad to be a group, but even with the odd social butterflies i felt myself sitting at the edge of a deep blue pool. [CLS] on wednesday morning, fighter jets billoured the area during the early hours of the morning. [CLS] i know of the sailing course, the scenery, the coastline, the whole thing very beautiful - - the land of water. [CLS] wednesday, after its peak height reached the ice's lowest point. [CLS] reader's thought is. for the basic requirements of team leadership, internship, part time, a..." |
| MDLM - 1024 | "[CLS] was killed as mr. giffords and as the caregiver, representative giffords's. [CLS] but i have no doubt that this process will produce meat, m̈s. rommel said. [CLS] and even if you don't need to hit an agency envelope during it, you should pay the rent through office expenses, r̈eferring to the possible error. [CLS] blog. view billshrift and discuss yahoo commercials, and what? [CLS] this month, the new logos of cin - armarge became scranton neck - based riverton and tollhorn. [CLS] when i arrived, he'd ducking behind the woman.." |
| UDLM - 16 | "[CLS] grandeur. [CLS] grand avenue an. [CLS] it upon request, such information plays only in the p̈rovision of the protocol äs guidelines for the nonprofit sector. [CLS] that too has to worry people about schools that shut down and closed schools for several weeks. [CLS] ï will not comment on anything important to the eu, where someone is blaming serbia to not a figurehead. [CLS] but i think that because an increasing number of coal is put at risk as a displaced gateway, and the other party of technology falls away from communities that can somehow show up as a starting point, one thing that the province will suffer is loss of almost [CLS]" |
| UDLM - 1024 | "[CLS] would [CLS] a 49 percent stake in the company under the terms of the deal. [CLS] the only subsequent trial out of ms, sykes is 19 february next year after prosecutors failed to overturn the death sentence against him. [CLS] but dugard has recorded several rings for baby jesus, a completely new thing as for calling muslim cold - callers and working with his confiscators. [CLS] i am forced to seek out convenient explanations while remaining in no position either to caucus. [CLS] of course, get me out of there. [CLS] the first thing being wrote about is having available food. [CLS] sir simon's guests included singer hayley wood and" |

