# OpenReview forum: "Simple Guidance Mechanisms for Discrete Diffusion Models"
_ICLR.cc/2025/Conference — ICLR 2025 Poster_

### Official Review · Reviewer_S2DB · 2024-11-01

**Soundness:** 2
**Presentation:** 3
**Contribution:** 1
**Rating:** 3
**Confidence:** 5

**Summary:**

This paper proposes to introduce uniform noising paths for discrete diffusion models. The paper derives an explicit formulation for the variational lower bound under a uniform prior. The paper further introduces methods to perform classifier-free and classifier-based guidance in this discrete diffusion setting. Experiments are conducted in regular text, images, genome, and molecular settings.

**Strengths:**

The paper is generally well written with the development of ideas following a logical narrative. The experimental settings considered are also diverse in terms of various discrete domains. Beyond this, it is difficult to pinpoint further clear strengths.

**Weaknesses:**

I appreciate the authors for trying to crisply write their simple uniform diffusion model framework but I have several critical concerns regarding various aspects of this paper that I will outline below.

**Novelty**

I very respectfully disagree with the author's contributions. The uniform prior and the corresponding are widely known and exist in the literature. In fact, I challenge the authors to demonstrate how their formulation differs from Discrete Flow Matching (Gat et. al 2024), and Dirichlet Flow/Diffusion (Stark et. al). There are other papers as well, but very explicitly your objective function in your variational lower bound can be shown $\mathbb{E}[-\log p(x_0 | x_t)]$ where $x_t \sim p_t(x)$ is the uniform path. This analysis is a direct application of the MDLM setup for the uniform path. I am open to being wrong here, but I do not see anything novel here despite equation 10 suggesting there is more going on than there is. Also, I believe the first-order Taylor approximation to guidance was already introduced in
(Nisonoff et. al 2024) and the adaptation in this paper is very respectfully a minor change.


**Technical Limitations**

Regarding the new classifier/classifier-free guidance formulation the fact that we need to compute the normalization constant means that this cannot be extended to actual larger-scale systems where we would want to apply discrete diffusion models as opposed to autoregressive ones. I have severe concerns regarding the usefulness of the claim that the proposed method works well on small vocabularies, this to me is a sign that this approach is fundamentally limited. Note that Dirichlet Flow Matching (Stark et. al 2024) point out this exact issue with a uniform prior paths, and hence suggest their Dirichlet paths.

**Experimental Concerns**

Regarding the experimental setup I have a few concerns regarding the results presented. I find the omission of MDLM for the LM1B table 3 to be awkward. Looking at Table 1 in the MDLM paper we notice that their test perplexity upper bound is 27.04 which is better than the UDLM. In fact, I really do not understand the need to report Table 1 and 3 for the same dataset, it feels contrived to change the vocabulary size. It is clear that uniform doesn't work in this setting and I bet playing with noise schedules and other small details would close the gap for MDLM and UDLM in the smaller scientific settings. I also found it surprising the Generative Perplexity under GPT-2 was not reported as MDLM does. Finally, there are no textual generated samples as done for even CIFAR 10. This is a strange omission, we should be able to visually inspect the difference in suspected performance.

**References**

Stark, Hannes, et al. "Dirichlet flow matching with applications to dna sequence design." arXiv preprint arXiv:2402.05841 (2024).

Gat, Itai, et al. "Discrete flow matching." arXiv preprint arXiv:2407.15595 (2024).

Nisonoff, Hunter, et al. "Unlocking Guidance for Discrete State-Space Diffusion and Flow Models." arXiv preprint arXiv:2406.01572 (2024).

**Questions:**

N/A

---

> ### Author Response · Authors · 2024-11-20
> **Response to Reviewer S2DB**
>
> We thank the reviewer for their detailed comments. Below we respond to the specific concerns raised in detail, the high level responses are as follows:
> - **Comparison to Flow Matching works:** UDLM and Gat et al. use different objectives (theirs is not an ELBO) and samplers; Stark et al. is significantly different from our work: it models continuous-states not discrete-states
> - **Comparison to Nisonoff et al.:** Our guidance formula is substantially different from Nisonoff et al. (see below); both use the Taylor trick from Vignac et al. 2022
> - We report experiments on large vocabularies; we also report new experiments where novelty is a concern (e.g., vs. Gat et al.)

---

> ### Author Response · Authors · 2024-11-20
> **Concern 1a: Novelty of UDLM**
>
> In the table below, we summarize the key differences between UDLM and other works that examine discrete corruption with uniform noise.
>
> | Method | Parameterization | Loss | Noise Process |
> |-------------------|-------------------------|--------|---------------------|
> | UDLM | $p_\theta(\mathbf{x}_0 \mid \mathbf{z}_t)$ (“mean”) | Simplified and tight continuous-time ELBO  | Interpolation between data and limiting distribution $Cat(\mathbf{z}_t \mid \alpha \mathbf{x}_0 + (1-\alpha)\boldsymbol{\pi})$ |
> | D3PM Uniform | $p_\theta(\mathbf{x}_0 \mid \mathbf{z}_t)$ (“mean”) | Discrete time ELBO |  Multiplication by state transition matrices $Q_1 \cdot … \cdot Q_t$ |
> | SEDD Uniform | $s_\theta(\mathbf{x})_{\mathbf{y},t} \approx q_t(\mathbf{y})/q_t(\mathbf{x})$ | Score-based $ELBO(s_\theta)$ | Corruption via rate matrices $\mathbf{x}_0 \cdot \int_0^t \exp(R_t)$ |
> | Gat et al. 2024 | $p_\theta(\mathbf{x}_0 \mid \mathbf{z}_t)$ (“mean”) | Cross-entropy $\log p_\theta(\mathbf{x}_0 \mid \mathbf{z}_t)$ | Interpolation between data and limiting distribution |
>
>
> **Comparison to Gat et. al**
>
> The main difference is the training objective: we use a principled ELBO, while Gat et al. simply train a BERT-style denoiser. Our variational bound is novel, tight, and performant (see Table 2 and 3). Being probabilistic enables applying UDLM to language modeling, which involves a principled comparison across models (e.g., AR transformers) using perplexity. There is no comparison in Gat et al. to baselines using true perplexity (or its lower bound); they only report generative perplexity.
>
> To underscore the difference, we re-train our model with uniform noise using the simple cross-entropy loss used in Gat et al. and find that training with the UDLM objective leads to better language modeling performance (experiment below reflects the QM9 and Species10 datasets):
>
> | Dataset      | Training Objective                     | Validation ELBO |
> |----------------|-------------------------------------------|-----------------------|
> | QM9           | Cross-entropy (as in Gat et al.) |                   2.99 |
> | QM9           | ELBO (as in our work)               |             **2.02** |
> | Species10  | Cross-entropy (as in Gat et al.) |                   5.20 |
> | Species10  | ELBO (as in our work)               |             **3.15** |
>
> Our method also uses a different sampler, although in theory one could extract a rate matrix from our model and use it with Gat’s sampler (and vice versa).
>
> **Comparison to Stark et. al**
>
> We emphasize that Stark et al. 2024 defines a flow over **continuous** representations of the data. This is distinct from our approach, which directly defines the corruption and denoising processes on the discrete data representations. Another difference is model parameterization: velocity in Stark et al. vs. mean parameterization in our work.
>
> Stark et al. 2024 can be seen as an analogous work to those that perform continuous diffusion for discrete data, e.g. Han et al. 2022, but for flow models. Of note, Stark et al. 2024 also explore classifier-free and classifier-based guidance for flows but rely on the fact that the velocity fields they learn reflect the scores obtained in continuous diffusion modeling. Thus the guidance methods of Stark et al. 2024 are more similar to the previous definitions of classifier-free and classifier-based guidance from the continuous diffusion literature.
>
> **_“Uniform prior is widely known and exists in the literature”_**
>
> We clarify that our claim is not to have developed discrete diffusion with uniform noise, but rather the novelty comes from: (1) the optimized continuous time ELBO with which we train UDLM (Section 3.2); (2) the application of uniform noise diffusion to controllable generation.
>
> We stress that the novelty of Equations 10 and 11 is significant in that it tightens the ELBO by taking $T \rightarrow \infty$ and allows us to avoid computing values that can be analytically set to zero (in terms of $\mathcal{L}\_{prior}$ and $\mathcal{L}\_{recons}$). The importance of this improved objective is reflected in our language modeling results, which show that **UDLM outperforms previous uniform-noise discrete data methods** (as shown in the bottom 3 rows of Tables 2 and 3, which we reproduce below).
>
> | Dataset   | D3PM Uniform (Austin et. al 2021) | SEDD Uniform (Lou et al. 2023) | UDLM (_Ours_) |
> |---------------------------- |----------------------------------------|-------------------------------------|----------------------|
> | text8 (BPC; $\downarrow$) | 1.61 | 1.47 | **1.44** |
> | LM1B (PPL; $\downarrow$) | 77.50 | 40.25 | **35.35** |

---

> ### Author Response · Authors · 2024-11-20
> **Concern 1b: Novelty of Guidance Methods**
>
> **Guidance methodology vs. Nisonoff et al. 2024**
>
> First, Nisonoff et al. 2024 is an unpublished manuscript that is concurrent with ours.
>
> Second, the method of Nisonoff et al. 2024 is different from ours. The guidance formula is different, see table below. In particular, the method of Nisonoff et al. 2024 requires sophisticated CTMC theory and several pages of mathematical derivation. In contrast, our guidance mechanisms involve only elementary probability and can be defined in under a page; however, they outperform all published baselines (Diffusion+NOS, AR+FUDGE etc.).
>
> |                            | Nisonoff et al.   | Ours |
> |-----------------------|----------------------|--------|
> | Classifier-free     | $R_{\theta, t}(\mathbf{z}’, \mathbf{z} \mid y)^{\gamma}R_{\theta, t}(\mathbf{z}’, \mathbf{z})^{1-\gamma}$ | $p_\theta(\mathbf{z}\_{s} \mid \mathbf{z}\_t, y)^{\gamma}  p_\theta(\mathbf{z}\_s \mid \mathbf{z}\_t)^{1-\gamma}$ |
> | Classifier-based | $\Bigg[\frac{p_\phi(y\mid \mathbf{z}’)}{p_\phi(y \mid \mathbf{z})}\Bigg]^{\gamma}R_{\theta, t}(\mathbf{z}’, \mathbf{z})$ |  $p_\phi(y \mid \mathbf{z}\_{s}, \mathbf{z}\_{t})^\gamma p_\theta(\mathbf{z}\_s \mid \mathbf{z}\_t)$ |
>
> Finally, please note that the Taylor approximation is not core of either our method nor that of Nisonoff et al. In fact, both methods cite earlier usage in Vignac et al. 2022. In both methods, it circumvents performing $L\cdot N$ passes through the classifier model, where $L$ is the sequence length and $N$ is the vocabulary size.

---

> ### Author Response · Authors · 2024-11-20
> **Concern 2: Guidance is limited to small vocabularies**
>
> We address the reviewer’s concern about performance on large vocabularies with classifier-free experiments on NLP datasets which use the `bert-base-uncased` tokenizer (vocab. size = 30,522).
>
> **Classifier-free Guidance: Part of speech generation**
>
> We find that **D-CFG with Diffusion models (MDLM and UDLM) is able to outperform AR models with large vocabularies** with comparable sample quality (generative perplexity) and much higher control scores (F1).
>
> Additionally, we see in the Table below, that **only UDLM seems to respond to higher $\gamma$ tuning compared to MDLM and AR**:
>
>
> | Model  | Guidance                        | Gen. PPL ($\downarrow$) | F1 ($\uparrow$) |
> |-----------|---------------------------------|-------------------------------------|----------------------|
> | AR       | D-CFG ($\gamma = 1$) | **9.80**                                    | 19 |
> | AR       | D-CFG ($\gamma = 2$) | 10.18                                    | 20 |
> | AR       | D-CFG ($\gamma = 4$) | 10.32                                    | 19 |
> | MDLM | D-CFG ($\gamma = 1$) | 9.92                                    | 85 |
> | MDLM | D-CFG ($\gamma = 2$) | 10.73                                    | 85 |
> | MDLM | D-CFG ($\gamma = 4$) | 13.83                                   | 78 |
> | UDLM | D-CFG ($\gamma = 1$) | 10.23                                    | 83 |
> | UDLM | D-CFG ($\gamma = 2$) | 10.34                                    | 85 |
> | UDLM | D-CFG ($\gamma = 4$) | 10.82                                    | **86** |
>
> The normalization in our discrete guidance methods is not a computational bottleneck. Rather it amounts to simply applying the `.softmax(dim=-1)` to the un-normalized log probabilities (along the vocabulary dimension) and is therefore not prohibitive, even in large vocabulary regimes.
>
> **Details about this experiment:** We train models on the E2E dataset, which consists of 50k restaurant reviews (Novikova et al. 2017). The goal is to generate sentences that follow a part-of-speech template, which acts as the guidance label. Quality is measured by generative perplexity and controllability is measured as the F1 score between the POS tagging of the generated sequences and that provided as the control target.

---

> > ### Author Response · Authors · 2024-11-20
> > **Concern 3a: Experimental concern: “It is clear that uniform doesn't work in this setting”**
> >
> > As demonstrated in the part-of-speech guidance experiment above, UDLM can indeed be useful for modeling natural language text, and, as we show in Table 1, there are domains where UDLM is competitive, if not superior, to MDLM. One of the main motivations behind UDLM was that it can improve guidance in settings where MDLM “locks in” certain predictions that it cannot change. In the experiment, below we make this clear, by examining generation on the CIFAR-10 dataset with smaller $T$ where the diffusion models predict multiple pixels in parallel. MDLM’s performance deteriorates with smaller $T$, whereas UDLM is robust to this setting:
> >
> > | $T$   | Model  | FID ($\downarrow$) | IS ($\uparrow$) | F1 ($\uparrow$) |
> > |--------|-----------|----------------------------|----------------------|------------------------|
> > | 128   | MDLM |                        64.09 |                  5.81 |                     0.63 |
> > | 128   | UDLM |                        30.48 |                  7.30 |                     0.80 |
> > | 256   | MDLM |                        41.53 |                  6.51 |                     0.70 |
> > | 256   | UDLM |                        28.30 |                  7.40 |                     0.81 |
> > | 512   | MDLM |                        31.97 |                  6.89 |                     0.74 |
> > | 512   | UDLM |                        27.12 |                  7.43 |                     0.74 |
> > | 1000 | MDLM |                        27.94 |                  7.14 |                     0.81 |
> > | 1000 | UDLM |                        26.70 |                  7.43 |                     0.81 |
> >
> > **Details about this experiment:** We generate 50k images sampled from a conditional model (which is equivalent to using $\gamma=1$ and D-CFG). F1 is the accuracy of a separate classifier in identifying the class label used for the conditional generation.

---

> > > ### Author Response · Authors · 2024-11-20
> > > **Concern 3b: Why are results for the same datasets reported in  Table 1 and Tables 2/3?**
> > >
> > > **Purpose of Table 1**
> > >
> > > Table 1 compares:
> > > - AR (state-of-the-art language modeling parameterization) to
> > > - MDLM (current state-of-the-art discrete diffusion model) to
> > > - UDLM.
> > >
> > > The goal of this table is to dispel the widely held belief (see for example Austin et. al 2021 and Lou et. al 2023) that uniform noise discrete diffusion is a worse paradigm than absorbing state. Our findings show that this is more nuanced: for certain small vocabulary datasets, UDLM is competitive with MDLM, both of which perform well relative to the dominant AR approach.
> > >
> > > **Purpose of Tables 2 & 3**
> > >
> > > These latter tables demonstrate how training with our refined ELBO in UDLM improves performance over other uniform discrete diffusion models. We see that **among uniform noise discrete diffusion models, UDLM performs best on both text8 and LM1b datasets.** In addition to comparing to D3PM uniform and SEDD uniform, in these tables we also provide other reported models so that we can better situate our work against prior models from the literature.
> > >
> > > To address this comment:
> > >
> > >
> > > > “...it feels contrived to change the vocabulary size"
> > >
> > > we want to clarify and emphasize that the vocabulary sizes reported in Table 1 and Table 3 (for LM1B) are identical, **we did not change this value** (the same point holds true for the text8 dataset in Table 1 and Table 2). The vocabulary size is determined by the tokenizer used, and we chose the standard one used by previous works for these datasets, i.e., `bert-base-uncased` for LM1B and character-level for text8.
> > >
> > > The omission of MDLM from Table 3 was not intentional, this was an oversight that we fix in the updated manuscript. Thank you for bringing this to our attention.
> > >
> > > In terms of the discrepancy to the reported MDLM number ppl (27.04): the value we report is from a model we re-trained (not taken from Sahoo et. al 2024). The discrepancy comes from the fact that we pre-processed the data differently:
> > > - In our work, text is **wrapped** which means we pack sequences to fill the window size of 128 tokens.
> > > - In Sahoo et. al 2024, text was padded, which means each item in the batch corresponds to a single sequence from the LM1B corpus, and any sequences less than 128 tokens are right-filled with the special `[PAD]` token.
> > >
> > > (We are currently re-running our UDLM experiment for LM1B using **padded** sequences to better compare to previous work. We will update here as soon as results are available.)
> > >
> > > Finally, we note that there was no hyperparameter or noise schedule tuning performed whatsoever. For all experiments, the exact same setup reported in Sahoo et al. 2024 for MDLM was reproduced for UDLM.

---

> > > > ### Author Response · Authors · 2024-11-20
> > > > **Concern 3c: Report generative perplexity and provide text samples**
> > > >
> > > > We thank the reviewer for this suggestion. Below we provide generative perplexity (under the GPT-2 Large model) for 1,024 sequences generated by AR, MDLM, and UDLM models that were trained on the Amazon Polarity dataset.
> > > >
> > > > | Model | Gen. PPL ($\downarrow$) |
> > > > |----------|------------------------------------|
> > > > | AR     | 51.60 |
> > > > | MDLM | 64.00 |
> > > > | UDLM | 91.79 |
> > > >
> > > > We also provide two random samples of generation from each model in the table below.
> > > >
> > > > | Model | Sample Generation |
> > > > |----------|------------------------------------|
> > > > | AR     | "bought this toy 5 years ago for my son from toys r us. it traded for the igloo tambourine for his brand new surfdoodle which sits in the garage with it on it's other flying object. same holds - up control and durability as the original. anyway, my son has now had his personal barbie / ariel for 2 years now and is still entertained with it. if you parents out there are trying to buy twins, make sure they are going to be willing to play with their barbie toy. the shark is made for children to have some fun and they love playing with it on their" |
> > > > | AR     | "we found this series of books on cd for our son to be extremely enjoyable for his aged eyes. the type was clear in printed form and the narrator was pleasant and recognizable. all the books were a great value and he seemed really enjoying them." |
> > > > | MDLM | “this device is just as described. the only drawback that i see is that it is not made for guys. the nuph is relatively heavy and it cannot withstand the weight of the device enough to compliment it. it keeps my attention." |
> > > > | MDLM | “this book was horrid! it's typical for a churchbook enhanced up with a plot that hasn't been told. one of the worst booksi have ever had to listen to.”
> > > > | UDLM | “i work with several holocaust story libraries over the years and found this truely lightweight. the story and hints of the olyndics are not fully developed throughout the book. a for one, they kill other people and the night shall be found with al guinion. i like the idea of bible and it is quite engaging, but it does not fully compensate for the narrative. one notable is that it's rambling because god did forgive himself, but the book is not explained. i did finish the book but” |
> > > > | UDLM | “i've been using this version on my ancient windows ibook. it installs pretty easy with good documentation and knowledge problems. the manual is unfortunately useless. poor graphics in margins. they have no files where you type more closely than text and speak english. ive since got them but they are far harder. they need tweak them by eyeball. as if they will look all of your squares at their non - clarity so you actually understand what they were talking about. that means it is an” |

---

> > > > > ### Author Response · Authors · 2024-11-20
> > > > > **References**
> > > > >
> > > > > Austin, Jacob, et al. "Structured denoising diffusion models in discrete state-spaces." Advances in Neural Information Processing Systems 34 (2021): 17981-17993.
> > > > >
> > > > > Han, Xiaochuang, Sachin Kumar, and Yulia Tsvetkov. "Ssd-lm: Semi-autoregressive simplex-based diffusion language model for text generation and modular control." arXiv preprint arXiv:2210.17432 (2022).
> > > > >
> > > > > Lou, Aaron, Chenlin Meng, and Stefano Ermon. "Discrete Diffusion Modeling by Estimating the Ratios of the Data Distribution." Forty-first International Conference on Machine Learning.
> > > > >
> > > > > Nisonoff, Hunter, et al. "Unlocking Guidance for Discrete State-Space Diffusion and Flow Models." arXiv preprint arXiv:2406.01572 (2024).
> > > > >
> > > > > Novikova, Jekaterina, Ondřej Dušek, and Verena Rieser. "The E2E dataset: New challenges for end-to-end generation." arXiv preprint arXiv:1706.09254 (2017).
> > > > >
> > > > > Sahoo, Subham Sekhar, et al. "Simple and Effective Masked Diffusion Language Models." arXiv preprint arXiv:2406.07524 (2024).
> > > > >
> > > > > Shi, Jiaxin, et al. "Simplified and Generalized Masked Diffusion for Discrete Data." arXiv preprint arXiv:2406.04329 (2024).
> > > > >
> > > > > Stark, Hannes, et al. "Dirichlet flow matching with applications to dna sequence design." arXiv preprint arXiv:2402.05841 (2024).

---

> > > > > > ### Comment · Reviewer_S2DB · 2024-11-24
> > > > > > **Re: Rebuttal**
> > > > > >
> > > > > > I thank the authors for their time and efforts in addressing the main concerns raised in my initial review. I especially appreciate the authors trying additional experiments and reporting some of the metrics I advocated for.
> > > > > >
> > > > > > Having reread the paper in detail, and looked at the rebuttal experiments I would like to maintain my initial review and I still remain unconvinced by both the novelty between Gat et al 2024 and the actual utility of UDLM in practice. I am sorry, but I do not think this is sufficient to change my rating of 3.

---

> > > > > > > ### Author Response · Authors · 2024-11-25
> > > > > > > **Response regarding Gat et al.**
> > > > > > >
> > > > > > > Thank you for engaging with our response.
> > > > > > >
> > > > > > > We first note that according to the [ICLR 2025 guidelines](https://iclr.cc/Conferences/2025/ReviewerGuide), Gat et al. is concurrent work, having only appeared on arxiv July 22, 2024 and set to be published at NeurIPS, which is after the ICLR submission deadline.
> > > > > > >
> > > > > > > Moreover, we respectfully disagree with the reviewer’s assessment. There are several points that differentiate our work from Gat et al:
> > > > > > > - The guidance mechanisms and our experimental results investigating how diffusion performs vs. AR is novel relative to this work. **There is no discussion of guidance mechanisms in Gat et al.**
> > > > > > > - While Gat’s formulation can accommodate uniform noise, **they focus exclusively on masking**. To the best of our understanding, all of their experimental results use absorbing state as the limiting distribution.
> > > > > > > - Gat et al. takes a flow-based approach and does not train with a principled ELBO. The result we ran (copied again below) clearly demonstrate that it is important to optimize the true ELBO as it leads to better language modeling (as measured by actual PPL lower bound):
> > > > > > >
> > > > > > > | Dataset      | Training Objective                     | Validation ELBO |
> > > > > > > |----------------|-------------------------------------------|-----------------------|
> > > > > > > | QM9           | Cross-entropy (as in Gat et al.) |                   2.99 |
> > > > > > > | QM9           | ELBO (as in our work)               |             **2.02** |
> > > > > > > | Species10  | Cross-entropy (as in Gat et al.) |                   5.20 |
> > > > > > > | Species10  | ELBO (as in our work)               |             **3.15** |
> > > > > > >
> > > > > > > - Previous work has all but written off uniform noise. Our work is a clear refutation of this and the results are clearly distinct from Gat et al. There are settings where uniform noise is useful and in fact preferable to absorbing state models:
> > > > > > >     - The CIFAR 10 example shows that with limited inference budget, UDLM is significantly better than MDLM
> > > > > > >     - In scientific applications, where vocab size is typically smaller, UDLM is better with similar/better language modeling performance and the added benefit of being more guidable.
> > > > > > >     - We also demonstrate in the new E2E experiment that even in large vocabulary regimes, UDLM is effective relative to MDLM.

---

> ### Comment · Reviewer_S2DB · 2024-11-25
> **Re: DFM**
>
> Dear authors,
>
> I appreciate your response to my rebuttal response. I wish to continue to respond in good faith, and I hope the authors will understand my issue with the suggestion that Gat et al. 2024 is a concurrent work.
>
> Very respectfully, this claim is not warranted. MDLM, MD4, and DFM are largely the same paper disguised under slightly different notation and under the guise of Flow Matching vs. Discrete Diffusion. In the discrete setting (discrete state space) there is **no difference** between MDLM and DFM. The end objective is a cross-entropy based denoiser---they lead to the same ELBO optimization for the masked diffusion case. The main technical difference in DFM is the path construction, which allows uniform paths. Given that you compare with MDLM which will also be published at NeurIPS 2024 this year, I maintain insufficient novelty between this work and DFM.
>
> If you truly wish to convince me that uniform paths have substantial value then I would expect a language modeling benchmark of size LM1B or larger where UDLM clearly outperforms MDLM and DFM. Alternatively, scale to a larger image benchmark e.g. imagenet 128x128, and show UDLM beats MDLM. A skim through the rebuttal experiment findings does not suggest a better Gen PPL or a better PPL upper bound. Maybe I missed your result, if so please point me to it.
>
> If either of these results are obtained, I shall happily increase my score to 6. I sincerely hope you do succeed, because I would love to update my, perhaps insufficient, understanding of UDLMs.

---

> ### Author Response · Authors · 2024-11-25
> **UDLM outperforms MDLM on LM1B Gen PPL (1/2)**
>
> For the language models we trained on LM1B, below we present Gen PPL, which demonstrate that UDLM  outperforms MDLM **especially for smaller $T$** where the value of UDLM is most apparent (since it can recover from mistakes).
>
> | Model  |  $T$            | Gen PPL (under GPT-2 Large) |
> |-----------|-----------------|------------------------------------------|
> | AR       | 128             | 67.46                                         |
> | MDLM | 16               | 170.64                                       |
> | MDLM | 32               | 140.47                                       |
> | MDLM | 64               | 130.62                                       |
> | MDLM | 128             | 120.93                                       |
> | MDLM | 256             | 119.96                                       |
> | MDLM | 512             | 118.34                                       |
> | MDLM | 1024           | 116.80                                       |
> | UDLM | 16               | 82.09                                         |
> | UDLM | 32               | 79.80                                         |
> | UDLM | 64               | 79.93                                         |
> | UDLM | 128             | 79.87                                         |
> | UDLM | 256             | 79.07                                         |
> | UDLM | 512             | 79.14                                         |
> | UDLM | 1024           | 78.22                                         |
>
> These results are significant because one of the key potential advantages of diffusion vs. AR is parallel sampling and the reduction of forward passes through the model by setting $T$ < $L$ (the sequence length). In this setting we see that UDLM significantly outperforms MDLM.
>
> (Note these outputs come from the models we re-trained where the LM1B sequences were _wrapped_ as opposed to pad filled)
>
> ---
>
> In terms of comparing our work to MDLM/MD4/DFM more broadly, it is important to note none of these works discuss adapting guidance to discrete diffusion. Our work is novel in this respect. Moreover, these works focus only on absorbing state diffusion. Our work investigates uniform noise and for the first time finds a training objective that renders this model competitive with the absorbing state counterparts, and we also find settings where UDLM outperforms these previous works, such as the LM1B results just shared, the CIFAR10 results previously discussed, and the extensive guidance experiments we present in our work.

---

> > ### Author Response · Authors · 2024-11-25
> > **UDLM outperforms MDLM on LM1B Gen PPL (2/2)**
> >
> > Below we also present some generated samples.
> >
> > | Model - $T$ | Generated Text |
> > |------------------|------------------------------------------|
> > | AR - 128        | _"[CLS] to move the global economy away from inappropriate trade practices and build an environment that rewards innovation. i. e. samples of \" seed money \" by jim mellon, paul pompa jr., gil familier, vanessa thomas, george rothman, george whitacre, john ziemba, mark sweitzer, little george nowi, thayer evans and allison gallagher. i. e. yang if you don't wanna fly, the statistics means you are going to make your rocket arm - rolls. [CLS] savings that made the difference in someone's life cannot be made in another country without foreign assistance on their [CLS]",_ |
> > | MDLM - 16 | _"[CLS] mezzo ev of dream spirits. [CLS] to naval cadets i didn't want mum and dad to be a group, but even with the odd social butterflies i felt myself sitting at the edge of a deep blue pool. [CLS] on wednesday morning, fighter jets billoured the area during the early hours of the morning. [CLS] i know of the sailing course, the scenery, the coastline, the whole thing very beautiful - - the land of water. [CLS] wednesday, after its peak height reached the ice's lowest point. [CLS] reader's thought is. for the basic requirements of team leadership, internship, part time, a...",_ |
> > | MDLM - 1024           | _"[CLS] was killed as mr. giffords and as the caregiver, representative giffords's. [CLS] but i have no doubt that this process will produce meat, \" ms. rommel said. [CLS] and even if you don't need to hit an agency envelope during it, you should pay the rent through office expenses, \" referring to the possible error. [CLS] blog. view billshrift and discuss yahoo commercials, and what? [CLS] this month, the new logos of cin - armarge became scranton neck - based riverton and tollhorn. [CLS] when i arrived, he'd ducking behind the woman..",_ |
> > | UDLM - 16 | _"[CLS] grandeur. [CLS] grand avenue an. [CLS] it upon request, such information plays only in the \" provision of the protocol \" as guidelines for the nonprofit sector. [CLS] that too has to worry people about schools that shut down and closed schools for several weeks. [CLS] \" i will not comment on anything important to the eu, where someone is blaming serbia to not a figurehead. [CLS] but i think that because an increasing number of coal is put at risk as a displaced gateway, and the other party of technology falls away from communities that can somehow show up as a starting point, one thing that the province will suffer is loss of almost [CLS]",_ |
> > | UDLM - 1024 | _"[CLS] would [CLS] a 49 percent stake in the company under the terms of the deal. [CLS] the only subsequent trial out of ms, sykes is 19 february next year after prosecutors failed to overturn the death sentence against him. [CLS] but dugard has recorded several rings for baby jesus, a completely new thing as for calling muslim cold - callers and working with his confiscators. [CLS] i am forced to seek out convenient explanations while remaining in no position either to caucus. [CLS] of course, get me out of there. [CLS] the first thing being wrote about is having available food. [CLS] sir simon's guests included singer hayley wood and",_ |

---

> ### Author Response · Authors · 2024-11-26
> **Update regarding re-training LM1B with padding**
>
> As promised [above](https://openreview.net/forum?id=i5MrJ6g5G1&noteId=E4baJRed42), please see below for our updated numbers regarding LM1B PPL when sequences are right-padded to fill the context size of 128 tokens (which is more in-line with the data processing of previous works)
>
> | Method           | PPL ($\downarrow$) |
> |------------------|--------------------|
> | AR                   | 22.32             |
> | MDLM             | 27.04             |
> | UDLM              | 31.28             |
> | SEDD Absorb  | 32.79             |
> | SEDD Uniform  | 40.25             |
> | D3PM Absorb  | 77.50             |
> | D3PM Uniform | 137.9             |
>
> Notice that UDLM:
> - even outperforms SEDD Absorbing state,
> - is the best uniform noise discrete diffusion model,
> - and narrows the gap between comparable models that use absorbing state vs. uniform relative to previous work.

---

> ### Author Response · Authors · 2024-12-03
> **ImageNet Experiment Response**
>
> Dear Reviewer,
>
> At your suggestion, we have scaled our image generation experiment to ImageNet (128x128), keeping the setup analogous to our CIFAR-10 experiment. Similar to our previous findings on CIFAR-10, we find that **UDLM outperforms MDLM** in the context of guidance or fast sampling, and otherwise attains similar performance.
>
> **Improved Conditional Generation:**
>
> UDLM responds better to increasing the guidance strength parameter $\gamma$, which controls the strength of conditioning on a class label. This improves over the FID of MDLM.
>
> | Model | FID ($\downarrow$) |
> |----------|----------------------------|
> | $\gamma = 1$                      |
> | MDLM |                 19.48 |
> | UDLM |                       20.14 |
> | $\gamma = 2$                      |
> | MDLM |                        9.43 |
> | UDLM |                   **9.07** |
> | $\gamma = 3$                      |
> | MDLM |                      17.17 |
> | UDLM |                 14.35 |
>
> The above settings include a standard conditional diffusion model ($\gamma=1$), and one with an increased level of guidance ($\gamma=2$) conditioned on a class label. Guidance that is too strong ($\gamma=3$) eventually reduces quality, as in prior work on diffusion (Ho & Salimans 2022). We find that being able to revise generated tokens in UDLM improves controllability (and hence quality) over MDLM.
>
> **Improved Faster Sampling**
>
> UDLM maintains sample quality when we reduce the number of sampling steps $T$ by 2x and 4x, while the quality of MDLM degrades.
>
> | Model | FID ($\downarrow$) |
> |----------|----------------------------|
> | $T = 128$                            |
> | MDLM |                 **19.48** |
> | UDLM |                       20.14 |
> | $T = 64$                               |
> | MDLM |                      24.28 |
> | UDLM |                 **20.18** |
> | $T = 32$                               |
> | MDLM |                       32.86 |
> | UDLM |                 **20.34** |
>
> We use $\gamma= 1$ in the above table (i.e., we sample from a conditional diffusion model without additional guidance) to study the benefits of UDLM for fast sampling independently of guidance. We again observe that being able to revise generated tokens over the course of sampling allows UDLM to produce high quality images using a fraction of the steps of MDLM.
>
> **Details about this experiments:** For both MDLM and UDLM we train a 117M parameter DiT model for 400k iterations using the exact same hyperparameters. We trained on a subset of ImageNet that consisted of 100 randomly sampled classes (to limit dataset size to 128k images and finish training by the rebuttal deadline). Note that this strategy has been used by previous works, e.g., SupContrast (Khosla et al., 2020). We patched the image into 16x16 blocks (i.e., each image consists of 16x16 tokens) that were quantized to a vocabulary of size 16k (as in Sun et al 2024). We evaluated FID using 10k samples (due to computational and time considerations). Samples were generated using our discrete classifier guidance algorithm by first sampling a random class id, and then sampling an image conditioned on that id.
>
> We thank the reviewer again for their continued engagement with our work and hope that these latest results address the previously raised concerns.
>
> ---
>
> **References**
>
> Ho, Jonathan, and Tim Salimans. "Classifier-free diffusion guidance." arXiv preprint arXiv:2207.12598 (2022).
>
> Khosla, Prannay, et al. "Supervised contrastive learning." Advances in neural information processing systems 33 (2020): 18661-18673.
>
> Sun, Peize, et al. "Autoregressive Model Beats Diffusion: Llama for Scalable Image Generation." arXiv preprint arXiv:2406.06525 (2024).

---

### Official Review · Reviewer_9Khq · 2024-11-02

**Soundness:** 3
**Presentation:** 3
**Contribution:** 2
**Rating:** 6
**Confidence:** 3

**Summary:**

In this paper, the authors derive classifier-free and classifier-based guidance for discrete diffusion-model controllable generation.  The proposed guidance techniques build upon discrete diffusion techniques that interpolate between clean data and a noisy prior.   Empirically,   the proposed method outperforms Autoregressive models and several diffusion-model baselines on genomic sequences, small molecule design and CIFAR-10.

**Strengths:**

1.  The paper is well-written and well-organized.

2. The proposed method is simple and easy to follow.

3. It seems the proposed method outperforms baselines on several discrete tasks.

**Weaknesses:**

1.  The description of the contribution is not clear enough.  Some parts may be overclaimed.

I am not sure whether the claimed contribution is classifier-free and classifier-based guidance for a discrete diffusion-model controllable generation or if it further includes the interpolating discrete diffusion model parts.  In Line 58 on page 2, the authors claim they introduce "a class of discrete diffusion models particularly amenable to guidance."  This part may be overclaimed because the key technique involved is the simplified discrete diffusion process with the transition probabilities as the interpolation between the current state and a prior distribution. This technique may not be new. See the following point two.

2. Very similar discrete diffusion works are not discussed and compared.

In both [1] and [2],  similar simplified discrete diffusion processes with the transition probabilities as the interpolation between the current state and a prior are employed,  which is very related to the proposed method. However,  discussion and comparison with these works are missing.   As a result, the claimed SOTA performance is unconvincing.

3. What is the key advantage of the proposed method over other discrete diffusion methods is not clearly discussed.

What is the key advantage of the proposed method compared with other discrete diffusion methods? Why does the proposed method perform better than other discrete diffusion methods?

4.   The equitation in Line 181 on page 4 may not be correct.

Why does the log(p) term equal the inner product term?



[1] Zheng et al.   A Reparameterized Discrete Diffusion Model for Text Generation.  2023

[2] Zhao. et al.  Unified Discrete Diffusion for Categorical Data.   2024

**Questions:**

1. Please clarify the contribution of this paper as the point one listed in the above weakness part.

2. What is the relationship and difference between the proposed discrete diffusion method and the one in [1] and [2] listed above?

3. What is the key advantage of the proposed method compared with other discrete diffusion methods? Why does the proposed method perform better than other discrete diffusion methods?

4. Please explain about the equitation in Line 181 on page 4.  Why does the log(p) term equal the inner product term?

---

> ### Author Response · Authors · 2024-11-24
> **Response to Reviewer 9Khq (1/2)**
>
> We thank the reviewer for their detailed feedback. Please see below for the answers to the specific questions raised.
>
> ---
>
> #### **Concern 1** Which aspects of the work are novel?
>
> We clarify that our claim is not to have developed discrete diffusion with uniform noise, but rather the novelty comes from the following:
> 1. We present novel guidance mechanisms for discrete diffusion
>     - Our derivation for discrete classifier-based guidance is novel and finds a simple yet principled way to get around the issue that the classifier model need not factorize the same way as the generative decoding model
> 2. We introduced UDLM (analogy of MDLM (Sahoo et al., 2024) for uniform noise), which is more guideable. Specific novel components include a refined ELBO with terms analytically set to zero and a continuous time limit.
>
> ---
>
> #### **Concern 2** Missing citation to Zheng et al., 2023 and Zhao. et al., 2024
>
> We thank the reviewer for pointing out Zheng et al., 2023 and Zhao. et al., 2024. These are indeed highly relevant in terms of the formulation for the interpolating discrete diffusion, and we cite them in our revised manuscript along with Sahoo et al. 2024 when introducing this class of models.
>
> Comparisons to other uniform discrete diffusion models were done using the metric of perplexity on standard language benchmarks (e.g., text8 and LM1b). Neither Zheng et al. nor Zhao et al. report this metric nor do they test on these standard benchmarks. Furthermore, Zheng et al. does not admit a true variational bound and therefore cannot be compared.
>
> Over the coming week, we will attempt to reproduce Zhao et al. in our experiments and will update here once we have the results.
>
> Below we list differences between our work and these references.
>
> **Comparison to Zheng et al. 2023**
>
> Most importantly, we train with a principled lower bound. In contrast, the use of “teacher forcing” in Zheng et al. (see Section 4.2 and Appendix D in their work) allows them to derive a loss term that amounts to a simple weighted cross entropy term, but this loss is **not** to a true variational bound. Additionally, our loss is improved by analyzing the continuous time case. Zheng et al. only examine discrete time.
>
> **Comparison to Zhao et al. 2024**
>
> The main difference between Zhao et al. and our work is the parameterization (see below for more details). Additionally, Zhao et al. train models with an “augmented” loss where they add a cross-entropy term to the ELBO. Our models directly optimize the ELBO that we derive.
>
> The difference in parameterization is the following: We use mean parameterization, meaning that our network outputs a distribution over the “clean” data given a noisy input. We then plug this estimate, which we denote as $\mathbf{x}\_\theta$ into the pre-defined noise process $q_t(\mathbf{z} \mid \mathbf{x} = \mathbf{x}_\theta)$. We can use this parameterization to derive an equivalence between our ELBO and the one from SEDD (Lou et al. 2023) – please see the new Appendix B in the updated pdf we posted to OpenReview. To minimize the SEDD ELBO, one could use the following parameterization for the score:
> $$ s\_\theta(\mathbf{z})\_{\mathbf{z}’} := \frac{q\_t(\mathbf{z}’ \mid \mathbf{x} = \mathbf{x}\_\theta)}{q\_t(\mathbf{z} \mid \mathbf{x} = \mathbf{x}\_\theta)}$$
>
> In contrast Zhao et al., which also use a model to output a distribution over the clean data, parameterize the score as follows:
> $$ s\_\theta(\mathbf{z})\_{\mathbf{z}’} := \sum_{x}\frac{q_t(\mathbf{z}’ \mid \mathbf{x})}{q_t(\mathbf{z} \mid \mathbf{x})}  \mathbf{x}^\top\mathbf{x}_\theta$$

---

> > ### Author Response · Authors · 2024-11-24
> > **Response to Reviewer 9Khq (2/2)**
> >
> > #### **Concern 3** What is the key advantage of UDLM? Why does it outperform previous methods?
> >
> > **Advantage 1: Improved steerability relative to absorbing state models**
> >
> > One of the main motivations behind UDLM was that it can improve generation in settings where MDLM “locks in” certain predictions that it cannot change. In the experiment below, we make this clear by examining generation on the CIFAR-10 dataset with smaller $T$ where the diffusion models predict multiple pixels in parallel. MDLM’s performance deteriorates with smaller $T$ (it cannot recover from mistakes), whereas UDLM is robust to this setting:
> >
> > | $T$   | Model  | FID ($\downarrow$) | IS ($\uparrow$) | F1 ($\uparrow$) |
> > |--------|-----------|----------------------------|----------------------|------------------------|
> > | 128   | MDLM |                        64.09 |                  5.81 |                     0.63 |
> > | 128   | UDLM |                        30.48 |                  7.30 |                     0.80 |
> > | 256   | MDLM |                        41.53 |                  6.51 |                     0.70 |
> > | 256   | UDLM |                        28.30 |                  7.40 |                     0.81 |
> > | 512   | MDLM |                        31.97 |                  6.89 |                     0.74 |
> > | 512   | UDLM |                        27.12 |                  7.43 |                     0.74 |
> > | 1000 | MDLM |                        27.94 |                  7.14 |                     0.81 |
> > | 1000 | UDLM |                        26.70 |                  7.43 |                     0.81 |
> >
> > **Advantage 2: Improved language modeling quality vs. other uniform noise models**
> >
> > The key advantage of UDLM relative to previous discrete diffusion methods is that (1) certain terms in the variational lower bound are set analytically to zero (and therefore we do not need to learn to optimize them) and (2) taking $T \rightarrow \infty$ tightens the lower bound. Training with our refined objective (Equation 10) therefore leads to better models, as demonstrated in the bottom 3 rows of Tables 2 and 3, which compare UDLM to other uniform noise discrete diffusion models.
> >
> >
> > **Details about CIFAR10 experiment above:** We generate 50k images sampled from a conditional model (which is equivalent to using $\gamma=1$ and D-CFG). F1 is the accuracy of a separate classifier in identifying the class label used for the conditional generation.
> >
> > ---
> >
> > #### **Concern 4** Clarifying why the reconstruction loss term typo in Line 181
> >
> > We thank the reviewer for noticing this typo: there should indeed be a $\log$ preceding the inner product. We are updating our manuscript with the correction below to reflect this:
> > $$ \log p_\theta(\mathbf{x}_0 \mid \z_{\frac{1}{T}}) =\log( \langle \mathbf{x}_0, \mathbf{x}_\theta(\mathbf{z}_{\frac{1}{T}}, 1/T)) \rangle $$.
> >
> > ---
> >
> > **References:**
> >
> > Lou, Aaron, Chenlin Meng, and Stefano Ermon. "Discrete diffusion language modeling by estimating the ratios of the data distribution." arXiv preprint arXiv:2310.16834 (2023).
> >
> > Sahoo, Subham Sekhar, et al. "Simple and Effective Masked Diffusion Language Models." arXiv preprint arXiv:2406.07524 (2024).
> >
> > Zhao. et al. Unified Discrete Diffusion for Categorical Data. 2024
> >
> > Zheng et al. A Reparameterized Discrete Diffusion Model for Text Generation. 2023

---

> > > ### Comment · Reviewer_9Khq · 2024-11-27
> > >
> > > Thanks for the authors' detailed clarification and response.  Most of my concerns have been addressed.  The technical contribution and novelty have become clearer now.  Although I am still not sure about the practical advantage compared with the closely related interpolating discrete diffusion methods,  I lean to support it now. So, I would like to increase my score from 5 to 6.

---

### Official Review · Reviewer_2Wfu · 2024-11-03

**Soundness:** 3
**Presentation:** 3
**Contribution:** 3
**Rating:** 6
**Confidence:** 3

**Summary:**

This paper introduces a discrete diffusion model and guidance mechanism that is effective at controllable generation. The paper uses discrete classifier-based and classifier-free guidance, and introduces uniform noise diffusion language models that can continuously edit discrete data.

**Strengths:**

1. Adapting diffusion model to discrete data is an important task as it can be utilized to a few data generation tasks, such as molecule and text generation.

3. The experiments demonstrate the effectiveness of guidance with discrete diffusion models on several domains, and show that UDLM can achieve state-of-the-art performance on small vocabulary datasets.

**Weaknesses:**

1. In Introduction, "D-CBG" is introduced before being explained.

2. I would recommend the paper to explain the logic, why the proposed UDLM can make the guidance easier? To me I think the only difference is to reset the $\pi$ value in the proposed method. How can we understand the motivation for this setting and what's the intuition?

3. In the first paragraph of Section 3.2, the paper mentions that the proposed method yields a simple expression for ELBO, but I didnt see ELBO below. Also, I would suggest the paper use a table to compare the new and the old ELBO to make the improvement more clear.

**Questions:**

See weakness above.

**Details Of Ethics Concerns:**

The paper uses dataset regarding genomic sequences, not sure if it needs ethics review. The paper doesn't mention it.

---

> ### Author Response · Authors · 2024-11-24
> **Response to Reviewer 2Wfu (1/2)**
>
> We thank the reviewer for recognizing the contributions of our work and address specific questions and comments below
>
> ---
>
> ### **Concern 1:** Clarifying why UDLM improves guidance
>
> Absorbing state discrete diffusion models suffer from a drawback that is shared with AR models: once a token is predicted, it cannot be unchanged (see for example Equations 6 and 7 in Sahoo et al. 2024).
>
> In contrast, when using uniform noise as the limiting distribution, even if the model samples a particular token at some time $t$, that token can change as the sequence is denoised at times $s < t$. We therefore hypothesize that discrete diffusion models trained with uniform noise will be more “guidable” than absorbing state counterparts, since mistakes made during the decoding process can be corrected. This hypothesis is made more clear in the experiment below.
>
> We examine what happens when we decode with smaller time steps $T$. MDLM needs to decode multiple tokens per denoising step and these tokens cannot be changed at later denoising iterations. Thus the model can potentially be “locked-in” to mistakes. In contrast, UDLM can update any given token on every iteration. In this experiment for CIFAR10 generation, we find that MDLM’s performance deteriorates with smaller $T$, whereas UDLM is robust to this setting:
>
> | $T$   | Model  | FID ($\downarrow$) | IS ($\uparrow$) | F1 ($\uparrow$) |
> |--------|-----------|----------------------------|----------------------|------------------------|
> | 128   | MDLM |                        64.09 |                  5.81 |                     0.63 |
> | 128   | UDLM |                        30.48 |                  7.30 |                     0.80 |
> | 256   | MDLM |                        41.53 |                  6.51 |                     0.70 |
> | 256   | UDLM |                        28.30 |                  7.40 |                     0.81 |
> | 512   | MDLM |                        31.97 |                  6.89 |                     0.74 |
> | 512   | UDLM |                        27.12 |                  7.43 |                     0.74 |
> | 1000 | MDLM |                        27.94 |                  7.14 |                     0.81 |
> | 1000 | UDLM |                        26.70 |                  7.43 |                     0.81 |
>
> **Details about this experiment:** We generate 50k images sampled from a conditional model (which is equivalent to using $\gamma=1$ and D-CFG). F1 is the accuracy of a separate classifier in identifying the class label used for the conditional generation.
>
> ---
>
> ### **Concern 2:** Which equation is the ELBO in Section 3.2?
>
> Equation 10 represents the continuous time (i.e., $T \rightarrow \infty$) ELBO for uniform noise. We derive this by showing how $\mathcal{L}\_{recon}$ and $\mathcal{L}\_{prior}$ can be analytically set to zero and analyzing $\mathcal{L}\_{diffusion}$ in the infinite time limit. Equation 11 adapts this ELBO for sequences of multiple tokens and is what we use to train UDLM.
>
> We highlight the difference between an the unsimplified diffusion loss term from the standard ELBO and the one in our objective below:
>
> _Unsimplified_
>
> $T\cdot\frac{N\alpha\_t \mathbf{x}\_i + \alpha\_{t|s} - \alpha\_t + (\alpha\_s - \alpha\_t)\mathbf{x}\_i + \frac{(\alpha\_s - \alpha\_t)(1- \alpha\_s)}{\alpha\_s N}}{N \alpha\_t \mathbf{x}\_i  + 1 - \alpha\_t} \cdot \log\Bigg[\frac{\frac{N \alpha\_t \mathbf{x}\_i + \alpha\_{t|s} - \alpha\_t + (\alpha\_s - \alpha\_t)\mathbf{x}\_i + \frac{(\alpha\_s - \alpha\_t)(1 - \alpha\_s)}{\alpha\_s N}}{N \alpha\_t \mathbf{x}\_i  + 1 - \alpha\_t}}{\frac{N \alpha\_t(\mathbf{x}\_\theta)_i + \alpha\_{t|s} - \alpha\_t + (\alpha\_s - \alpha\_t)(\mathbf{x}\_\theta)_i + \frac{(\alpha\_s - \alpha\_t)(1 - \alpha\_s)}{\alpha\_s N}}{N \alpha\_t (\mathbf{x}\_\theta)_i  + 1 - \alpha\_t}}\Bigg] + T\cdot \sum\_{j \in [N]: (\mathbf{z}\_t)\_j = 0} \frac{(\alpha\_s - \alpha\_t)\mathbf{x}\_j + \frac{(\alpha\_s - \alpha\_t)(1-\alpha\_s)}{N \alpha\_s}}{N \alpha\_t\mathbf{x}\_i + 1 - \alpha\_t}\log\Bigg[\frac{\frac{(\alpha\_s - \alpha\_t)\mathbf{x}\_j + \frac{(\alpha\_s -\alpha\_t)(1-\alpha\_s)}{N \alpha\_s}}{N \alpha_t\mathbf{x}\_i + 1 -\alpha\_t}}{\frac{(\alpha\_s-\alpha\_t)(\mathbf{x}\_\theta)\_j +\frac{(\alpha\_s - \alpha\_t)(1-\alpha\_t)}{N \alpha\_s}}{N \alpha\_t(\mathbf{x}\_\theta)\_i + 1 - \alpha\_t}}\Bigg]$
>
> _Ours_
>
> $\frac{\alpha\_t'}{N\alpha\_t}\Bigg[\frac{N}{N\alpha\_t\mathbf{x}\_i + 1 - \alpha\_t} - \frac{N}{N\alpha\_t(\mathbf{x}\_\theta)\_i + 1 - \alpha\_t}\Bigg]  +\frac{-\alpha_t'}{N\alpha\_t}\sum\_{j \in [N] : (\mathbf{z}_t)_j = 0} \Bigg(\frac{N\alpha_t\mathbf{x}\_j + 1 -\alpha\_t}{N\alpha\_t\mathbf{x}\_i + 1 -\alpha\_t}\Bigg)\log\Bigg[\Bigg(\frac{N\alpha\_t(\mathbf{x}\_\theta)\_i + 1 - \alpha\_t}{N\alpha\_t(\mathbf{x}\_\theta)\_j + 1 - \alpha\_t}\Bigg)\Bigg(\frac{N\alpha_t\\mathbf{x}\_j + 1 - \alpha\_t}{N\alpha\_t\mathbf{x}\_i + 1 - \alpha\_t}\Bigg)\Bigg]$

---

> ### Author Response · Authors · 2024-11-24
> **Response to Reviewer 2Wfu (2/2)**
>
> ### **Concern 3:** D-CBG is mentioned in the intro before being defined
>
> Thank you for bringing this to our attention. We have corrected this in our revised manuscript.
>
> ---
>
> **References**
>
> Ou, Jingyang, et al. "Your Absorbing Discrete Diffusion Secretly Models the Conditional Distributions of Clean Data." arXiv preprint arXiv:2406.03736 (2024).
>
> Sahoo, Subham Sekhar, et al. "Simple and Effective Masked Diffusion Language Models." arXiv preprint arXiv:2406.07524 (2024).
>
> Shi, Jiaxin, et al. "Simplified and Generalized Masked Diffusion for Discrete Data." arXiv preprint arXiv:2406.04329 (2024).

---

> ### Comment · Reviewer_2Wfu · 2024-11-25
> **Thanks for the response**
>
> Thank the author for the response. My concerns have been addressed so I would maintain my score to support this paper.

---

### Official Review · Reviewer_LFVk · 2024-11-04

**Soundness:** 2
**Presentation:** 2
**Contribution:** 3
**Rating:** 6
**Confidence:** 3

**Summary:**

This paper presents several advances in discrete diffusion models, specifically focusing on uniform noise diffusion and guidance mechanisms. The authors make three main contributions:

-  developing classifier-based and classifier-free guidance for discrete diffusion models,

-  introducing Uniform Diffusion Language Models (UDLM) with an improved variational bound

-  demonstrating superior controlled generation compared to autoregressive baselines across multiple domains.

**Strengths:**

-  Theoretical foundation with clear derivations of the continuous-time variational bound

- Practical guidance mechanisms that are easy to implement

- Thorough ablation studies and hyperparameter analysis

**Weaknesses:**

The primary limitation is the scope of improvements being mainly restricted to multinomial diffusion with small vocabulary settings, with a persistent performance gap in larger vocabulary NLP tasks. The paper lacks computational complexity analysis and detailed runtime comparisons. Some hyperparameter choices, particularly sampling steps T, would benefit from more thorough justification. The theoretical analysis of guidance mechanisms could be more extensive, and more exploration of failure cases would strengthen the work.

**Questions:**

- What are the primary factors limiting performance on larger vocabularies, and how might these be addressed?

- How to understand equation 9, does it have any connection with the loss in Lou 2023?

- How does the choice of sampling steps T affect the trade-off between generation quality and computational cost? Is there a systematic way to choose optimal T?

- Have the authors explored potential failure modes of their guidance mechanisms, especially in cases where the guidance strength $\gamma$ is high?

---

> ### Author Response · Authors · 2024-11-24
> **Response to Reviewer LFVk**
>
> We thank the reviewer for their feedback and for recognizing the contributions of our work. In the comments below we respond to the specific concerns and questions raised by the reviewer.

---

> > ### Author Response · Authors · 2024-11-24
> > **Concern 1: “Scope of improvements is mainly restricted to small vocabulary settings, with a gap in larger vocabulary NLP tasks”**
> >
> > Our improvements are not restricted to small vocabulary settings. In the two experiments below we show that UDLM + guidance is useful in large vocabulary NLP settings,  and we also demonstrate why UDLM could be preferred to MDLM.
> >
> > ### **Experiment 1: Large vocabulary NLP setting**
> >
> > | Model  | Guidance                        | Gen. PPL ($\downarrow$) | F1 ($\uparrow$) |
> > |-----------|---------------------------------|-------------------------------------|----------------------|
> > | AR       | D-CFG ($\gamma = 1$) | **9.80**                                    | 19 |
> > | AR       | D-CFG ($\gamma = 2$) | 10.18                                    | 20 |
> > | AR       | D-CFG ($\gamma = 4$) | 10.32                                    | 19 |
> > | MDLM | D-CFG ($\gamma = 1$) | 9.92                                    | 85 |
> > | MDLM | D-CFG ($\gamma = 2$) | 10.73                                    | 85 |
> > | MDLM | D-CFG ($\gamma = 4$) | 13.83                                   | 78 |
> > | UDLM | D-CFG ($\gamma = 1$) | 10.23                                    | 83 |
> > | UDLM | D-CFG ($\gamma = 2$) | 10.34                                    | 85 |
> > | UDLM | D-CFG ($\gamma = 4$) | 10.82                                    | **86** |
> >
> > We find that **Diffusion models (both MDLM and UDLM) is able to outperform AR models,** with comparable sample quality (generative perplexity) and much higher control scores (F1). Additionally, we see that **only UDLM seems to respond to higher $\gamma$ tuning compared to MDLM and AR**.
> >
> > ---
> >
> > ### **Experiment 2: UDLM outperforms MDLM for smaller T (CIFAR-10 generation)**
> >
> > | $T$   | Model  | FID ($\downarrow$) | IS ($\uparrow$) | F1 ($\uparrow$) |
> > |--------|-----------|----------------------------|----------------------|------------------------|
> > | 128   | MDLM |                        64.09 |                  5.81 |                     0.63 |
> > | 128   | UDLM |                        30.48 |                  7.30 |                     0.80 |
> > | 256   | MDLM |                        41.53 |                  6.51 |                     0.70 |
> > | 256   | UDLM |                        28.30 |                  7.40 |                     0.81 |
> > | 512   | MDLM |                        31.97 |                  6.89 |                     0.74 |
> > | 512   | UDLM |                        27.12 |                  7.43 |                     0.74 |
> > | 1000 | MDLM |                        27.94 |                  7.14 |                     0.81 |
> > | 1000 | UDLM |                        26.70 |                  7.43 |                     0.81 |
> >
> > In the experiment above, as we use smaller $T$, the diffusion models predict multiple pixels in parallel. MDLM’s performance deteriorates with smaller $T$, whereas UDLM is robust to this setting. This validates one of main motivations behind UDLM: in settings where MDLM “locks in” certain predictions that it cannot change, UDLM is more robust given that all tokens can change throughout the decoding process.
> >
> > While it is true that in several unconditional settings UDLM has worse perplexity than MDLM, this paper focuses on controlled generation, and in all settings UDLM is better than all previous uniform noise discrete diffusion, i.e., D3PM (Austin et al.) and SEDD (Lou et al.).
> >
> > ---
> >
> > **Details about Experiment 1:** We train models on the E2E dataset, which consists of 50k restaurant reviews (Novikova et al. 2017). We use the `bert-base-uncased` tokenizer which has vocab. size of ~30k tokens. The goal is to generate sentences that follow a part-of-speech template, which acts as the guidance label. Quality is measured by generative perplexity and controllability is measured as the F1 score between the POS tagging of the generated sequences and that provided as the control target.)
> >
> > **Details about this Experiment 2:** We generate 50k images sampled from a conditional model (which is equivalent to using $\gamma=1$ and D-CFG). F1 is the accuracy of a separate classifier in identifying the class label used for the conditional generation.

---

> > > ### Author Response · Authors · 2024-11-24
> > > **Concern 2: Paper lacks computational complexity analysis and detailed runtime comparisons**
> > >
> > > We thank the reviewer for this suggestion. In the table below, we provide complexity analysis and runtimes for the guidance mechanisms explored in this work:
> > > | Method                        | Complexity                                    | Runtime (sec) |
> > > |-------------------------------|----------------------------------------------|------------|
> > > | _Classifier-free_                                                                                 |
> > > | D-CFG (for AR)           | $\mathcal{O}(L p_\theta)$              |       0.8 |
> > > | D-CFG (for Diffusion)  | $\mathcal{O}(T p_\theta)$              |       0.8 |
> > > | _Classifier-based_                                                                              |
> > > | FUDGE (for AR)          | $\mathcal{O}(L\cdot(p_\theta + p_\phi\cdot k))$ |      1.1 |
> > > | D-CBG (for Diffusion)* | $\mathcal{O}(TLN\cdot(p_\theta + p_\phi))$   |       38.7 |
> > > | D-CBG w/first-order approx. (for Diffusion)* | $\mathcal{O}(T\cdot(p_\theta + p_\phi))$   |       1.9 |
> > > | PPLM (for AR)             | $\mathcal{O}(L\cdot ((2K + 2) * (p_{\theta, backbone} + K * p_{\phi}))$ |       18.4 |
> > > | NOS (for Diffusion)      | $\mathcal{O}(T\cdot (p_{\theta, backbone} + K * (p_\phi + p_{\theta,head})$   |       2.8 |
> > >
> > > **Explanation of complexity notation:**
> > > - $L$ is sequence length
> > > - $T$ is the number of diffusion steps
> > > - $N$ is the vocabulary size
> > > - $p_\theta$ is the complexity of a forward pass through the language model (AR and Diffusion differ only in the attention mask used which does not affect big-O analysis)
> > > - $p_\phi$ is the complexity of a forward/backward pass through the classifier model
> > > - For FUDGE, $k$ is the _top k_ hyperparameter that controls how many candidates to pass to the classifier model
> > > - For PPLM/NOS, $K$ is the number of langevin updates we make to the hidden states and we split up the runtime of the language model into the backbone that produces hidden states ($p_{\theta, backbone}$) and the language modeling head ($p_{\theta, head}$), which typically involves a simple embedding weight matrix multiplication of the final hidden states to project them into the size of the vocabulary.
> > >     - Note PPLM operates on all hidden states whereas NOS only uses the last hidden state leading to a large gap in the runtime of these two approaches.
> > >
> > > **Runtime details:** Runtimes are based on decoding speed for generating a single batch of 16 samples with $L = T = 32$ samples on a single A6000 GPU. For diffusion runtimes, we use UDLM. We use models trained on the QM9 dataset ($N = 40$) and guidance is used to maximize the QED property with $\gamma=2$. For FUDGE, we set the _top k_ parameter equal to the full vocabulary size. For NOS/PPLM we use $K=5$.

---

> > > > ### Author Response · Authors · 2024-11-24
> > > > **Concern 3: Better exploration of the effect of $T$?**
> > > >
> > > > We thank the reviewer for this suggested analysis. Below we present an analysis which we will include in our updated manuscript.
> > > >
> > > > _CIFAR 10_
> > > >
> > > > First, the CIFAR-10 table presented above shows a clear relationship between increasing $T$ and sample quality. Of note, UDLM is far more robust to decoding with smaller $T$ compared to MDLM, since as hypothesized, MDLM locks-in potentially poor choices for certain pixels/tokens when decoding in parallel.
> > > >
> > > > _Species10_
> > > >
> > > > Similarly, in the DNA setting, we find that increasing $T$ directly impacts sample quality (as measured by the AUROC of an external discriminator trained to distinguish between real and generated samples)
> > > >
> > > > | $T$   | Model  | Disc AUCROC ($\downarrow$) |
> > > > |--------|-----------|-------------------------------------------|
> > > > | 64     | MDLM |                                           0.58 |
> > > > | 128   | MDLM |                                          0.56 |
> > > > | 256   | MDLM |                                          0.51 |
> > > > | 64     | UDLM |                                           0.54 |
> > > > | 128   | UDLM |                                           0.52 |
> > > > | 256   | UDLM |                                           0.48 |

---

> > > > > ### Author Response · Authors · 2024-11-24
> > > > > **Concern 4: What are the failure cases, especially with large guidance strength?**
> > > > >
> > > > > Across settings we observe that there exists some $\gamma$ strength after which sample quality begins to deteriorate. For example, in the QM9 experiment (small vocab and short sequence lengths) we found that novelty of generated molecules is greatly impacted by increasing $\gamma$, i.e., forcing the model to focus on modes of the conditional distribution tends to interfere with diversity of generation. In the Species10 generation (small vocabulary and long sequences) we found that generation was highly sensitive to $\gamma$ and even increasing beyond $\gamma = 1$ led to sample quality reduction, especially for AR and MDLM models.

---

> > > > > > ### Author Response · Authors · 2024-11-24
> > > > > > **Question 1: What are the primary factors limiting performance on larger vocabularies, and how might these be addressed?**
> > > > > >
> > > > > > For uniform discrete diffusion, the main hurdle for language modeling with large vocabularies is a combinatorial one:
> > > > > >
> > > > > > For absorbing state diffusion, the corruption and denoising processes are constructed such that only “masked” tokens need to be denoised. That is, if the absorbing-state denoising model sees a sequence with $K$ masked tokens, then to predict the next step of latents, the model does not need to compute the $L - K$ unmasked locations, since these will not change between iterations, as dictated by the pre-defined corruption process. Thus the space of possible sequences $\mathbf{z}_s^{1:L}$ given $\mathbf{z}_t^{(1:L)}$ is $N^K$, where $N$ is the size of the vocabulary.
> > > > > >
> > > > > > In contrast, for uniform noise discrete diffusion, the model cannot immediately infer which tokens have been corrupted and which need to remain unchanged, but rather must learn to identify this. Thus given $\mathbf{z}_t^{(1:L)}$, the model needs to predict the denoised latents from among $N^L$ possible sequences regardless of how many tokens have actually been corrupted. Therefore, we believe that in smaller vocabulary regimes where the search space of possible latents between denoising iterations is more similar for absorbing state and uniform noise discrete diffusion models, these two models will be more on par with each other.

---

> > > > > > > ### Author Response · Authors · 2024-11-24
> > > > > > > **Question 2: Does UDLM ELBO have any connection with the loss in Lou et al. 2023?**
> > > > > > >
> > > > > > > This is a great question; the connection to score-based parameterization is interesting. In the updated manuscript posted to OpenReview, we added a new Appendix B where we show that the lower bounds are equivalent.

---

> > > > > > > > ### Author Response · Authors · 2024-11-24
> > > > > > > > **References**
> > > > > > > >
> > > > > > > > Austin, Jacob, et al. "Structured denoising diffusion models in discrete state-spaces." Advances in Neural Information Processing Systems 34 (2021): 17981-17993.
> > > > > > > >
> > > > > > > > Lou, Aaron, Chenlin Meng, and Stefano Ermon. "Discrete Diffusion Modeling by Estimating the Ratios of the Data Distribution." Forty-first International Conference on Machine Learning.

---

> > > > > > > > > ### Comment · Reviewer_LFVk · 2024-11-24
> > > > > > > > >
> > > > > > > > > Thank you for your response. Your E2E experiments and complexity analysis effectively address my practical concerns about UDLM's performance. Based on these clarifications, I am revising my rating from 5 to 6. Further exploration of even larger vocabulary settings could strengthen the work, but this can be addressed in future research.

---

> > > > > > > > > > ### Author Response · Authors · 2024-11-25
> > > > > > > > > > **Thank you**
> > > > > > > > > >
> > > > > > > > > > We thank the reviewer for engaging with our response and for adjusting their score. We appreciate all the helpful feedback during this process.

---

### Author Response · Authors · 2024-12-03
**Summary of feedback and improvements (1/2)**

We thank the reviewers for their time and useful comments on our work.

We summarize comments that were common to several reviewers and our high level responses here. For details please refer to the specific responses posted to each reviewer. Summary of added experiments is as follows:
- New large-scale Imagenet (128x128 pixels) experiment with large vocabulary (16k tokens) that shows **UDLM outperforming MDLM in conditional generation and faster inference settings**
- New large-vocabulary (30k tokens) experiment on generating sentences with specific part-of-speech tags that shows how **diffusion models are more guidable than AR** and **UDLM outperforms MDLM**
- New analysis probing the effect of faster sampling on diffusion models in CIFAR-10 generation. As above, we find that **UDLM is more robust and has less image quality degradation (since it can correct mistakes) relative to MDLM** as sampling steps $T$ decrease
- Comparison of language modeling using UDLM ELBO vs. objective proposed in Discrete Flow Matching (Gat et al. 2024) that shows **training with our objective leads to better performance**.

In the comment below we provide more details about these experiments and the specific concerns they addressed.

---

> ### Author Response · Authors · 2024-12-03
> **Summary of feedback and improvements (2/2)**
>
> ### **Concern:** Does UDLM provide benefits over MDLM (especially with larger vocabularies)?
>
> Our new results demonstrate that UDLM outperforms MDLM in the context of guidance or fast sampling, and otherwise attains similar performance in many settings.
>
> **Benefits of UDLM for Guidance**
>
> _ImageNet_
>
> We scaled our image generation experiment to ImageNet (128x128), which was quantized to a large vocabulary of 16k tokens.
>
> **UDLM responds better to increasing the guidance strength parameter $\gamma$,** which controls the strength of conditioning on a class label. This improves over the FID of MDLM.
>
> | Model | FID ($\downarrow$) |
> |---|---|
> | $\gamma = 1$ |
> | MDLM | 19.48 |
> | UDLM | 20.14 |
> | $\gamma = 2$ |
> | MDLM | 9.43 |
> | UDLM | **9.07** |
> | $\gamma = 3$ |
> | MDLM | 17.17 |
> | UDLM | 14.35 |
>
> The above settings include a standard conditional diffusion model ($\gamma=1$), and one with an increased level of guidance ($\gamma=2$) conditioned on a class label. Guidance that is too strong ($\gamma=3$) eventually reduces quality, as in prior work on diffusion (Ho & Salimans 2022). We find that being able to revise generated tokens in UDLM improves controllability (and hence quality) over MDLM.
>
> _NLP_
>
> We added an NLP experiment with a large vocabulary (~30k) that demonstrates the benefits of UDLM and our guidance methods vs AR. The goal is to generate sentences that follow a part-of-speech template, which acts as the guidance label. We show that **diffusion models with our guidance outperform AR**. Moreover, **UDLM is competitive with MDLM on generation quality and responds better to higher guidance weights.**
>
> | Model  | Guidance | Gen. PPL ($\downarrow$) | F1 ($\uparrow$) |
> |---|---|---|---|
> | AR       | D-CFG ($\gamma = 1$) | **9.80** | 19 |
> | AR       | D-CFG ($\gamma = 2$) | 10.18 | 20 |
> | AR       | D-CFG ($\gamma = 4$) | 10.32 | 19 |
> | MDLM | D-CFG ($\gamma = 1$) | 9.92 | 85 |
> | MDLM | D-CFG ($\gamma = 2$) | 10.73 | 85 |
> | MDLM | D-CFG ($\gamma = 4$) | 13.83 | 78 |
> | UDLM | D-CFG ($\gamma = 1$) | 10.23 | 83 |
> | UDLM | D-CFG ($\gamma = 2$) | 10.34 | 85 |
> | UDLM | D-CFG ($\gamma = 4$) | 10.82 | **86** |
>
>
> **Benefits of UDLM in Faster Sampling Settings**
>
> _ImageNet_
>
> **UDLM maintains sample quality when we reduce the number of sampling steps $T$** by 2x and 4x, while the quality of MDLM degrades.
>
> | Model | FID ($\downarrow$) |
> |---|---|
> | $T = 128$ |
> | MDLM | **19.48** |
> | UDLM | 20.14 |
> | $T = 64$ |
> | MDLM | 24.28 |
> | UDLM | **20.18** |
> | $T = 32$ |
> | MDLM | 32.86 |
> | UDLM | **20.34** |
>
> We use $\gamma= 1$ in the above table (i.e., we sample from a conditional diffusion model without additional guidance) to study the benefits of UDLM for fast sampling independently of guidance. We again observe that being able to revise generated tokens over the course of sampling allows UDLM to produce high quality images using a fraction of the steps of MDLM.
>
> _CIFAR10_
>
> For CIFAR10, as with ImageNet, with smaller $T$ where the diffusion models predict multiple pixels in parallel we find a setting where MDLM’s performance deteriorates, whereas UDLM is more robust:
>
> | $T$   | Model  | FID ($\downarrow$) | IS ($\uparrow$) | F1 ($\uparrow$) |
> |---|---|---|---|---|
> | 128   | MDLM | 64.09 | 5.81 | 0.63 |
> | 128   | UDLM | 30.48 | 7.30 | 0.80 |
> | 256   | MDLM | 41.53 | 6.51 | 0.70 |
> | 256   | UDLM | 28.30 | 7.40 | 0.81 |
> | 512   | MDLM | 31.97 | 6.89 | 0.74 |
> | 512   | UDLM | 27.12 | 7.43 | 0.74 |
> | 1000 | MDLM | 27.94 | 7.14 | 0.81 |
> | 1000 | UDLM | 26.70 | 7.43 | 0.81 |
>
> ---
>
> ### **Concern:** Is UDLM novel relative to DFM/MDLM and other work?
>
> **Contributions of our work**
>
> 1. We present novel guidance mechanisms for discrete diffusion. This is a core focus of our work and is novel relative to both DFM (Gat et al. 2024) no MDLM (Sahoo et al. 2024)
> 2. We introduced UDLM (analogy of MDLM for uniform noise), which is more guideable. Specific novel components include a refined ELBO with terms analytically set to zero and a continuous time limit.
>
> **Training with UDLM ELBO outperforms DFM**
>
> We first make clear that DFM is concurrent work with ours. Moreover, we find that using the training objective proposed in DFM leads to worse language modeling performance (as measured by PPL, the standard quality metric):
>
> | Dataset | Training Objective | Validation ELBO |
> |---|---|---|
> | QM9 | Cross-entropy (as in DFM) | 2.99 |
> | QM9 | UDLM ELBO | **2.02** |
> | Species10  | Cross-entropy (as in DFM) | 5.20 |
> | Species10  | UDLM ELBO | **3.15** |

---

### Meta-Review · Area_Chair_XgnH · 2024-12-21

**Metareview:**

The paper focuses on discrete diffusion models, proposing classifier-based and classifier-free guidance mechanisms, introducing Uniform Noise Diffusion Language Models (UDLM), and improving performance through a continuous-time ELBO objective. The reviewers found that the paper is well-written and well-organized, the derivation of a simplified ELBO for discrete diffusion with uniform noise offers theoretical and practical improvements, and the proposed guidance mechanism is practical and easy to implement. The reviewers raised concerns about novelty compared to concurrent works, particularly MDLM, leading to scepticism about the contributions, the performance of UDML on large-scale language tasks (e.g., LM1B) showed limited improvement compared to MDLM, scepticism about scalability, and missing complexity analysis and runtime comparisons. During the rebuttal, the authors added new experiments addressing scalability concerns (ImageNet, LM1B), and clarifications on distinctions from related works were provided, though some reviewers found these insufficient. Given that the novelty concerns are partially related to concurrent work and that the authors addressed the majority of the concerns raised by the reviewers, leading to the majority of the reviewers recommending a marginal acceptance, I recommend accepting the paper.

**Additional Comments On Reviewer Discussion:**

As mentioned in the meta-review, the authors addressed many of the concerns raised by the reviewers in the reburial and discussion phase. The main outstanding concern, particularly with Reviewer S2DB, was related to novelty.

---

### Decision · Program_Chairs · 2025-01-22

Accept (Poster)